# Brain-wide representations of prior information in mouse decision-making

Charles Findling[1✉], Félix Hubert[1], International Brain Laboratory*, Luigi Acerbi[2], Brandon Benson[3], Julius Benson[4], Daniel Birman[5], Niccolò Bonacchi[6], E. Kelly Buchanan[3], Sebastian Bruijns[7], Matteo Carandini[8], Joana A. Catarino[9], Gaelle A. Chapuis[1], Anne K. Churchland[10], Yang Dan[11], Felicia Davatolhagh[10], Eric E. J. DeWitt[9], Tatiana A. Engel[12], Michele Fabbri[9], Mayo A. Faulkner[8], Ila Rani Fiete[13], Laura Freitas-Silva[9], Berk Gerçek[1], Kenneth D. Harris[8], Michael Häusser[8,14], Sonja B. Hofer[15], Fei Hu[11], Julia M. Huntenburg[7], Anup Khanal[10], Chris Krasniak[16], Christopher Langdon[12], Christopher A. Langfield[17], Peter E. Latham[18], Petrina Y. P. Lau[8], Zach Mainen[9], Guido T. Meijer[9], Nathaniel J. Miska[15], Thomas D. Mrsic-Flogel[15], Jean-Paul Noel[19], Kai Nylund[5], Alejandro Pan-Vazquez[12], Liam Paninski[17], Jonathan Pillow[12], Cyrille Rossant[8], Noam Roth[5], Rylan Schaeffer[13], Michael Schartner[9], Yanliang Shi[12], Karolina Z. Socha[8], Nicholas A. Steinmetz[5], Karel Svoboda[20], Charline Tessereau[7], Anne E. Urai[21], Miles J. Wells[8], Steven Jon West[15], Matthew R. Whiteway[17], Olivier Winter[9], Ilana B. Witten[12], Anthony Zador[16], Yizi Zhang[17], Peter Dayan[7] & Alexandre Pouget[1]

The neural representations of prior information about the state of the world are poorly understood[1]. Here, to investigate them, we examined brain-wide Neuropixels recordings and widefield calcium imaging collected by the International Brain Laboratory. Mice were trained to indicate the location of a visual grating stimulus, which appeared on the left or right with a prior probability alternating between 0.2 and 0.8 in blocks of variable length. We found that mice estimate this prior probability and thereby improve their decision accuracy. Furthermore, we report that this subjective prior is encoded in at least 20% to 30% of brain regions that, notably, span all levels of processing, from early sensory areas (the lateral geniculate nucleus and primary visual cortex) to motor regions (secondary and primary motor cortex and gigantocellular reticular nucleus) and high-level cortical regions (the dorsal anterior cingulate area and ventrolateral orbitofrontal cortex). This widespread representation of the prior is consistent with a neural model of Bayesian inference involving loops between areas, as opposed to a model in which the prior is incorporated only in decision-making areas. This study offers a brain-wide perspective on prior encoding at cellular resolution, underscoring the importance of using large-scale recordings on a single standardized task.

The ability to combine sensory information with prior knowledge through probabilistic inference is crucial for perception and cognition. In simple cases, inference is performed near-optimally by the brain, following key precepts of Bayesian decision theory[1–5]. For example, when interpreting a visual scene, we assume a priori that light comes from above—a sensible assumption that enables us to interpret otherwise ambiguous images[4].

Although much theoretical work has been devoted to the neural representation of Bayesian inference[6–9], it remains unclear where and how prior knowledge is represented in the brain. At one extreme, the brain might combine prior information with sensory evidence in high-level decision-making brain regions, right before decisions are turned into actions. This would predict that prior information is encoded only in late stages of processing, as has indeed been reported in parietal, orbitofrontal and prefrontal cortical areas[10–16]. At the other extreme, the brain might operate like a very large Bayesian network, in which probabilistic inference is the modus operandi in all brain regions and inference can be performed in all directions[17–22]. This would allow neural circuits to infer beliefs over variables from observations of arbitrary combinations of other variables. For example, after seeing an object, the brain might be able to infer its auditory and tactile properties; but could just as well perform the reverse inference, that is, predicting its

[1]University of Geneva, Geneva, Switzerland. [2]University of Helsinki, Helsinki, Finland. [3]Stanford University, Stanford, CA, USA. [4]New York University, New York, NY, USA. [5]University of Washington, Seattle, WA, USA. [6]William James Center for Research, Instituto Universitário, Lisbon, Portugal. [7]Max Planck Institute, University of Tübingen, Tübingen, Germany. [8]University College London, London, UK. [9]Champalimaud Foundation, Lisbon, Portugal. [10]University of California Los Angeles, Los Angeles, CA, USA. [11]University of California Berkeley, Berkeley, CA, USA. [12]Princeton University, Princeton, NJ, USA. [13]Massachusetts Institute of Technology, Cambridge, MA, USA. [14]The University of Hong Kong, Hong Kong, China. [15]Sainsbury Wellcome Centre, University College London, London, UK. [16]Cold Spring Harbor Laboratory, Cold Spring Harbor, NY, USA. [17]Columbia University, New York, NY, USA. [18]Gatsby Computational Neuroscience Unit, University College London, London, UK. [19]Department of Neuroscience, University of Minnesota, Minneapolis, MN, USA. [20]The Allen Institute for Neural Dynamics, Seattle, Washington, WA, USA. [21]Leiden University, Leiden, The Netherlands. *A list of authors and their affiliations appears at the end of the paper. ✉e-mail: charles.findling@internationalbrainlab.org

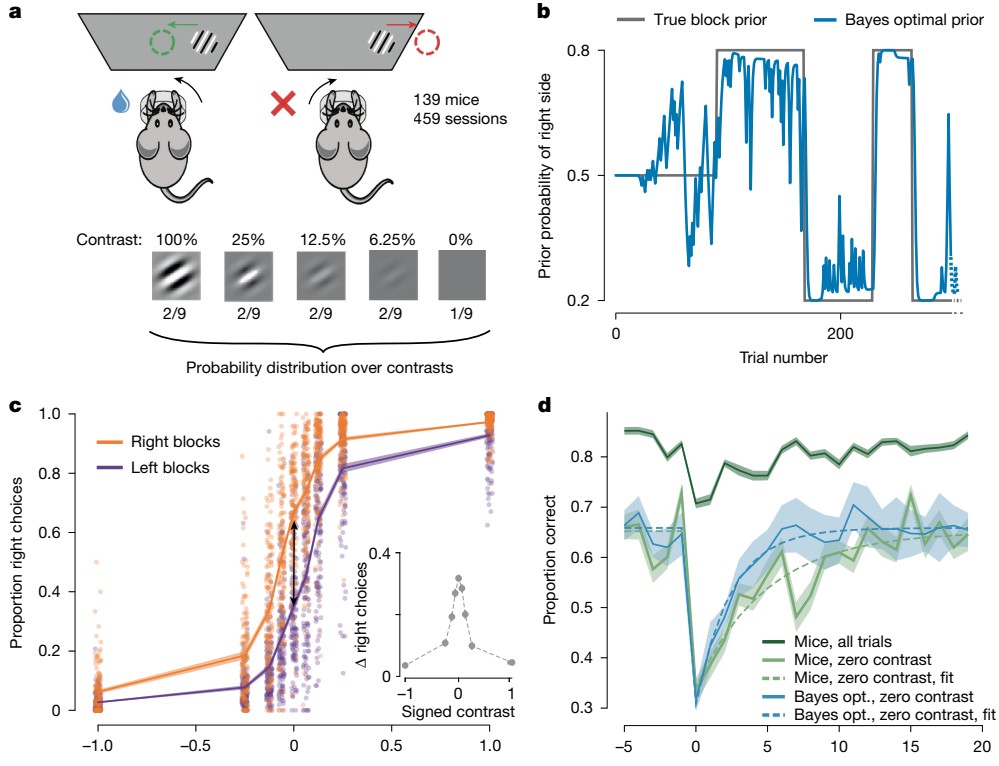

**Fig. 1 | Mice use the block prior to improve performance. a**, Mice had to move a 35° peripheral visual grating to the centre of the screen by turning a wheel with their front paws. The contrast of the visual stimulus varied from trial to trial. Adapted from ref. 33, *eLife*, under a CC BY 4.0 license. **b**, The prior probability that the stimulus appeared on the right side was maintained at either 0.2 or 0.8 over blocks, after an initial block of 90 trials during which the prior was set to 0.5. The block length was drawn from a truncated exponential distribution (20–100 trials, scale = 60). After a wheel turn, the mice were provided with positive feedback (water reward) or negative feedback (white noise pulse and timeout). The next trial began after a delay and a quiescence period that was uniformly sampled between 400–700 ms during which the mice had to hold the wheel still. **c**, Psychometric curves averaged across animals and sessions and conditioned on block identity, plotted as a function of signed contrast (negative values corresponding to stimulus on the left, positive values to stimulus on the right). The proportion of right choices on zero-contrast trials was different across blocks (Wilcoxon signed-rank test: $t = 15$, $P = 2.04 \times 10^{-24}$, $n = 139$) and displaced in the direction predicted by the true block prior (double arrow). Inset: the difference between curves. **d**, Reversal curves showing the percentage of correct responses after the block switches. The average performance across all animals and all contrasts is shown (dark green). The light green line shows the same as the dark green line, but for zero-contrast trials. The performance of an observer generating choices stochastically according to the Bayes-optimal estimate of the prior is shown (blue). This simulation was limited to zero-contrast trials to focus on the influence of prior knowledge without stimulus information (Extended Data Fig. 1 and Methods). Dashed curves are exponential fits (Extended Data Fig. 1 and Methods). Shaded region shows the s.e.m. across mice for the curves showing mouse behaviour (light and dark green curves) and the s.d. for the Bayes-optimal model (blue curve), as there is no interindividual variability to account for.

visual appearance after hearing or touching it. Such a model would predict that prior information should be available throughout the brain, even in low-level cortical sensory areas[18,19,21,22]. The current literature offers a contradictory and, therefore, inconclusive perspective on whether the prior is indeed encoded in brain regions associated with early processing[11,16,19,23–30]. This is because past studies have collectively recorded from only a limited set of areas and, as they use different tasks, even these results cannot be fully integrated.

To address this problem, we analysed brain-wide data from the International Brain Laboratory—electrophysiological recordings from 242 brain regions and wide-field imaging (WFI) from layers 2/3 of cortex in mice performing the same decision-making task—all registered to the Allen Common Coordinate Framework[31,32]. Our results suggest that the prior is encoded cortically and subcortically, across all levels of the brain, including early sensory regions.

## Mice use the prior to optimize their performance

Mice were trained to discriminate whether a visual stimulus, of different contrasts, appeared in the right or left visual field (Fig. 1a). Importantly, the prior probability that the stimulus appeared on the right side switched in a random and uncued manner between 0.2 and 0.8 in blocks of 20–100 trials (Fig. 1b). Knowledge of the current prior would help the mice to perform well; in particular, the prior is the only source of information on zero contrast trials, as the probability of reward on these trials is determined by the block probability. We refer to the experimentally determined prior as the 'true block prior'. As the presence of the blocks was not explicitly cued, mice could form only a subjective estimate of the true block prior from the trial history. At best, they could compute the estimate of the true block prior given full knowledge of the task structure and the sequence of previous stimulus sides since the start of a session. Hereafter, we refer to this as the Bayes-optimal prior (Methods and Figs. 1b and 2a).

Analysing choice behaviour revealed that mice used the block structure to improve their performance. Psychometric curves conditioned on right and left blocks, averaged across all animals and all sessions, were displaced relative to each other, in a direction consistent with the true block prior (two-tailed signed-rank Wilcoxon paired test comparing the proportion of right choices on zero-contrast trials: $t = 15$, $P = 2.0 \times 10^{-24}$, $n = 139$ mice; Fig. 1c). The shift was most pronounced at zero contrast and nearly disappeared at signed contrasts of −1 and 1 (Fig. 1c (inset)), suggesting that it stemmed from a prior-based mechanism rather than an action bias (for example, a perseverative motor

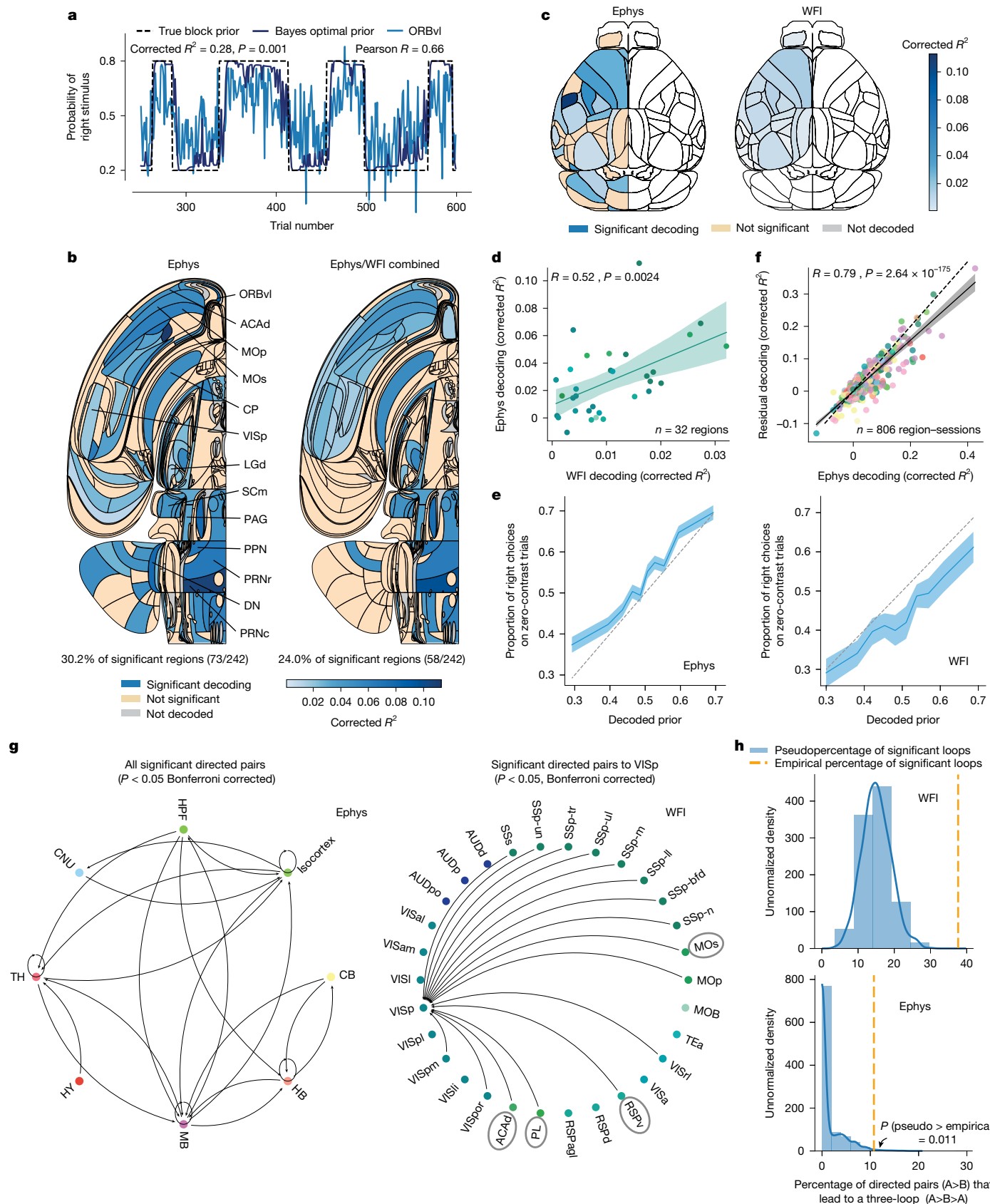

**Fig. 2 | See next page for caption.**

bias), which would have produced the same shift at all contrasts. As a result, the mice performed at 58.7 ± 0.4% (mean ± s.e.m.) correct for zero-contrast trials. This is statistically significantly better than

chance (two-tailed signed-rank Wilcoxon $t = 89$, $P = 1.5 \times 10^{-23}$, $n = 139$ mice), albeit significantly worse than an observer that generates actions by sampling from the Bayes-optimal prior, which would perform at

**Fig. 2 | Prior decoding during the ITI. a**, The Bayes-optimal prior versus the prior decoded from the ORBvl in one session. **b**, Swanson maps of cross-validated corrected $R^2$ for significant areas (Methods) Left, the Ephys map. Right, Ephys and WFI combined. A region is significant if the Fisher combined $P < 0.05$ on the left map and passes Benjamini–Hochberg correction (1% false-discovery rate) on the right. DN, dentate nucleus; MOp, primary motor area; PAG, periaqueductal gray; PPN, pedunculopontine nucleus; PRNc, pontine reticular nucleus caudal part; PRNr, pontine reticular nucleus. A full list of region names and their abbreviations is available online (https://github.com/int-brain-lab/paper-brain-wide-map/blob/plotting/brainwidemap/meta/region_info.csv). **c**, Ephys versus WFI results for the dorsal cortex. All areas significantly encode the Bayes-optimal prior in the WFI data (Fisher combined $P < 0.05$). Blue, significant; orange, not significant; grey, not decoded because we lack quality-controlled data (Methods); white, not decoded due to a lack of recordings or because it was out of the scope of analysis (although both hemispheres were recorded in WFI, only the left is decoded here to match Ephys). **d**, The corrected $R^2$ for Ephys and WFI are significantly correlated (the colour scheme is shown in Extended Data

Fig. 5b; shading represents the 95% confidence intervals). **e**, The proportion of right choices on zero-contrast trials versus cross-validated decoded Bayes-optimal prior from neural activity: higher decoded priors are associated with more right choices (Methods; the shading shows the s.e.m.). **f**, The corrected $R^2$ for decoding the prior from neural activity correlates with the corrected $R^2$ for decoding the residual prior (prior minus prior decoded from DLC), indicating that the prior decoded from neural activity is not explained by DLC motor features (the colour scheme is provided in Extended Data Fig. 5a; shading shows the 95% confidence intervals). **g**, Granger graph at the Cosmos level (Methods and Extended Data Fig. 5) in Ephys showing the bidirectional flow of prior information between the subcortical and cortical regions (right). Left, directed pairs targeting the VISp in WFI data reveal significant feedback from higher-order areas (grey circles) to early sensory regions. **h**, The proportion of significant directed pairs forming loops of size 3 (orange dashed line) in the WFI (top) and Ephys (bottom) data. The flow of prior information forms more loops than expected by chance (blue null distribution).

---

61.1 ± 1.8% (mean ± s.d.; two-tailed signed-rank Wilcoxon paired test, $t = 2,171$, $P = 1.5 \times 10^{-8}$, $n = 139$ mice).

Tracking performance around block switches provided further evidence that the animals estimated and used the prior. Indeed, around block switches, the performance dropped, presumably due to the mismatch between the subjective and true block prior. Thereafter, the performance on zero-contrast trials recovered with a decay constant of 4.97 trials (jackknife median; Methods). This is slower than an observer that generates actions by sampling the Bayes-optimal prior (jackknife median: 2.43 trials, two-tailed paired $t$-test, $t = 3.35$, $P = 0.001$, $n = 139$ jackknife replicates; Extended Data Fig. 1).

## Decoding the prior during the ITI

To determine where the prior is encoded in the mouse's brain, we used linear regression to decode the Bayes-optimal prior from neural activity during the intertrial interval (ITI) when wheel movements are minimized[33] (from −600 ms to −100 ms before stimulus onset; Methods). Note that decoding the Bayes-optimal prior is more sensible than decoding the true block prior, as mice are not explicitly cued about block identity and therefore cannot possibly know this latter quantity. We assess the quality of the decoding with an $R^2$ measure. However, to assess the statistical significance of this value, we cannot use standard linear regression methods, as these assume independence of trials, while both neural activity (for example, from slow drift in the recordings stemming from movements of the probes across trials) and the prior exhibit temporal correlations. Instead, we use a pseudosession method[34]: we first construct a null distribution by decoding the (counterfactual) Bayes-optimal priors computed from stimulus sequences generated by sampling from the same process as that used to generate the stimulus sequence that was actually shown to the mouse (Methods). A session was deemed to encode the prior significantly if $R^2$ computed for the actual stimuli is larger than the 95th percentile of the null distribution generated from pseudosessions; effect sizes are reported as a corrected $R^2$, the difference between the actual $R^2$ and the median $R^2$ of the null distribution. All values of $R^2$ reported in this paper are corrected $R^2$ unless specified otherwise.

For completeness, we decoded the Bayes-optimal prior, $\hat{p}$, its log odds ratio ($\log(\hat{p}/(1-\hat{p}))$) (to test whether neural activity is linearly related to log probabilities as assumed by the theory of probabilistic population codes[8]), the true block prior (Fig. 1b) and the Bayes-optimal prior on a narrower decoding time window (from −400 ms to −100 ms). For the Bayes-optimal prior, the analysis of the electrophysiological data (Ephys) revealed that around 30.2% of brain areas (73 out of 242 regions), spanning the forebrain, midbrain and hindbrain, encoded the prior significantly ($P < 0.05$, pseudosession test; Fisher's method to combine $P$ values from multiple recordings of one region, no multiple-comparison

correction; sagittal slices are shown in Fig. 2b and Extended Data Fig. 3). For example, we could decode the Bayes-optimal prior from a population of 160 neurons in the ventrolateral orbitofrontal cortex (ORBvl) with an accuracy of $R^2 = 0.28$ (Fig. 2a, $P = 0.001$, uncorrected $R^2 = 0.35$). Regions with significant prior encoding include associative cortical areas like the ORBvl and the dorsal anterior cingulate area (ACAd), as well as early sensory areas such as the primary visual cortex (VISp) and the lateral geniculate nucleus (LGd). The Bayes-optimal prior could also be decoded from cortical and subcortical motor areas, such as primary and secondary motor cortex, the intermediate layer of the superior colliculus (SCm), the gigantocellular reticular nucleus and the pontine reticular nucleus, even though we decoded activity during the ITI, when wheel movements were minimal (Extended Data Fig. 2). The encoding of the Bayes-optimal prior is also visible in the peristimulus time histogram of single neurons (Extended Data Fig. 4). Decoding the log odds ratio of the Bayes-optimal prior, as opposed to the linear version, revealed consistent findings, with 38.0% (92 out of 242 regions) of regions encoding it significantly across all brain processing levels (Extended Data Fig. 6). When decoded from a narrower time window (−400 ms to −100 ms), the Bayes-optimal prior was still significantly decoded across all brain processing levels, albeit with a reduced overall decodability (25.6% of regions, 62 out of 242 regions; Extended Data Fig. 6). An even smaller percentage of regions (19.4%, 47 out of 242 regions; Extended Data Fig. 6) was found to encode the prior significantly when decoding the true block prior, suggesting that the animal's subjective prior aligns more closely with the Bayes-optimal prior than with the true block prior. This observation is supported by a behavioural analysis, which revealed that a model using the true block as a prior was less effective at explaining behaviour compared with the Bayes-optimal model (Extended Data Fig. 6d). An analysis to determine the necessary number of recordings per region indicated that around ten recordings per region are required to reach the obtained significance levels (Extended Data Fig. 5e). Given that the median number of sessions per region in Ephys is 6 (Extended Data Fig. 5c,d), it is likely that the reported levels of significance are underestimated.

The analysis of WFI data suggests an even more widespread encoding of the prior in cortical regions. Indeed, the Bayes-optimal prior was found to be significantly reflected in all dorsal cortical regions (Fig. 2c). This result may reflect a better signal-to-noise ratio, but it might also be due to the calcium signal from axons arising outside these specific areas. However, we also found that the corrected region-specific $R^2$ values for the WFI and Ephys modalities were significantly correlated (Spearman correlation, $R = 0.52$, $P = 0.0024$, $n = 32$ regions; Fig. 2d). Interpreting the effect size in both Ephys and WFI modalities is challenging due to confounding factors such as the number of sessions and units in Ephys, and the number of pixels in WFI (Extended Data Fig. 7c,d). To control for correlations between these confounds across modalities (Extended Data Fig. 7e), we corrected the widefield effect

size for region sizes. Despite this correction, the correlation between effect sizes across modalities remained significant and even strengthened (Extended Data Fig. 7f), therefore suggesting that the effect sizes that we decode are at least partly specific to the decoded regions.

A quarter of the regions (24.0%, 58 out of 242), still at all levels of brain processing, were found to be significant when merging this larger Ephys dataset and WFI data into a single map (using Fisher's method to combine $P$ values across Ephys and WFI (Methods) and applying Benjamini–Hochberg correction for multiple comparisons with a conservative false-discovery rate of 1%; sagittal slices are shown in Fig. 2b and Extended Data Fig. 3).

If the decoded prior is truly related to the subjective prior inferred and used by the animal, the amplitude of the decoded prior should be correlated with the animals' performance on zero-contrast trials. Figure 2e shows that this is indeed the case for both the Ephys and WFI data: on zero-contrast trials, the probability that the mice chose the right side was proportional to the cross-validated decoded Bayes-optimal prior of the stimulus appearing on the right (Methods). Importantly, this relationship remained significant even after controlling for possible drift in the recordings (Extended Data Fig. 8a) and was sensitive to contrast strength (Extended Data Fig. 8b,c): consistent with Fig. 1c, this relationship was strongest at zero contrast and nearly vanished at the highest contrasts. Further analysis at the regional level (Extended Data Fig. 8d) shows a significant relationship in 17.8% of Ephys regions and 90.1% of WFI regions across all hierarchical levels: LGd, SCm, caudoputamen (CP), medial secondary motor cortex (MOs) and ACAd. Moreover, regions that more strongly reflect the prior were more predictive of the animal's decisions, suggesting the behavioural relevance of the decoded prior (Extended Data Fig. 8e).

Our results indicate that the Bayes-optimal prior was encoded in multiple areas throughout the brain. However, it is conceivable that mice adjusted their body posture or movement according to the subjective prior and that neural activity in some areas simply reflected these body adjustments. We call this an embodied prior. To test for this possibility, we analysed video recordings using Deep Lab Cut (DLC)[35,36] to estimate the position of multiple body parts, whisking motion energy and licking during the ITI (Methods). We then trained a decoder of the Bayes-optimal prior from these features, and found significant decoding in 38.0% (65 out of 171) of sessions. For these sessions, we found that the $R^2$ for the prior decoded from video features was correlated with the $R^2$ for the prior decoded from neural activity (at the brain region level), therefore suggesting that the prior signal might be an embodied prior related to body posture (Pearson correlation, $R = 0.18$, $P = 1.6 \times 10^{-7}$, $n = 806$ region sessions; Extended Data Fig. 9a). To test for this possibility further, we decoded the prior residual, defined as the Bayes-optimal prior minus the Bayes-optimal prior estimated from video features, from neural activity (again, at the session-region level). If the neural prior simply reflects the embodiment of features extracted by DLC, we should not be able to decode the prior residual from the neural activity and the $R^2$ of the prior residual should not be correlated with the $R^2$ of the full prior decoded from neural activity. Crucially, this is not what we observed. Instead, these two values of $R^2$ are strongly correlated (Fig. 2f; Pearson correlation, $R = 0.89$, $P = 8 \times 10^{-279}$, $n = 806$ region sessions), therefore suggesting that the neural prior is not an embodied prior or, at least, that it cannot be fully explained by the motor features extracted from the video.

To enhance the robustness of our analysis further, we repeated the embodiment study, this time also including eye position data (on sessions on which these were available). This additional step demonstrated that the neural prior could not be entirely attributed to a combination of both motor features and eye position (the feature importance is shown in Extended Data Fig. 9b,c). We also specifically checked whether changes in eye position across blocks could account for the significant results in early visual areas such as VISp or LGd. It is indeed conceivable that mice look in the direction of the expected stimulus before a trial. If so, what we interpret as a prior signal in these early sensory

areas might simply be due to a signal related to eye position. Consistent with this possibility, we found a significant correlation (Pearson correlation $R = 0.36$, $P = 0.0163$, $n = 44$ region sessions) between the neural decoding $R^2$ and the eye position decoding $R^2$, that is, the $R^2$ for decoding the Bayes-optimal prior from eye position (using sessions in which video was available and recordings were performed in the VISp and LGd; $n = 44$ region sessions; Extended Data Fig. 9d). Following the same approach as for the body posture and motion features, we next decoded the prior residual (Bayes-optimal prior minus Bayes-optimal prior estimated from eye position) from neural activity and found that the residual decoding $R^2$ was correlated with the neural decoding $R^2$ (Pearson correlation, $R = 0.8$, $P = 7 \times 10^{-11}$, $n = 44$ region sessions; Extended Data Fig. 9d). Thus, the prior signals found in the VISp and LGd did not simply reflect subtle changes in eye position across blocks.

Our decoding analysis reveals a robust, distributed representation of the Bayes-optimal prior throughout the brain, suggesting a complex network of information flow. To investigate the dynamics of the prior information network, we conducted a Granger causality analysis during the ITI, between the time series of the decoded prior from one brain region and that of another (Methods and Extended Data Fig. 10). This analysis revealed several key findings: (1) the flow of prior information between brain areas is significantly greater than expected by chance (Extended Data Fig. 10a); (2) this prior flow includes comprehensive communications across the entire brain, from subcortical to cortical areas and vice versa (Fig. 2g, left); (3) it includes significant feedback connections from higher-order areas to early sensory areas (Fig. 2g, right); and (4) there is a higher prevalence of loops within this communication network than would be anticipated by chance (Fig. 2h), including between higher-order and early sensory areas (Extended Data Fig. 10e). These results collectively highlight a loopy and intricate interarea communication of prior information within the brain.

## Post-stimulus prior

We also decoded the Bayes-optimal prior during the 100 ms interval after stimulus onset and found similarities between the encoding of the prior before and after stimulus onset. To avoid confounding the prior with the stimulus identity, two variables that are highly correlated (Spearman correlation, $R = 0.40$, $P < 1 \times 10^{-308}$), we first trained a linear decoder of signed contrast from neural activity in each region. We used the output of this decoder to fit two neurometric curves (the proportion of decoded right stimulus as a function of contrast; Methods) conditioned on the Bayes-optimal prior being above 0.7 or below 0.3. We next computed the vertical displacement of the fitted neurometric curves for zero contrast. If an area encodes the prior beyond the stimulus, we expect a shift between these two curves (an example is shown in Fig. 3a). The null distribution was generated using the pseudosession method previously described[34]. Note that the same analysis can be performed during the ITI, although, in this case, the neurometric curves are expected to be flat (Fig. 3b), which is indeed what we observed (Extended Data Fig. 11a). This approach enables us to separate the encoding of the prior from the encoding of the stimulus; however, it is possible that some of our results are related to the emergence of the animal's choices as the animals can respond in less than 100 ms on some trials[33].

Using this approach, we found that we can detect the prior significantly from 17.8% (43 out of 242) and 84.4% (27 out of 32) of areas during the post-stimulus period for Ephys (Extended Data Fig. 11b) and WFI data (Fig. 3e), respectively. When applying this methodology to the ITI, we found smaller percentages than when using direct decoding, in part because this neurometric shift measure is less sensitive (in the ITI, only 15.7% of regions for Ephys and 93.8% for WFI are significant for the Bayes-optimal prior when using the neurometric shift on Ephys/WFI data, versus 30.2% and 100%, respectively, for conventional decoding). As was the case during the ITI, we found that the Ephys and WFI post-stimulus shifts were correlated (Spearman correlation,

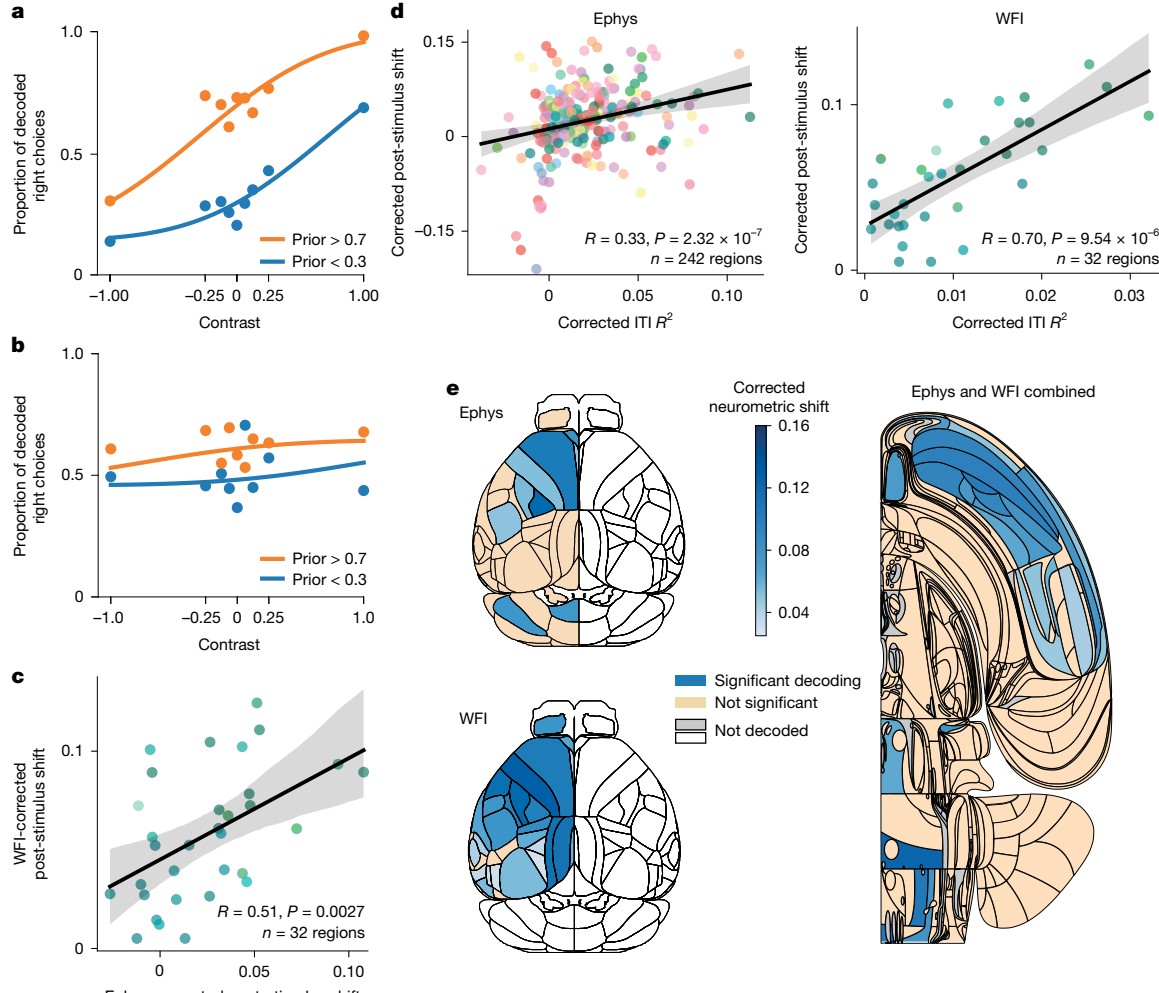

**Fig. 3 | Encoding of the prior across the brain during the post-stimulus period. a**, Example of neurometric curves for the post-stimulus period from an SCm recording. **b**, The same as in **a**, but for the ITI period. **c**, The correlation between shifts from Ephys and WFI data. Each dot corresponds to one cortical region (two-sided Spearman correlation; the shaded area indicates the 95% confidence interval). **d**, The post-stimulus shifts are correlated with the ITI $R^2$ for both Ephys and WFI data (two-sided Spearman correlation; the colour scheme is shown in Extended Data Fig. 5a; the shading shows the 95% confidence interval). **e**, Comparison between the corrected post-stimulus neurometric shift in the Ephys and WFI data for the dorsal cortex (left). A region is deemed to

be significant if its Fisher combined $P$ value is below 0.05 (Methods). Right, Swanson map of the corrected $R^2$ averaged across Ephys and WFI data for areas that have been deemed to be significant given both datasets (using Fisher's method for combining $P$ values), and after applying the Benjamini–Hochberg correction for multiple comparisons. Blue, significant; orange, not significant; grey, not decoded because we lack quality-controlled data (Methods); white, not decoded because of insufficient recordings or because it was out of the scope of analysis (although both hemispheres were recorded in WFI, only the left is decoded here to match Ephys).

$R = 0.51$, $P = 0.0027$, $n = 32$; Fig. 3c). Moreover, the post-stimulus neurometric shift is correlated with the $R^2$ obtained in the same areas during the ITI period (Spearman correlation, $R = 0.33$, $P = 2.32 \times 10^{-7}$, $n = 242$ (Ephys); $R = 0.70$, $P = 9.54 \times 10^{-6}$, $n = 32$ (WFI); Fig. 3d). In other words, areas encoding the prior in the ITI also tend to do so during the post-stimulus period. This was confirmed by comparing the shifts during the post-stimulus and ITI periods, which were also found to be correlated (Extended Data Fig. 11c,d).

We obtained similar results when merging the Ephys and WFI data into a single map (using Fisher's method to combine $P$ values across Ephys and WFI) and applying Benjamini–Hochberg correction for multiple comparisons with a false-discovery rate of 1% (11.2% of significant regions, 27 out of 242; Fig. 3e). Importantly, as observed during the ITI, areas encoding the prior were found at all levels of brain processing.

Moreover, we examined whether regions encoding the stimulus also encoded the prior, as would be expected if these regions are involved in inferring the posterior distribution over the stimulus side. We found

that the corrected $R^2$ for the stimulus decoding was indeed correlated with the corrected $R^2$ for the Bayes-optimal prior decoding (Spearman correlation, $R = 0.29$, $P = 2.4 \times 10^{-5}$, $n = 201$ regions from BWM analysis[32]; Extended Data Fig. 12a). Moreover, among the 40 areas that were found to encode the stimulus significantly, 25 also encoded the prior significantly, including, once again, areas at all levels of brain processing (for example, the LGd, VISp, SCm, CP, MOs and ACAd; Extended Data Fig. 12b).

## Decoding the action kernel prior

So far, we have established that mice leveraged the block structure and that the Bayes-optimal prior can be decoded from the neural data at all levels of brain processing. However, it remains to be seen whether the mice truly compute the Bayes-optimal prior or, perhaps, use heuristics to compute a subjective, approximate, prior[37].

To address this, we developed several behavioural models and used session-level Bayesian cross-validation followed by Bayesian model selection[38] to identify the one that fits the best (Methods). This analysis

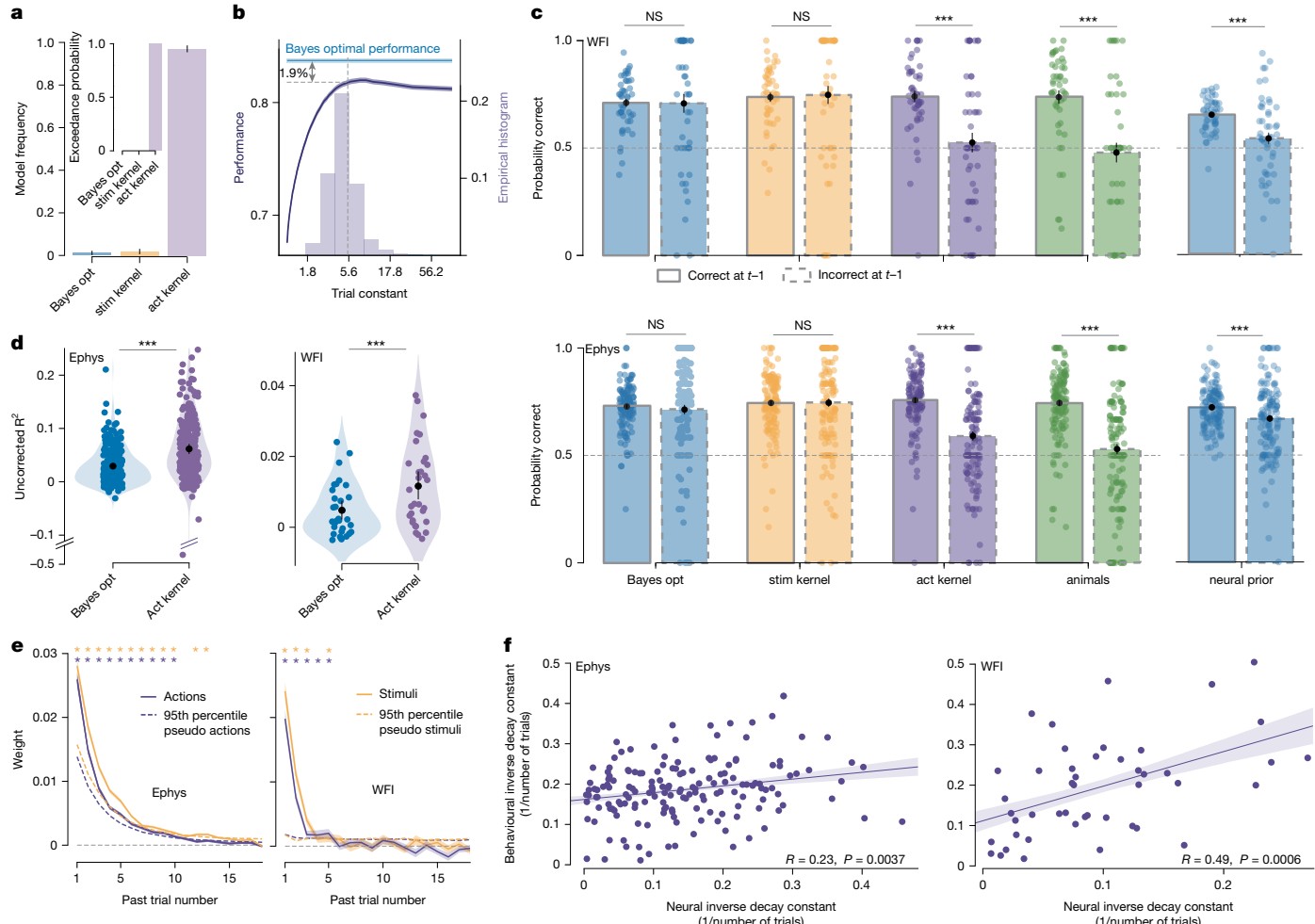

**Fig. 4 | Action kernel prior. a**, The model frequency and exceedance probabilities for three subjective prior models, using session-wise cross-validation in 107 mice (≥2 sessions; Methods; the error bars show the s.d.). The best model involves filtering recent actions with an exponential kernel. Act. kernel, action kernel; Bayes opt., Bayes optimal; Stim. kernel, stimulus kernel. **b**, The decay constant for the action kernel across sessions and animals (light purple; median = 5.45 trials, dashed line). The proportion correct for the action kernel as a function of the decay constant is shown (dark purple). The median trial constant aligns with the optimal performance achievable with the action kernel, only 1.9% below the Bayes-optimal performance. **c**, Performance on zero-contrast trials conditioned on whether the previous action was correct or incorrect, across behavioural models and animal behaviour. Right, the same analysis for a simulated agent using the Bayes-optimal prior decoded from neural data (neural prior) to generate decisions. The performance drop between correct and incorrect previous trials for the neural prior suggests that the action

kernel model better accounts for neural activity, consistent with behaviour. Top, WFI data; $n = 51$ sessions. Bottom, Ephys data; $n = 139$ mice. Statistical analysis was performed using Wilcoxon signed-rank tests. The error bars show the s.e.m. **d**, The uncorrected $R^2$ is higher when decoding the action kernel prior compared with when decoding the Bayes-optimal prior during the ITI, for the Ephys and WFI modalities. Statistical analysis was performed using Wilcoxon signed-rank tests. **e**, The weight of the previous actions (purple) and previous stimuli (yellow) on the decoded Bayes-optimal prior, estimated from neural activity (left, Ephys; right, WFI). The dashed lines show the 95th percentile of the null distribution (Methods). **f**, The correlation between neural inverse decay constants (estimating the temporal dependency of the neural signals on previous actions) and behavioural inverse decay constants (from fitting the action kernel to behaviour). Both Ephys and WFI data show correlations (two-sided Pearson test; Methods; the shading shows the 95% confidence intervals). NS, not significant; \*$P < 0.05$, \*\*\*$P < 0.001$.

suggests that most mice on most sessions estimate what we will refer to as the action kernel prior, which is obtained by calculating an exponentially weighted average of recent past actions (Fig. 4a). The action kernel prior explains the choices of the mice better than the Bayes-optimal prior and better than models of behavioural strategies that calculate an exponentially weighted average of recent stimuli (the 'stimulus kernel'), or assume a one-step repetition bias or a multi-step repetition bias[39] or the presence of positivity and confirmation biases[40] (Extended Data Fig. 13a and Supplementary Information). Consistent with the action kernel model, mice updated their subjective prior on the first 90 unbiased trials, even though the true prior is set to 0.5 during that phase (Extended Data Fig. 13d). Moreover, mice relied on more than just zero-contrast trials to update their subjective prior (Extended Data Fig. 13b,c). The decay constant of the exponential action kernel had a

median of 5.45 trials across all mice (Fig. 4b, blue histogram), similar to the decay constant of recovery after block switches (4.97 trials; Fig. 1d and Extended Data Fig. 1). Notably, this is close to the value of the decay constant, which maximizes the percentage of correct responses, given this (suboptimal) form of prior, losing only 1.9% compared with the performance of the Bayes-optimal version (Fig. 4b). These curves are obtained by simulating the action kernel by varying the decay constant while keeping all other parameters at their best-fitting values.

If, as our behavioural analysis suggests, mice use the action kernel prior, then we should find that, when we decode the prior inferred from the action kernel, $R^2$ should be higher than when we decode the prior predicted by any other method. This is borne out by the data in both Ephys and WFI during the ITI (Fig. 4d; Wilcoxon signed-rank test, $t = 2,230, P = 2.6 \times 10^{-30}, n = 242$ regions (Ephys); and $t = 13, P = 4.1 \times 10^{-8}$,

$n$ = 32 regions (WFI)). However, unfortunately, and in contrast to the Bayes-optimal prior, we cannot determine which areas encode the action kernel prior significantly, owing to the impossibility of generating a null distribution, as this would formally require having access to the exact statistical model of the animal behaviour (see the 'Assessing the statistical significance of the decoding of the action kernel prior' section in the Methods).

To explore further whether neural activity better reflects the action kernel prior, as opposed to the stimulus kernel prior or the Bayes-optimal prior, we looked at changes in performance on zero-contrast trials after correct and incorrect actions. When considering behaviour within blocks, a decision-making agent using an action kernel prior should achieve a higher percentage of correct responses after a correct block-consistent action than an incorrect one because, on incorrect trials, it updates the prior with an action corresponding to the incorrect stimulus side. Models simulating agents using either the Bayes-optimal prior or the stimulus kernel prior do not show this asymmetry as they perform their updates using the true stimulus, which can always be correctly inferred from the combination of action and reward (see also ref. 37). Mouse behaviour showed the asymmetry in performance (Fig. 4c). To test whether the neural data shared this asymmetry, we decoded the Bayes-optimal prior from ITI neural activity and simulated the animal's decision on each trial by selecting a choice according to whether the decoded prior was greater or smaller than 0.5 (that is, assuming every trial had a zero-contrast stimulus). We then examined whether the resulting sequence of hypothetical choices would show the asymmetry. If so, this is a property of the neural data as the predicted quantity, the Bayes-optimal prior, does not show the asymmetry. As shown in Fig. 4c, the performance for both modalities, Ephys and WFI, was indeed higher after correct versus incorrect trials, therefore strengthening our hypothesis that neural activity more closely reflects the action kernel prior.

We next tested the sensitivity of the decoded Bayes-optimal prior, estimated from neural activity, to previous actions (decoding the Bayes-optimal prior instead of the action kernel prior to enable us to test for statistical significance; Methods). If the prior that we estimate from neural activity reflects the subjective prior estimated from behaviour, we should find that the neural prior is sensitive to the past 5 or 6 trials. Using an orthogonalization approach, we estimated the influence of past actions on the decoded Bayes-optimal prior and found that this influence extends at least to the past five trials in both Ephys and WFI (Fig. 4e; see the 'Orthogonalization' section of the Methods). A similar result was obtained when testing the influence of the past stimuli (Fig. 4e). These numbers are consistent with the decay constant estimated from behaviour (5.45 trials). These results were obtained at the session level, by analysing all available neurons. Furthermore, we analysed single regions for which we had a large number of neurons recorded simultaneously (SCm, CP and ventral posteromedial nucleus of the thalamus (VPM)) or strong imaging signals (primary motor area, VISp, MOs). In all cases, we found that an asymmetry in the neural data after correct and incorrect choices as well as evidence that the Bayes-optimal prior decoded from these regions is influenced by the past 5 or 6 actions (Extended Data Fig. 14).

These analyses also address one potential concern with our decoding approach. It is well known in the literature that animals keep track of the last action or last stimulus[41–43]. It is therefore conceivable that our ability to decode the prior from neural activity is simply based on the encoding of the last action in neural circuits, which indeed provides an approximate estimate of the Bayes-optimal prior as actions are influenced by the prior (Fig. 1c). The fact that we observe an influence of the last 5 or 6 trials, and not just the last trial, rules out this possibility.

To test this even further, we estimated the temporal dependency of the WFI single-pixel and Ephys single-unit activities on past actions directly and compared them to the behavioural sensitivity to past actions on the same sessions (both expressed in terms of neural learning rates, that is, the inverse of the decay constants; Methods). Note that this analysis tests whether the temporal dynamics of neural activity is similar to the temporal dynamics of the mouse behaviour, defined by fitting the action kernel model, but without regressing first the neural activity against any prior. We found that the inverse decay constants of the neural activity are indeed correlated across sessions with the inverse decay constants obtained by fitting the action kernel model to behaviour (Fig. 4f). Critically, this correlation goes away if we perform the same analysis using stimulus kernels instead of action kernels (Extended Data Fig. 15a). Moreover, these results established at a session level remained when accounting for the variability across mice (Extended Data Fig. 15b).

We next examined the link between behavioural performance and specific brain regions by comparing their neural inverse decay constants with the behavioural inverse decay constants. Notably, associative areas like the secondary motor cortex and retrosplenial areas more closely mirrored these behavioural constants than the primary visual and motor cortex (Extended Data Fig. 15c). We also observed that the correlation between behavioural and neural decay constants reflected the prior-corrected $R^2$ from the same regions, indicating that regions with higher prior-decoding $R^2$ scores best align with the animal's cognitive strategies as measured by the action kernel lengths (Extended Data Fig. 15c). This analysis was not extended to electrophysiology recordings due to the limited number of available sessions per region (Extended Data Fig. 7a,b).

## Discussion

In summary, we report that mice bias their decisions nearly optimally according to their prior expectations. As we have seen, the subjective prior of the mice is based on previous actions, not previous stimuli—a result consistent with past studies in rodents[44] and primates[45]. Notably, this subjective prior is encoded, at least to some extent, at all levels of processing in the brain, including early sensory regions (for example, LGd and VISp), associative regions (ORBvl, ACAd and SCm) and motor regions (MOs, primary motor area and gigantocellular reticular nucleus). Moreover, a Granger analysis revealed the existence of reciprocal loops, communicating specifically the subjective prior between cortical and subcortical regions as well as between sensory and associative cortical areas. These findings lend further support to the hypothesis that information flows across the brain in a way that could support the sort of multidirectional inference apparent in Bayesian networks[17,18,20,22].

One might argue that what we call a 'subjective prior' might be better called 'motor preparation' in motor-related areas, or a top-down 'attentional signal' in early sensory areas. However, ultimately, what is important is not the term that we use to refer to this signal but, rather, that it has properties consistent with the subjective prior: (1) it is predictive of the animal's choices, particularly on zero-contrast trials (Fig. 2e); (2) it depends on previous choices (Fig. 4c); and (3) it reflects more than the last choice or last stimulus, but depends instead on the past 5 or 6 choices (Fig. 4e). As we have seen, the signals that we have recovered throughout the mouse brain fulfil all of these properties.

There are several proposals in the literature as to how probability distributions might be encoded in neural activity. These include linear probabilistic population codes[8], sampling based codes[6], other activity-based codes[7,37,46–48] and the synaptic weights of neural circuits[9]. We note that our results are compatible with two requirements of linear probabilistic population codes[8,49]: (1) the log odds of the Bayes-optimal prior is linearly decodable from neural activity (Extended Data Fig. 6); and (2) changes in the Bayes-optimal prior from trial to trial are reflected in the population activity[49].

If the likelihood is also encoded with a linear probabilistic population code, having the prior in the same format would greatly simplify the computation of the posterior distribution, as it would simply require a linear combination of the neural code for the prior and likelihood. As it turns out, it is likely that the likelihood indeed relies on a linear probabilistic population code. Indeed, the neural code for contrast, which is the variable that controls the uncertainty of the visual stimulus in our

experiment, has been shown to be compatible with linear probabilistic population code in primates[8].

Whether our results are also compatible with sampling-based codes is more difficult to assess, as there is still a debate as to which aspects of neural activity correspond to a sample of a probability distribution[6,24,50]. Moreover, the fact that our prior follows a Bernoulli distribution, which is particularly simple, makes it harder to tease apart the various probabilistic coding schemes.

Ultimately, determining the exact nature of the neural code for the prior will require developing a neural model of Bayesian inference in a large, modular, loopy network—a pressing, remaining task. A critical foundation for this development is the remainder of the extensive data in the International Brain Laboratory brain-wide map (described in the companion paper[32]). This provides a picture, at a considerable scale, of the neural processes underpinning decision-making, in which the prior plays such a critical part.

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

**International Brain Laboratory**

**Charles Findling**[1], **Félix Hubert**[1], **Luigi Acerbi**[2], **Brandon Benson**[3], **Julius Benson**[4], **Daniel Birman**[5], **Niccolò Bonacchi**[6], **E. Kelly Buchanan**[3], **Sebastian Bruijns**[7], **Matteo Carandini**[8], **Joana A. Catarino**[9], **Gaelle A. Chapuis**[1], **Anne K. Churchland**[10], **Yang Dan**[11], **Felicia Davatolhagh**[10], **Eric E. J. DeWitt**[9], **Tatiana A. Engel**[12], **Michele Fabbri**[9], **Mayo A. Faulkner**[8], **Ila Rani Fiete**[13], **Laura Freitas-Silva**[9], **Berk Gerçek**[1], **Kenneth D. Harris**[8], **Michael Häusser**[8,14], **Sonja B. Hofer**[15], **Fei Hu**[11], **Julia M. Huntenburg**[7], **Anup Khanal**[10], **Chris Krasniak**[16], **Christopher Langdon**[17], **Christopher A. Langfield**[17], **Peter E. Latham**[18], **Petrina Y. P. Lau**[8], **Zach Mainen**[9], **Guido T. Meijer**[9], **Nathaniel J. Miska**[15], **Thomas D. Mrsic-Flogel**[15], **Jean-Paul Noel**[19], **Kai Nylund**[5], **Alejandro Pan-Vazquez**[12], **Liam Paninski**[17], **Jonathan Pillow**[12], **Cyrille Rossant**[8], **Noam Roth**[5], **Rylan Schaeffer**[13], **Michael Schartner**[9], **Yanliang Shi**[12], **Karolina Z. Socha**[8], **Nicholas A. Steinmetz**[5], **Karel Svoboda**[20], **Charline Tessereau**[7], **Anne E. Urai**[21], **Miles J. Wells**[8], **Steven Jon West**[15], **Matthew R. Whiteway**[17], **Olivier Winter**[9], **Ilana B. Witten**[12], **Anthony Zador**[16], **Yizi Zhang**[17], **Peter Dayan**[7] & **Alexandre Pouget**[1]

## Methods

For the Ephys data, we used the 2024 IBL public data release[51], which is organized and shared using a modular architecture described previously[52]. It comprises 699 recordings from Neuropixels 1.0 probes. One or two probe insertions were realized over 459 sessions of the task, performed by a total of 139 mice. For a detailed account of the surgical methods for the headbar implants, see appendix 1 of ref. 33. For a detailed list of experimental materials and installation instructions, see appendix 1 of ref. 35. For a detailed protocol on animal training, see the methods of refs. 33,35. For details on the craniotomy surgery, see appendix 3 of ref. 35. For full details on the probe tracking and alignment procedure, see appendices 5 and 6 of ref. 35. The spike sorting pipeline used by IBL is described in detail in ref. 35. For the WFI data, we used another dataset consisting of 52 recordings of the dorsal cortex, realized over 52 sessions of the task, performed by a total of 6 mice. A detailed account of the WFI data acquisition and preprocessing was reported previously[53].

All experimental procedures involving animals were conducted in accordance with local laws and approved by the relevant institutional ethics committees. Approvals were granted by the Animal Welfare Ethical Review Body of University College London, under licences P1DB285D8, PCC4A4ECE and PD867676F, issued by the UK Home Office. Experiments conducted at Princeton University were approved under licence 1876-20 by the Institutional Animal Care and Use Committee (IACUC). At Cold Spring Harbor Laboratory, approvals were granted under licences 1411117 and 19.5 by the IACUC. The University of California at Los Angeles granted approval through IACUC licence 2020-121-TR-001. Additional approvals were obtained from the University Animal Welfare Committee of New York University (licence 18-1502); the IACUC at the University of Washington (licence 4461-01); the IACUC at the University of California, Berkeley (licence AUP-2016-06-8860-1); and the Portuguese Veterinary General Board (DGAV) for experiments conducted at the Champalimaud Foundation (licence 0421/0000/0000/2019).

### Mice

Animals were housed under a 12 h–12 h light–dark cycle—either normal or inverted, depending on the laboratory—and had unrestricted access to food and water except on training days. Depending on the laboratory, electrophysiology recordings and behavioural training took place during either the light or dark phase of the cycle. In total, data were collected from 139 adult C57BL/6 mice (94 male, 45 female), purchased from Jackson Laboratory or Charles River. On the day of electrophysiological recording, these mice ranged in age from 13 to 178 weeks (mean, 44.96 weeks; median, 27.0 weeks) and weighed between 16.1 and 35.7 g (mean 23.9 g, median 23.84 g).

### Inclusion criteria for the analysis

**Criteria for trial inclusion.** All trials were included except when the animals did not respond to the stimulus (no movement, no response) or when the first wheel movement time (reaction time) was shorter than 80 ms or longer than 2 s.

**Criteria for session inclusion.** All sessions were included except sessions with fewer than 250 trials (counting only included trials). In total, 41 sessions in Ephys and 1 session in WFI did not meet the criteria for the minimum number of trials.

**Criteria for neural recording inclusion.** An insertion was included in the analysis if it had been resolved, that is, if histology clearly revealed the path of the probe throughout the brain, as defined previously[35]. A neuron, identified during the spike sorting process, was included if it passed three quality control (QC) criteria (amplitude > 50 μV; noise cut-off < 20; refractory period violation). A region recorded along one or two probes was included in the analysis if there were at least five units across the session's probes that passed the QC. For WFI, we used all the image pixels and included a region recorded during a session in the analysis if there were at least five recorded pixels.

For the region-level analysis, after applying these criteria, we were left with 414 sessions for the Ephys dataset. Initially, we considered 418 (459 − 41) sessions that had more than 250 included trials, but 4 of these did not have any recorded regions meeting the minimum number of units required. Our region-level analysis spans 242 brain regions, defined by the Allen Common Coordinate Framework[31], recorded by at least one included insertion. Our Ephys region-level analysis spans 2,289 region–sessions, which are aggregated across sessions to give results at the region level. For the WFI dataset, we were left with 51 (52 − 1) sessions that had more than 250 included trials. They all had at least one recorded region meeting the minimum number of recorded pixels required. The imaging spans 32 regions of the dorsal cortex—which are included among the 242 regions decoded in the Ephys analysis, for a total of 1,539 ((51 × 32) − 28 − 65) region–sessions. This total accounts for the fact that not every region was visible in all sessions, summing to 28 non-observed region–sessions. Moreover, 65 region–sessions were excluded because the regions recorded had fewer than 5 pixels.

For the session-level analysis, neurons along the probes were used and most of the sessions in Ephys (457 out of 459) had at least 5 recorded units that passed the QC. Taking into account the session inclusion criteria, session-level analysis was performed on 416 sessions. All of the 51 WFI sessions passed the minimal number of trials criteria and were therefore included in the analysis.

**Criteria for the embodiment analysis.** Only sessions with available DLC features could be used for the embodiment prior analysis, which requires access to body position. For the Ephys dataset, we analysed the 171 sessions (out of 459) for which the DLC features met the quality criteria defined previously[35], and for which the other inclusion criteria were met. This resulted in a total of 806 region–sessions. WFI sessions were excluded from this analysis as no video recordings were available.

**Criteria for the eye position analysis.** Reliable tracking of eye position from video recordings was not possible for some sessions due to video quality issues. Thus, we recovered reliable eye position signals from 44 out of the 53 of sessions in which we had recorded from either VISp or LGd, the two regions for which we specifically analysed the impact of eye position.

**Joint decoding of DLC features and eye position signals.** We performed the joint decoding of DLC features and eye position signals on the 124 sessions in which the DLC features met the QC criteria and also in which the eye position signals were reliable, for a total of 660 region–sessions.

**Difference compared with the Brain Wide Map inclusion criteria.** There were two key differences between our inclusion criteria and those used in the Brainwide Map[32]. First, the Brainwide Map included only regions that had at least two recording sessions, whereas we included regions irrespective of the number of recording sessions. Second, we excluded sessions that had fewer than 250 trials after applying trial inclusion criteria, a criterion not applied in the Brainwide Map.

### Electrophysiology data

Spike counts were obtained by summing the spikes across the decoding window for each neuron and included trial. If there were $U$ units and $T$ trials, this binning procedure resulted in a matrix of size $U \times T$. For the ITI, the time window for the main decoding was (−600 ms,−100 ms) relative to stimulus onset and, for the post-stimulus window, it was (0 ms, +100 ms) relative to stimulus onset. We used L1-regularized

linear regression to decode the Bayes-optimal prior from the binned spike count data using the scikit-learn function sklearn.linear_model. Lasso (using one regularization parameter). We used L1 for Ephys because it is more robust to outliers, which are more likely to occur in single-cell recordings, notably because of drift. The Bayes-optimal prior was inferred from the sequence of stimuli for each session (see the 'Behavioural models' section and the Supplementary Information). This decoding procedure yielded a continuous-valued vector of length $T$.

### WFI data

For the WFI data, we used L2-regularized regression as implemented by the scikit-learn function sklearn.linear_model.Ridge (one regularization parameter). We used L2 regularization instead of L1 for WFI data because L2 tends to be more robust to collinear features, which is the case across WFI pixels. We decoded the activity from the vector of the region's pixels for a specific frame of the data. The activity is the change in fluorescence intensity relative to the resting fluorescence intensity $\Delta F/F$. Data were acquired at 15 Hz. Frame 0 corresponds to the frame containing stimulus onset. For the ITI, we used frame −2 relative to the stimulus onset. This frame corresponds to a time window of which the start ranges from −198 to −132 ms before the stimulus onset, and of which the end ranges from −132 to −66 ms, depending on the timing of the last frame before stimulus onset. For the post-stimulus interval, we used frame +2, which corresponds to a time window of which the start ranges from 66 to 132 ms from stimulus onset, and extends to 132 to 198 ms after onset. This interval is dependent on the timing of the first frame after stimulus onset, which can occur at anytime between 0 and 66 ms after onset. If there are $P$ pixels and $T$ trials, this selection procedure results in a matrix of size $P \times T$.

### Reversal curves

To analyse mouse behaviour around block reversals, we plotted the reversal curves defined as the proportion of correct choices as a function of trials, aligned to a block change (Fig. 1d). These were obtained by computing one reversal curve per mouse (pooling over sessions) and then averaging and computing the s.e.m. across the mouse-level reversal curves. For comparison purposes, we also showed the reversal curves for the Bayes-optimal model with a probability-matching decision policy. We did not plot s.e.m. values but instead s.d. values in this case as there was no variability across agents to account for.

To assess the differences between the mouse behaviour and the agent that samples actions from the Bayes-optimal prior, we fitted the following parametric function to the reversal curves:

$$p(\text{correct at trial } t) = (B + (A - B) \times e^{-t/\tau}) \times (t \geq 0) + B \times (t < 0)$$

with $t = 0$ corresponding to the trial of the block reversal, $\tau$ the decay constant, $B$ the asymptotic performance and $A$ the drop in performance right after a block change.

We fitted this curve using only zero-contrast trials, between the 5 pre-reversal trials and the 20 post-reversal trials. We restricted our analysis to the zero-contrast trials to focus on trials in which mice could only rely on block information to decide. This implied that we only used a small fraction of the data. To be precise, across the 459 sessions, we had an average of 10 reversals per session, and the proportion of zero-contrast trials is 11.1%. Fitting only on the zero-contrast trials around reversals led us to use around 28 trials per session, which accounts for around 3% of the behavioural data—when excluding the first 90 unbiased trials, the average session consists of 555 trials.

To make up for this limited amount of data, we used a jackknifing procedure for fitting the parameters. The procedure involved iteratively leaving out one mouse and fitting the parameters on the $N − 1 = 138$ zero-contrast reversal curves of the held-in mice. The results of the jackknifing procedure are shown in Extended Data Fig. 1.

### Nested cross-validation procedure

Decoding was performed using cross-validated, maximum-likelihood regression. We used the scikit-learn Python package to perform the regression[53], and implemented a nested cross-validation procedure to fit the regularization coefficient.

The regularization coefficient was determined by two nested fivefold cross-validation schemes (outer and inner). We first described the procedure for the Ephys data. In the outer cross-validation scheme, each fold was based on a training/validation set comprising 80% of the trials and a test set of the remaining 20% (random interleaved trial selection). The training/validation set was itself split into five sub-folds (inner cross-validation) using an interleaved 80–20% partition. Cross-validated regression was performed on this 80% training/validation set using a range of regularization weights, chosen for each type of dataset so that the bounds of the hyperparameter range are not reached. For each modality, we searched a logarithmically spaced grid of ridge-regularization weights that was tuned to the dimensionality of the corresponding feature space: for Ephys, the grid was $C \in \{10^{-5}, 10^{-4}, 10^{-3}, 10^{-2}, 10^{-1}\}$; for WFI, $C \in \{10^{-5}, 10^{-4}, 10^{-3}, 10^{-2}\}$; for the set of DLC-extracted behavioural features, $C \in \{10^{-4}, 10^{-3}, 10^{-2}, 10^{-1}, 1, 10, 100\}$; for eye-position features, $C \in \{10^{-4}, 10^{-3}, 10^{-2}, 10^{-1}, 1, 10, 100, 1,000, 10,000\}$; for the combined DLC-features + eye-position model, the same broad grid $C \in \{10^{-4}, 10^{-3}, 10^{-2}, 10^{-1}, 1, 10, 100, 1,000, 10,000\}$ was used; for the Ephys-based neurometric decoder, $C \in \{10^{-5}, 10^{-4}, 10^{-3}, 10^{-2}, 10^{-1}, 1\}$; and for the widefield neurometric decoder, $C \in \{10^{-5}, 10^{-4}, 10^{-3}, 10^{-2}\}$.

The regularization weight selected with the inner cross-validation procedure on the training/validation set was then used to predict the target variable on the 20% of trials in the held-out test set. We repeated this procedure for each of the five 'outer' folds, each time holding out a different 20% of test trials such that, after the five repetitions, 100% of trials have a held-out decoding prediction. For WFI, the procedure was very similar but we increased the number of outer folds to 50 and performed a leave-one-trial-out procedure for the inner cross-validation (using the efficient RidgeCV native sklearn function). We did this because the number of features in WFI (number of pixels) is much larger than in Ephys (number of units): around 167 units on average in Ephys when decoding on a session-level from both probes after applying all quality criteria, versus around 2,030 pixels on a session-level in WFI when decoding from the whole brain.

Furthermore, to average out the randomness in the outer randomization, we ran this procedure ten times. Each run used a different random seed for selecting the interleaved train/validation/test splits. We then reported the median decoding score $R^2$ across all runs. Regarding the decoded prior, we took the average of the predicted priors (estimated on the held-out test sets) across the ten runs.

### Assessing statistical significance

Decoding a slow varying signal such as the Bayes-optimal prior from neural activity can easily lead to false-positive results even when properly cross-validated. For example, slow drift in the recordings can lead to spurious, yet significant, decoding of the prior if the drift is partially correlated with the block structure[34,54]. To control for this problem, we generated a null distribution of $R^2$ values and determined significance with respect to that null distribution. This pseudosession method is described in detail previously[34].

We denote $X \in \mathbb{R}^{N \times U}$ the aggregated neural activity for a session and $Y \in \mathbb{R}^N$ the Bayes-optimal prior. Here, $N$ is the number of trials and $U$ the number of units. We generated the null distribution from pseudosessions, that is, sessions in which the true block and stimuli were resampled from the same generative process as the one used for the mice. This ensures that the time series of trials in each pseudosession shares the same summary statistics as the ones used in the experiment. For each true session, we generated $M = 1,000$ pseudosessions, and

used their resampled stimulus sequences to compute Bayes-optimal priors $Y_i \in \mathbb{R}^N$, with $i \in [1, M]$ the pseudosession number. We generated pseudoscores $R_i^2 \in R$, $i \in [1, M]$ by running the neural analysis on the pair $(X, Y_i)$. The neural activity $X$ is independent of $Y_i$ as the mouse did not see $Y_i$ but $Y$. Any predictive power from $X$ to $Y_i$ would arise from slow drift in $X$ unrelated to the task itself. These pseudoscores $R_i^2$ were compared to the actual score $R^2$ obtained from the neural analysis on $(X, Y)$ to assess statistical significance.

The actual $R^2$ is deemed to be significant if it is higher than the 95th percentile of the pseudoscores distribution $\{R_i^2, i \in [1, M]\}$. This test was used to reject the null hypothesis of no correlation between the Bayes optimal prior signal $Y$ and the decoder prediction. We defined the $P$ value of the decoding score as the quantile of $R^2$ relative to the null distribution $\{R_i^2, i \in [1, M]\}$.

For each region of the brain that we recorded, we obtained a list of decoding $P$ values, where a $P$ value corresponds to the decoding of the region's unit activity during one session. We used Fisher's method to combine the session-level $P$ values of a region into a single region-level $P$ value (see the 'Fisher's method' section for more details).

For effect sizes, we computed a corrected $R^2$, defined as the actual score $R^2$ minus the median of the pseudoscores distribution, $\{R_i^2, i \in [1, M]\}$. The corrected $R^2$ of a region is the mean of the corrected $R^2$ for the corresponding sessions.

### Choosing between Pearson and Spearman correlation methods

In our statistical analyses, we prioritized using Spearman's correlation when datasets included outliers, as it is robust against non-normal distributions. In other cases, we opted for Pearson's correlation to assess linear relationships. For paired comparisons, we used the Wilcoxon signed-rank test, which likewise makes no assumption of normality while retaining sensitivity to systematic shifts between conditions.

### Fisher's method

Fisher's method is a statistical technique used to combine independent $P$ values to assess the overall significance. It works by transforming each $P$ value into a $\chi^2$ statistic and summing these statistics. Specifically, for a set of $P$ values (one per session given a region), $p_1, p_2, p_3, \ldots,$ Fisher's method computes the test statistic

$$X^2 = -2 \sum_i \ln(p_i)$$

This statistic follows a $\chi^2$ distribution with $2 \times N_{\text{sessions}}$ d.f., $\chi^2_{2 \times N_{\text{sessions}}}$, under the null hypothesis that all individual tests are independent and their null hypotheses are true. If the computed test statistic $X^2$ exceeds a critical value from the $\chi^2$ distribution, the combined $P$ value $p(\chi^2_{2 \times N_{\text{sessions}}} \geq X^2)$ is considered significant and the null hypothesis is rejected.

### Cosmos atlas

We defined a total of ten annotation regions for coarse analyses. Annotations include the major divisions of the brain only: isocortex, olfactory areas, hippocampal formation, cortical subplate, cerebral nuclei, thalamus, hypothalamus, midbrain, hindbrain and cerebellum. A detailed breakdown of the Cosmos atlas is provided in Extended Data Fig. 5.

### Granger causality

To understand how prior information flows between brain regions, we performed a Granger causality analysis on the Ephys and WFI data during the ITI.

For each Ephys session, we considered the neural activity from −600 ms to −100 ms before stimulus onset, segmented into 50 ms bins, yielding 10 bins per region. For each bin, we predicted the Bayes-optimal from the neural activity using the native LassoCV sklearn function, with its default regularization candidates. This leads to a decoded Bayes-optimal prior for each region and bin. We next used a Granger causality analysis to explore whether the prior information in some region Granger-causes prior information in other regions. Granger analysis was run with the spectral connectivity Python library from the Eden-Kramer lab (https://github.com/Eden-Kramer-Lab/spectral_connectivity).

Given a directed pair of regions (for example, from ACAd to VISp) within a session, the Granger analysis assigns an amplitude to each frequency in the discrete Fourier transform. We calculate an overall Granger score by session by averaging the amplitudes across frequencies[55].

To assess significance of the overall Granger score for a directed pair and session, we build a null distribution by applying our analysis to 1,000 pseudosessions (see the 'Assessing statistical significance' section). After decoding these pseudopriors from neural activity for each region and bin, we perform Granger analysis on these decoded pseudopriors. This creates 1,000 pseudo-Granger scores per directed pair and session. Significance is assessed by comparing the actual Granger score against the top 5% of the pseudo scores.

Granger analysis for the WFI data is very similar to that for Ephys. The main differences are the use the last nine frames before stimulus onset as individual bins and the use of the RidgeCV native function from sklearn for the decoding (see the 'WFI data' section).

Initially, we investigate whether communication between regions exceeds what might be expected by chance. To assess this, we analyse the percentage of significant directed pairs between two regions that significantly reflect the prior; we find an average of 71.6% in WFI and 35.9% in Ephys across sessions. We then repeat this analysis for each session across 1,000 pseudosessions. Subsequently, we assess whether the average percentage of significant pairs across sessions falls within the top 5% of the average percentages calculated from these pseudo-sessions, which indeed it does (Extended Data Fig. 10a).

Next, we explore whether the flow of prior information involves more loops than would typically occur by chance. Specifically, we assess whether triadic loops (A>B>A) within a session occur more frequently than expected. To evaluate this, we calculate the percentage of instances in which a significant Granger pair results in a loop of size 3 for each session. We find that an average (across sessions) of 37.7% in WFI and 10.8% in Ephys exceed what would be anticipated by chance, confirming a higher prevalence of loops (Fig. 2h and Extended Data Fig. 10b).

To obtain Granger graphs at the region level, we use Fisher's method to combine the session-level $P$ values of a directed pair. Lastly, to construct the Granger causality graph at the Cosmos level, we further combine the $P$ values from each directed pair using Fisher's method once again.

### Controlling for region size when comparing decoding scores across Ephys and WFI

With WFI data, the activity signal of a region has always the same dimension across sessions, corresponding to the number of pixels. To control for the effect of region size on the region $R^2$, we performed linear regression across 32 recorded regions to predict the decoding $R^2$ from the number of pixels per region. We found a significant correlation between $R^2$ and the size of the regions (Extended Data Fig. 7d; $R = 0.82$, $P = 9.1 \times 10^{-9}$). To determine whether this accounts for the correlations between Ephys and WFI $R^2$ correlation (Fig. 2d), we subtracted the $R^2$ predicted by region's size from the WFI $R^2$ and recomputed the correlation between Ephys $R^2$ and these size-corrected WFI $R^2$ (Extended Data Fig. 7f).

### Number of recording sessions per region required to reach significance

For each region showing significance in prior decoding, we conducted a subsampling process to see how many recorded sessions were

necessary on average to reach significance. A region $r$ is associated with set of $N_r$ sessions in which its activity was recorded. For a particular region, we can randomly subselect $N \in [1, N_r]$ sessions from this set and test the significance of the region given only these $N$ prior decodings.

For each significant region and each possible number $N$, we repeated this procedure 1,000 times, getting a distribution of $P$ values. We report $p_r^N$, the median of the $P$-value distribution, as a measure of the significance for prior decoding of the region $r$ given that only $N$ recording sessions are available.

For each number of available sessions $N$, we report the fraction of the total number of significant regions for which the statistic $p_r^N$ is less than 0.05, as a measure of the number of recordings per region required to reach back our obtained significance levels.

## Assessing the significance of decoding weights

Let $w_i$ be the weight associated with the neuron $i$—where $i$ ranges from 1 to $U$, with $U$ the number of units, for the decoding of $U$ neurons' activity on a particular region–session. We determine whether the weight is significant by comparing it to the distribution of pseudoweights $\{\hat{w}_i^k, k \in [1, M]\}$ derived from decoding the $M = 200$ pseudosessions priors based on neural activity on that same region-session.

$w_i$ is deemed to be significant if it is higher than the 97.5th percentile of the pseudoweights distribution or lower than the 2.5th percentile.

## Proportion of right choices as a function of the decoded prior

To establish a link between the decoded prior, as estimated from the neural activity, and the mouse behaviour, we plotted the proportion of right choices on zero-contrast trials as a function of the decoded Bayes-optimal prior. For each Ephys (or WFI) session, we decoded the Bayes-optimal prior using all neural activity from that session. We focused this analysis on test trials, held-out during training, according to the procedure described in the 'Nested cross-validation procedure' section. For Fig. 2e, we then pooled over decoded priors for all sessions, assigned them to deciles and computed the associated proportion of right choices. In other words, we computed the average proportion of right choices on trials in which the decoded prior is part of each decile.

To quantify the significance of this effect at a session level (Extended Data Fig. 8a), we additionally performed a logistic regression predicting the choice (right or left) as a function of the decoded prior. Let $j$ be the session number; we predict the actions on that session $a_t^j$ (with $t$ the trial number) as a function of the decoded prior:

$$p(a_t^j = \text{right}) = 1/[1 + \exp(-(\mu^j \times \hat{Y}_t^j + c^j))]$$

with $\hat{Y}_t^j$ the cross-validated decoded Bayes-optimal prior, $\mu^j$ the slope (coefficient of the logistic regression associated with the decoded prior) and $c^j$ an intercept. The logistic regression fitting was performed using the default sklearn LogisticRegression function, which uses L2 regularization on weights with regularization strength $C = 1$.

To assess the statistical significance of these slopes, $\mu^j$, we generated null distributions of slopes over $M = 200$ pseudosessions (pseudosessions are defined in the 'Assessing statistical significance' section). For each pseudosession, we computed the slope of the logistic regression between proportion of correct choices as a function of the decoded pseudo-Bayes-optimal prior. The decoded pseudo-Bayes-optimal prior was obtained by first computing the pseudo-Bayes-optimal prior for each pseudosession, and then using the neural data from the original session to decode this pseudo-Bayes-optimal prior. The percentage of correct choice was more complicated to obtain on pseudosessions because it requires simulating the mice choices as accurately as possible. As we do not have a perfect model of the mouse choices, we had to approximate this step with our best model, that is, the action kernel model. We used the action kernel model fitted to the original behaviour session and simulated it on each pseudosession to obtain the actions on each trial of the pseudosessions.

From the set of decoded pseudo-Bayes-optimal priors and pseudoactions, we obtained $M$ pseudoslopes $\mu_i^j$, $i = 1 \dots M$ using the procedure described above. As the mouse did not experience the pseudosessions or perform the pseudoactions, any positive coefficient $\mu_i^j$ has to be the result of spurious correlations. Formally, to assess significance, we examine whether the mean slope ($\mu = 1/J \times \sum_{j=1}^{J} \mu^j$) is within the 5% top percentile of the averaged pseudoslopes: $\{\mu_i; \mu_i = 1/J \times \sum_{j=1}^{J} \mu_i^j; i \in [1, M]\}$. Extended Data Fig. 8a shows this set of $M$ averaged pseudoslopes as a histogram. The red vertical dashed line is the average slope $\mu$.

When applying this null-distribution procedure in Ephys and WFI data, we find that the pseudoslopes in Ephys data are much more positive than in WFI data. This is due to the fact that spurious correlations in Ephys data are likely induced by drift in the Neuropixels probes, whereas WFI data barely exhibit any drift.

## Neurometric curves

We used the same decoding pipeline described for the Bayes-optimal prior decoding to train a linear decoder of the signed contrast from neural activity in each region, for the ITI $[-600, -100]$ ms and post-stimulus $[0, 100]$ ms intervals. There are 9 different signed contrasts $\{-1, -0.25, -0.125, -0.0625, 0, 0.0625, 0.125, 0.25, 1\}$ where the left contrasts are negative and the right contrasts are positive. Given a session of $T$ trials, we denote $\{s_i\}_{i \in [1, T]}$ the sequence of signed contrasts, $\{\hat{s}_i\}_{i \in [1, T]}$ the cross-validated decoder output given the neural activity $X$ and $\{p_i\}_{i \in [1, T]}$ the Bayes-optimal prior. We defined two sets of trial indices for each session based on the signed contrast $c$ and the Bayes-optimal prior: $I_c^{\text{low}} = \{i | (s_i = c) \text{ and } p_i < 0.5\}$ and $I_c^{\text{high}} = \{i | (s_i = c) \text{ and } p_i > 0.5\}$ corresponding to the trials with signed contrast $c$ and a Bayes-optimal prior lower or higher than 0.5 respectively.

For these sets, we computed the proportions $P_c^{\text{low}} = \#\{\hat{s}_i > 0; i \in I_c^{\text{low}}\}/\#I_c^{\text{low}}$ and $P_c^{\text{high}} = \#\{\hat{s}_i > 0; i \in I_c^{\text{high}}\}/\#I_c^{\text{high}}$. These are the proportions of trials decoded as right stimuli conditioned on the Bayes-optimal prior being higher or lower than 0.5. We fitted a low prior curve to $\{(c, P_c^{\text{low}})\}_{c \in \Gamma}$ and a high prior curve to $\{(c, P_c^{\text{high}})\}_{c \in \Gamma}$, which we called neurometric curves. We used an erf() function from 0 to 1 with two lapse rates for the curves fit to obtain the neurometric curve:

$$f(c) = \gamma + (1 - \gamma - \lambda) \times (\text{erf}((c - \mu)/\sigma) + 1)/2$$

where $\gamma$ is the low lapse rate, $\lambda$ is the high lapse rate, $\mu$ is the bias (threshold) and $\sigma$ is the rate of change of performance (slope). Importantly, we assumed some shared parameters between the low-prior curve and the high-prior curve: $\gamma$, $\lambda$ and $\sigma$ are shared, while the bias $\mu$ is free to be different for low and high prior curves. This assumption of shared parameters provides a better fit to the data compared to models with independent parameters for each curve, as evidenced by lower Bayesian information criterion (BIC) scores during both pre-stimulus ($\Delta$BIC = BIC(independent parameters) − BIC(shared parameters) = 6,482 for Ephys, $\Delta$BIC = 822 for WFI) and post-stimulus periods ($\Delta$BIC = 6,435 for Ephys, $\Delta$BIC = 812 for WFI). We used the psychofit toolbox to fit the neurometric curves using maximal-likelihood estimation (https://github.com/cortex-lab/psychofit). Finally, we estimated the vertical displacement of the fitted neurometric curves for the zero contrast $f^{\text{high}}(c = 0) - f^{\text{low}}(c = 0)$, which we refer to as the neurometric shift.

We used the pseudosession method to assess the significance of the neurometric shift, by constructing a neurometric shift null distribution. $M = 200$ pseudosessions are generated with their signed contrast sequences, which are used as target to linear decoder on the true neural activity. We fitted neurometric curves to the pseudosessions decoder outputs, conditioned on the Bayes-optimal prior inferred from the pseudosessions contrast sequences.

## Stimulus side decoding

To compare the representation of prior information across the brain to the representation of stimulus, we used the stimulus side decoding results from our companion paper[32]. The decoding of the stimulus side

was performed using cross-validated logistic regression with L1 regularization, on a time window of [0, 100] ms after stimulus onset. Only regions with at least two recorded sessions were included, a criteria applied in the companion paper[32]. The bottom panel of Extended Data Fig. 12b is a reproduction from figure 5a of our companion paper[32].

## Embodiment

Video data from two cameras were used to extract 7 behavioural variables which could potentially be modulated according to the mice's subjective prior[35]:

- Paw position (left and right): Euclidean distance of the DLC-tracked paws to a camera frame corner, computed separately for the right and left paw.
- Nose position: horizontal position of the DLC-tracked nose captured by the left camera.
- Wheeling: wheel speed, obtained by interpolating the wheel position at 5 Hz and taking the derivative of this signal.
- Licking: the left and right edges of the tongue are DLC-tracked with both lateral cameras. A lick is defined to have occurred in a frame if the difference for either coordinate to the subsequent frame is larger than 0.25 times the s.d. of the difference of this coordinate across the whole session. The licking signal is defined as the number of licking events during each time bin of 0.02 s.
- Whisking (left and right): motion energy of the whisker-pad area in the camera view, quantified as the mean across pixels of the absolute grey-scale difference between adjacent frames; computed separately for the left- and right-side cameras.

If we are able to significantly decode the Bayes-optimal prior from these behavioural variables during the [−600 ms, −100 ms] ITI, we say that the subject embodies the prior. For the decoding, we used L1-regularized maximum-likelihood regression with the same cross-validation scheme used for neural data (see the 'Nested cross-validation procedure' section). Sessions and trials are subject to the same QC as for the neural data, so that we decode the same sessions and the same trials as the Ephys session-level decoding. For each trial, the decoder input is the average over the ITI [−600, −100] ms of the behavioural variables. For a session of $T$ trials, the decoder input is a matrix of size $T \times 7$ and the target is the Bayes-optimal prior. We use the pseudosession method to assess the significance of the DLC features decoding score $R^2$. To investigate the embodiment of the Bayes-optimal prior signal, we compared session-level decoding of the prior signal from DLC regressors to region–session-level decoding of the prior signal from the neural activity of each region during the session.

## DLC residual analysis

The DLC prior residual signal is the part of the prior signal which was not explained away by the DLC decoding, defined as the prior signal minus the prediction of the DLC decoding. We decoded this DLC prior residual signal from the neural activity, using the same linear decoding schemes as previously described.

## Eye position decoding

Video data from the left camera were used to extract the eye position variable, a 2D signal corresponding to the position of the centre of the mouse pupil relative to the video border. The camera as well as the mouse's head were fixed. DeepLabCut was not able to achieve sufficiently reliable tracking of the pupils; we therefore used a different pose-estimation algorithm[56], trained on the same labelled dataset used to train DeepLabCut. For the decoding, we used L2-regularized maximum-likelihood regression with the same cross-validation scheme used for neural data, during the [−600 ms, −100 ms] ITI.

The eye-position prior residual signal is the part of the prior signal which is not explained away by the eye position decoding, defined as the prior signal minus the prediction of the eye position decoding.

We decode this eye position prior residual signal from the neural activity of early visual areas (LGd and VISp) using the same linear decoding schemes as previously described.

## Contribution of DLC and eye-position features to prior embodiment: feature importance

To assess the contribution of DLC and eye position features to prior embodiment, we performed a leave-one-out decoding procedure of the DLC + eye position features. There are five different types of DLC features: licking, wheeling, nose position, whisking and paws positions. Moreover, with the $x$ and $y$ coordinates of the eye position, we had a total of seven types of variables for which we individually performed a separate leave-one-out decoding analysis. The difference between the full decoding $R^2$ and the leave-one-out decoding $R^2$ is a measure of the importance of the knocked-out variable in the full decoding.

## Behavioural models

To determine the behavioural strategies used by the mice, we developed several behavioural models and used Bayesian model comparison to identify the one that fits best. We considered three types of behavioural models that differ as to how the integration across trials is performed (how the subjective prior probability that the stimulus will be on the right side is estimated based on history). Within a trial, all models compute a posterior distribution by taking the product of a prior and a likelihood function (the probability of the noisy contrast given the stimulus side; Supplementary information).

Among the three types of models of the prior, the first, called the Bayes-optimal model, assumes knowledge of the generative process of the blocks. Block lengths are sampled as follows:

$$p(l_k = N) \propto \exp(-N/\tau) \times \mathbb{1}[20 \leq N \leq 100]$$

with $l_k$ the length of block $k$ and $\mathbb{1}$ the indicator function. Block lengths are therefore sampled from an exponential distribution with parameter $\tau = 60$ and constrained to be between 20 and 100 trials. When block $k-1$ comes to an end, the next block $b_k$, with length $l_k$, is defined as a right block (where the stimulus is likely to appear more frequently on the right) if block $b_{k-1}$ was a left block (where the stimulus was likely to appear more frequently on the left) and conversely. During left blocks, the stimulus is on the left side with probability $\gamma = 0.8$ (and similarly for right blocks). Defining $s_t$ as the side on which the stimulus appears on trial $t$, the Bayes-optimal prior probability the stimulus will appear on the right at trial $t$, $p(s_t|s_{1:(t-1)})$ is obtained through a likelihood recursion[57].

The second model of the subjective prior, called the stimulus kernel model[58], assumes that the prior is estimated by integrating previous stimuli with an exponentially decaying kernel. Defining $s_{t-1}$ as the stimulus side on trial $t-1$, the prior probability that the stimulus will appear on the right $\pi_t$ is updated as follows:

$$\pi_t = (1 - \alpha) \times \pi_{t-1} + \alpha \times \mathbb{1}[s_{t-1} = \text{right}]$$

with $\pi_{t-1}$ the prior at trial $t-1$ and $\alpha$ the learning rate. The learning rate governs the speed of integration: the closer $\alpha$ is to 1, the more weight is given to recent stimuli $s_{t-1}$.

The third model of the subjective prior, called the action kernel model, is similar to the stimulus kernel model but assumes an integration over previous chosen actions with, again, an exponentially decaying kernel. Defining $a_{t-1}$ as the action at trial $t-1$, the prior probability that the stimulus will appear on the right $\pi_t$ is updated as follows:

$$\pi_t = (1 - \alpha) \times \pi_{t-1} + \alpha \times \mathbb{1}[a_{t-1} = \text{right}]$$

For the Bayes-optimal and stimulus kernel models, we additionally assume the possibility of capturing a simple autocorrelation between

choices with immediate or multistep repetition biases or choice- and outcome-dependent learning rate[40,59]. Further details on model derivations are provided in the Supplementary Information.

## Model comparison

To perform model comparison, we implemented a session-level Bayesian cross-validation procedure. In this procedure, for each mouse with multiple sessions, we held out one session $i$ and fitted the model on the held-in sessions. For each mouse, given a held-out session $i$, we fitted each model $k$ to the actions of held-in sessions, denoted here as $A^{\setminus i}$, and obtained the posterior probability, $p(\theta_k|A^{\setminus i}, m_k)$, over the fitted parameters $\theta_k$ through an adaptive Metropolis–Hastings procedure[60]. Four adaptive Metropolis–Hastings chains were run in parallel for a maximum of 5,000 steps, with the possibility of early stopping (after 1,000 steps) implemented with the Gelman–Rubin diagnostic[61]. $\theta_k$ typically includes sensory noise parameters, lapse rates and the learning rate (for stimulus and action kernel models); the formal definitions of these parameters are provided in the Supplementary Information. Let $\{\theta_{k,n}; n \in [1, N_{MH}]\}$ be the $N_{MH}$ samples obtained with the Metropolis–Hastings (MH) procedure for model $k$ (after discarding the burn-in period). We then computed the marginal likelihood of the actions on the held-out session, denoted here as $A^i$.

$$p(A^i|A^{\setminus i}, m_k) = \int p(A^i, \theta_k|A^{\setminus i}, m_k)\mathrm{d}\theta_k$$
$$= \int p(A^i|\theta_k, m_k)p(\theta_k|A^{\setminus i}, m_k)\mathrm{d}\theta_k$$
$$\approx \frac{1}{N_{MH}} \sum_n^{N_{MH}} p(A^i|\theta_{k,n}, m_k)$$

For each subject, we obtained a score per model $k$ by summing over the log-marginal likelihoods $\log p(A^i|A^{\setminus i}, m_k)$, obtained by holding out one session at a time. Given these subject-level log-marginal likelihood scores, we performed Bayesian model selection[38] and reported the model frequencies (the expected frequency of the $k$th model in the population) and the exceedance probabilities (the probability that a particular model $k$ is more frequent in the population than any other considered model).

## Assessing the statistical significance of the decoding of the action kernel prior

Given that the action kernel model better accounts for the mouse behaviour, it would be desirable to assess the statistical significance of the decoding of the action kernel prior. Crucially, as assessing significance involves a null hypothesis (the neural activity is independent of the prior), a rigorous construction of the corresponding null distribution is key.

For the Bayes optimal prior decoding, constructing the null distribution is straightforward. It requires that we generate stimulus sequences with the exact same statistics as those experienced by the mice. We do this by simulating the same generative process used to generate the stimulus during the experiment, yielding what we called pseudosessions in previous sections.

However, for the action kernel prior (and contrary to the Bayes optimal model), we also need to generate action sequences with the same statistics as those generated by the animals. In turn, this would require a perfect model of how the animals make decisions. As we lack such a model, we would need to come up with an approximation. There are multiple approximations that we could use, including the following:

- Synthetic sessions, in which we use the action kernel model, using the parameters fitted to each mouse on each session, to generate fake responses. However, the action kernel model is not a perfect model of the animal's behaviour, it is merely the best model that we have among the ones we have tested. Moreover, there could be some concerns about the statistical validity of using a null distribution,

which assumes that the action kernel is the perfect model when testing for the presence of this same model in the mouse's neural activity
- Imposter sessions, in which we use responses from other mice. However, other animals are most unlikely to have used the exact same model/parameters as the mouse we are considering. This implies that the actions in these imposter sessions do not have the same statistics as the decoded session. There is indeed a large degree of between-session variability, as can be seen from the substantial dispersion in the fitted action kernel decay constants shown in Fig. 4b.
- Shifted sessions, in which we decode the action kernel prior on trial $M$, using the recording on trial $M + N$, with periodic boundaries for the 'edges'[34]. The problems here are twofold. First, $N$ must be chosen large enough such that the block structure of the shifted session is independent of the block structure of the non-shifted session. Because blocks are about 50 trials long, $N$ must be large for the independence assumption to hold. This adds a constraint on the number of different shifted sessions that we can generate, leading to a poor null distribution with little diversity (made from only a few different shifted sessions). Second, it has been shown that there is within-session variability[62] such that when $N$ is chosen to be large, we cannot consider the shifted actions to have the same statistics as the non-shifted actions.

There may be other options. However, as they would all rely on approximations, the degree of statistical inaccuracy associated with their use would be unclear. We would not even know which one to favour, as it is hard to establish the quality of the approximations. Overall, we have access to the exact generative process to construct the null distribution for the Bayes optimal prior, versus only approximations for the action kernel prior. As a result, we decided to err on the side of caution and focus primarily on the Bayes optimal prior decoding whenever possible. In analyses involving animal behaviour—such as in Fig. 2e (see the 'Proportion of right choices as a function of the decoded prior' section) and Fig. 4e (see the 'Orthogonalization' section)—we had to rely on an approximation. For these, we used the synthetic session approach to establish a null distribution.

## Orthogonalization

To assess the dependency on past trials of the decoded Bayes-optimal prior from neural activity, we performed stepwise linear regression as a function of the previous actions (or previous stimuli). The Bayes-optimal prior was decoded from neural activity at the session level, therefore considering the activity from all accessible cortical regions in WFI and all units in Ephys.

The stepwise linear regression involved the following steps. We started by linearly predicting the decoded Bayes-optimal prior on trial $t$ from the previous action (action on trial $t - 1$), which enables us to compute a first-order residual, defined as the difference between the decoded neural prior and the decoded prior predicted by the last action. We then used the second-to-last action (action at trial $t - 2$) to predict the first-order residual to then compute a second-order residual. We next predicted the second-order residual with the third-to-last action and so on. We use this iterative stepwise procedure to take into account possible autocorrelations in actions.

The statistical significance of the regression coefficients is assessed as follows. Let us use $T_j$ to denote the number of trials of session $j$, $Y^j \in R^{T_j}$ the decoded Bayes-optimal prior and $X^j \in R^{T_j \times K}$ the chosen actions, where $K$ is the number of past trials considered in the stepwise regression. When running the stepwise linear regression, we obtain a set of weights $\{W_k^j, k \in [1, K]\}$, with $W_k^j$ the weight associated with the $k$th-to-last chosen action. We test for the significance of the weights for each step $k$, using as a null hypothesis that the weights associated with the $k$th-to-last chosen action are not different from weights predicted by the 'pseudosessions' null distribution.

To obtain a null distribution, we followed the same approach as described in the 'Proportion of right choices as a function of the

decoded prior' section. Thus, we generated decoded pseudo-Bayes-optimal priors and pseudoactions. For each session, these pseudovariables are generated as follows: first, we fitted the action kernel model (our best-fitting model) to the behaviour of session $j$. Second, we generated $M$ pseudosessions (see the 'Assessing the statistical significance' section). Lastly, we simulated the fitted model on the pseudosessions to obtain pseudoactions. Regarding the decoded pseudo-Bayes-optimal priors, we first infer with the Bayes-optimal agent, the Bayes-optimal prior of the pseudosessions, and second, we decoded this pseudoprior with the neural activity. For each session $j$ and pseudo-$i$, we have generated a decoded pseudo-Bayes-optimal prior $Y_i^j$ as well as pseudoactions $X_i^j$. When applying the stepwise linear regression procedure to the couple $(X_i^j, Y_i^j)$, we obtain a set of pseudoweights $\{W_{k,i}^j, k \in [1, K]\}$. As the mouse did not experience the pseudosessions or perform the pseudoactions, any non-zero coefficients $W_{k,i}^j$ must be the consequence of spurious correlations. Formally, to assess significance, we examine whether the average of the coefficients over sessions $W_k = 1/N_{\text{sessions}} \times \sum_{j=1}^{N_{\text{sessions}}} W_k^j$ is within the 5% top percentile of $\{W_{k,i}; W_{k,i} = 1/N_{\text{sessions}} \times \sum_{j=1}^{N_{\text{sessions}}} W_{k,i}^j; i \in [1, M]\}$.

The statistical significance procedure when predicting the decoded Bayes-optimal prior from the previous stimuli is very similar to the one just described for the previous actions. The sole difference is that, for this second case, we do not need to fit any behavioural model to generate pseudostimuli. Pseudostimuli for session $j$ are defined when generating the $M$ pseudosessions. Pseudoweights are then obtained by running the stepwise linear regression predicting the decoded pseudo-Bayes-optimal prior from the pseudostimuli. Formal statistical significance is established in the same way as for the previous actions case.

When applying this null-distribution procedure to Ephys and WFI, we find that the strength of spurious correlations (as quantified by the amplitude of pseudoweights $W_{k,i}$) for Ephys is much greater than for WFI data. This is due to the fact that spurious correlations in electrophysiology are mainly produced by drift in the Neuropixels probes, which is minimal in WFI.

## Behavioural signatures of the action kernel model

To study why the Bayesian model selection procedure favours the action kernel model, we sought behavioural signatures that can be explained by this model but not the others. As the action kernel model integrates over previous actions (and not stimuli sides), it is a self-confirmatory strategy. This means that, if an action kernel agent was incorrect on a block-conformant trial (trials in which the stimulus is on the side predicted by the block prior), then it should be more likely to be incorrect on the subsequent trial (if it is also block-conformant). Other models integrating over stimuli, such as the Bayes-optimal or the stimulus Kernel model, are not more likely to be incorrect after an incorrect trial, because they can use the occurrence or non-occurrence of the reward to determine the true stimulus side, which could then be used to update the prior estimate correctly. To test this, we analysed the proportion correct of each session at trial $t$, conditioned on whether it was correct or incorrect at trial $t - 1$. To isolate the impact of the last trial, and not previous trials or other factors such as block switches and structure, we restricted ourselves to the following:
- Zero-contrast trials.
- Trial $t$, $t - 1$ and $t - 2$ had stimuli that were on the expected, meaning block-conformant, side.
- On trial $t - 2$, the mouse was correct, meaning that it chose the block-conformant action.
- On trials that were at least ten trials from the last reversal.

## Neural signature of the action kernel model from the decoded Bayes-optimal prior

To test whether the behavioural signature of the action kernel model discussed in the previous section is also present in the neural activity, we simulated an agent of which the decisions are based on the cross-validated decoded Bayes-optimal prior and tested whether this agent also shows the same action kernel signature. The decoded Bayes-optimal prior was obtained by decoding the Bayes-optimal prior from the neural activity (see the 'Nested cross-validation procedure' section) on a session-level basis, considering all available WFI pixels or Ephys units.

Note that, if the decoded Bayes-optimal agent exhibits the action kernel behavioural signature, this must be a property of the neural activity as the Bayes-optimal prior on its own cannot produce this behaviour.

The agent is simulated as follows. Let us denote $Y \in R^N$ the Bayes-optimal prior with $N$ is the number of trials. When performing neural decoding of the Bayes-optimal prior $Y$, we obtain a cross-validated decoded Bayes-optimal prior $\hat{Y}$. We define an agent which, on each trial, greedily selects the action predicted by the decoded Bayes-optimal prior $\hat{Y}$, meaning that the agent chooses right if $\hat{Y} > 0.5$, and left otherwise.

On sessions that significantly decoded the Bayes optimal prior, we then test whether the proportion of correct choices depends on whether the previous trial was correct or incorrect. We do so at the session level, applying all but one criterion of the behavioural analysis described previously in the 'Behavioural signatures of the action kernel model' section:
- Trial $t$, $t - 1$ and $t - 2$ had stimuli that were on the expected, meaning on the block-conformant, side.
- On trial $t - 2$, the mouse was correct, meaning that it chose the block-conformant action.
- On trials that were at least ten trials from the last reversal.

Note that, given that the neural agent uses the pre-stimulus activity to make its choice, we do not need to restrict ourselves to zero-contrast trials.

## Neural decay rate

To estimate the temporal dependency of the neural activity in Ephys and WFI, we assumed that the neural activity was the result of an action kernel (or stimulus kernel) integration and fitted the learning rate (inverse decay rate) of the kernel to maximize the likelihood of observing the neural data.

We first describe the fitting procedure for WFI data. Given a session, let us call $X_{t,n}$ the WFI activity of the $n$th pixel for trial $t$. Similarly to the procedure that we used for decoding the Bayes-optimal prior, we took the activity at the second-to-last frame before stimulus onset. We assumed that $X_{t,n}$ is a realization of Gaussian distribution with mean $Q_{t,n}$ and with s.d. $\sigma_n$, $X_{t,n} \sim N(Q_{t,n}, \sigma_n)$. $Q_{t,n}$ was obtained through an action kernel (or stimulus kernel) integration process:

$$Q_{t,n} = (1 - \alpha_n) \times Q_{t-1,n} + \alpha_n \times \zeta_n \times a_{t-1}$$

with $\alpha_n$ the learning rate, $a_{t-1} \in \{-1, 1\}$ the action at trial $t - 1$ and $\zeta_n$ a scaling factor. $\alpha_n$, $\zeta_n$ and $\sigma_n$ are found by maximizing the probability of observing the widefield activity $p(X_{1:T,n}|a_{1:T}; \alpha_n, \zeta_n, \sigma_n)$, with $1:T = \{1, 2, \ldots T\}$ and $T$ the number of trials in that session. If a trial is missed by the mouse, which occurs when reaction time exceeds 60 s (1.5% of the trials, see the companion paper[32]), $Q_{t,n}$ is not updated. For the electrophysiology now, let us call $X_{t,n}$ the neural activity of unit $n$ at trial $t$. Similarly to what we did when decoding the Bayes-optimal prior, we took the sum of the spikes between −600 and −100 ms from stimulus onset. We assumed here that $X_{t,n}$ is a realization of a Poisson distribution with parameter $Q_{t,n}$, $X_{t,n} \sim \text{Poisson}(Q_{t,n})$. $Q_{t,n}$ was obtained through an action kernel (or stimulus kernel) integration process:

$$Q_{t,n} = (1 - \alpha_n) \times Q_{t-1,n} + \alpha_n \times \zeta_n^{a_{t-1}}$$

with $\alpha_n$ the learning rate and $\zeta_n^{a_{t-1}}$ scaling factors, one for each possible previous action. Note that in the case of Ephys, as $Q_{t,n}$ can only be

positive, two scaling factors are necessary to define how $Q_{t,n}$ is adjusted after a right or left choice. $\alpha_n$, $\zeta_n^1$ and $\zeta_n^{-1}$ are found by maximizing the probability of observing the Ephys activity $p(X_{1:T,n}|a_{1:T}; \alpha_n, \zeta_n^1, \zeta_n^{-1})$, with $1:T = \{1, 2, \dots T\}$ and $T$ the number of trials in that session. For electrophysiology, we added constraints on the units. Specifically, we only considered units (1) of which the median (pre-stimulus summed) spikes was not 0; (2) with at least 1 spike every 5 trials; and (3) where the distribution of (pre-stimulus summed) spikes was different when the Bayes-optimal prior is greater versus lower than 0.5 (significance is asserted when the $P$ value of a Kolmogorov–Smirnov test was below 0.05).

To restrict our analysis to units (or pixels) which are likely to reflect the subjective prior, we considered only those that were part of regions-sessions that significantly decoded the Bayes-optimal prior, resulting in $N = 164$ sessions for Ephys and $N = 46$ for WFI. Significance was assessed according to the pseudosession methodology (see the 'Assessing statistical significance' section), which accounts for spurious correlations (which a unit-level Kolmogorov–Smirnov test would not). Then, to obtain a session-level neural learning rate, we averaged across pixel-level or unit-level learning rates. To compare neural and behavioural temporal timescales, we correlated the session-level neural learning rate with the behavioural learning rate, obtained by fitting the action kernel to the behaviour.

In both Ephys and WFI, when considering that the neural activity is a result of the stimulus kernel, the calculations were all identical except replacing actions $a_{1:T}$ with stimuli side $s_{1:T}$.

This analysis (Fig. 4f) makes the assumption that sessions could be considered to be independent from another—an assumption that can be questioned given that we have a total of 459 sessions across 139 mice in Ephys and 52 sessions across 6 mice in WFI. To test the presence of the correlation between neural and behavioural timescales while relaxing this assumption, we developed a hierarchical model that takes into account the two types of variability, within mice and within sessions given a mouse. This model defines session-level parameters, which are sampled from mouse-level distributions, which are themselves dependent on population-level distributions. The exact definition of the hierarchical model is provided in the Supplementary information. This hierarchical approach confirmed the session-level correlation between neural and behavioural timescales (Extended Data Fig. 15b).

## Reporting summary

Further information on research design is available in the Nature Portfolio Reporting Summary linked to this article.

## Data availability

All data supporting the findings of this study are available at GitHub (https://int-brain-lab.github.io/iblenv/notebooks_external/data_release_brainwidemap.html and https://int-brain-lab.github.io/iblenv/notebooks_external/loading_widefield_data.html). Detailed information on each recorded region—including the number of recordings, neurons and decoding scores—is provided at GitHub (https://github.com/int-brain-lab/paper-brain-wide-map/blob/plotting/brainwidemap/meta/region_info.csv). Users are allowed to distribute, remix, adapt and build on the material in any medium or format, provided that attribution is given to the creator (data license, CC-BY). The Swanson flat map is available at GitHub (https://int-brain-lab.github.io/iblenv/notebooks_external/atlas_swanson_flatmap.html).

## Code availability

The code associated with this paper is available at GitHub (https://github.com/int-brain-lab/prior-localization/tree/main).

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

**Acknowledgements** This work was supported by grants from the Wellcome Trust (209558 and 216324), the Simons Foundation, The National Institutes of Health (NIH U19NS12371601), the National Science Foundation (NSF 1707398), the Gatsby Charitable Foundation (GAT3708) and the Max Planck Society and the Humboldt Foundation. We thank our colleagues in the International Brain Laboratory (IBL) consortium, whose collaborative platform and shared resources were essential to this research, and the staff at the University of Geneva for providing the computational support that made these results possible.

**Author contributions** Level of contribution is indicated in parentheses (H, high; M, medium; L, low). Conceptualization: C.F. (H), F. Hubert (H), M.C. (M), A.K.C. (M), E.E.J.D. (H), K.D.H. (L), M.H. (L), S.B.H. (L), P.E.L. (L), Z.M. (L), R.S. (L), I.B.W. (L), A.Z. (M), P.D. (H) and A.P. (H). Formal analysis: C.F. (H), F. Hubert (H), B.B. (L), G.A.C. (L), B.G. (L), J.P. (L), R.S. (L), M.S. (L), Y.S. (L), M.R.W. (L), A.Z. (M), Y.Z. (L), P.D. (M) and A.P. (H). Funding acquisition: M.C. (M), A.K.C. (M), E.E.J.D. (M), T.A.E. (M), I.R.F. (H), K.D.H. (H), M.H. (H), S.B.H. (M), P.E.L. (H), Z.M. (M), T.D.M.-F. (L), L.P. (L), J.P. (M), K.S. (M), I.B.W. (L), P.D. (M) and A.P. (H). Experimental investigation—Ephys: J.B. (H), J.A.C. (H), G.A.C. (L), F.D. (H), L.F.-S. (L), F. Hu (H), A.K. (H), C.K. (H), P.Y.P.L. (H), G.T.M. (H), N.J.M. (H), J.P.-N. (H), A.P.-V. (H), N.R. (H), K.Z.S. (H), A.E.U. (H) and A.P. (L). Experimental investigation—WFI: C.K. (H) and A.Z. (M). Methodology: C.F. (H), F. Hubert (H), L.A. (L), B.B. (H), N.B. (H), S.B. (L), G.A.C. (L), E.E.J.D. (L), B.G. (L), K.D.H. (L), M.H. (L), C.L. (L), L.P. (L), R.S. (L), N.A.S. (L), K.S. (L), M.R.W. (L) and P.D. (M). Project administration: C.F. (H), G.A.C. (L), A.K.C. (H), E.E.J.D. (M), S.B.H. (L), R.S. (L), P.D. (M) and A.P. (H). Resources: C.F. (H), N.B. (H), M.C. (M), G.A.C. (L), A.K.C. (H), Y.D. (L), K.D.H. (M), M.H. (L), S.B.H. (H), Z.M. (L), T.D.M.-F. (H), L.P. (L), S.J.W. (M), O.W. (M) and A.P. (L). Software (pipeline development): C.F. (H), F. Hubert (H), B.B. (M), D.B. (L), N.B. (H), E.K.B. (M), G.A.C. (L), E.E.J.D. (L), M.F. (H), M.A.F. (H), B.G. (M), K.D.H. (L), J.M.H. (H), C.A.L. (L), C.R. (L), R.S. (L), M.J.W. (H), M.R.W. (M) and O.W. (H). Supervision: C.F. (H), M.C. (M), G.A.C. (L), A.K.C. (H), Y.D. (M), E.E.J.D. (M), T.A.E. (L), K.D.H. (L), M.H. (M), S.B.H. (M), Z.M. (L), T.D.M.-F. (M), L.P. (L), N.A.S. (L), K.S. (L), I.B.W. (L), A.Z. (L), P.D. (M) and A.P. (H). Validation: C.F. (H), F. Hubert (H), B.B. (L), N.B. (H), G.A.C. (L), A.K.C. (H), I.R.F. (L), R.S. (L), Y.S. (M), Y.Z. (L) and P.D. (L). Visualization: C.F. (H), F. Hubert (H), D.B. (L), K.N. (L), C.R. (M), P.D. (L) and A.P. (H). Writing—original draft preparation: C.F. (H), F. Hubert (H), G.A.C. (L), P.D. (M) and A.P. (H). Writing—review and editing: C.F. (H), F. Hubert (H), L.A. (L), N.B. (L), G.A.C. (L), A.K.C. (L), E.E.J.D. (L), P.E.L. (M), Z.M. (L), J.P.-N. (L), L.P. (L), J.P. (L), N.A.S. (L), C.T. (L), P.D. (M) and A.P. (H).

**Funding** Open access funding provided by University of Geneva.

**Competing interests** The authors declare no competing interests.

**Additional information**
**Correspondence and requests for materials** should be addressed to Charles Findling.

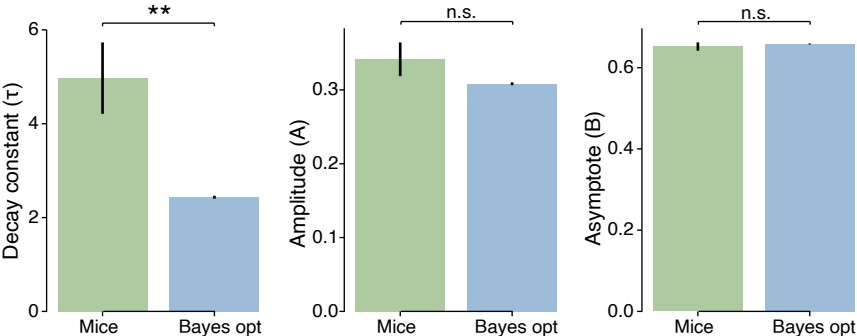

**Extended Data Fig. 1 | Bar plots of the decay constant (τ), amplitude (*A*) and asymptote (*B*) of the zero contrast reversal curves across all mice.** The parameters are obtained by fitting the following parametric curve: $p(correct\ at\ trial\ t) = B$ on the zero contrast pre-reversal trials (the 5 trials before a block switch) and $p(correct\ at\ trial\ t) = B + (A - B) \cdot e^{-t/\tau}$ on the zero contrast post-reversal trials (the 20 trials after a block switch). $\tau$ reflects the reversal timescale. To make up for the limited amount of available zero contrast reversal trials, we fit these curves using a jackknife procedure (see Methods). Bars and error bars indicate jackknife means ± SEM (jackknifing was applied on n = 139 mice). Mice have a significantly longer mean recovery decay constant than the Bayes-optimal observer (4.97 vs 2.43 trials, 2-tailed paired t-test t = 2.94, $p = 0.001$), while the other parameters are not significantly different. (for *A*: t = 1.43, $p = 0.15$ and for *B*: t = −0.64, $p = 0.53$) (\*\**p* < 0.01, n.s. not significant).

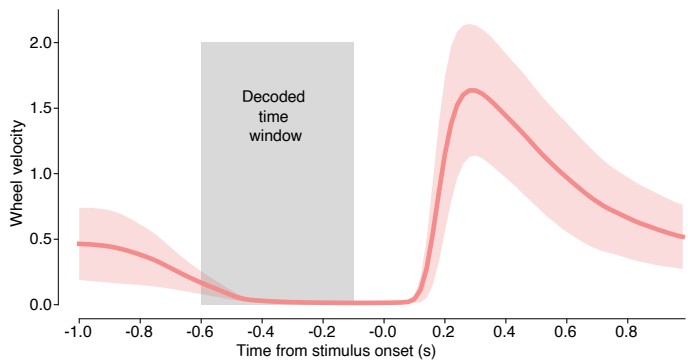

**Extended Data Fig. 2 | Average wheel speed averaged across sessions before and after stimulus onset.** The decoded time window used for Ephys data is indicated in light grey. For WFI, the data was decoded on the second-to-last frame relative to the stimulus onset, corresponding to a time window that ranges from −198 to −132 ms at the start to −132 to −66 ms at the end, depending on the timing of the last frame before the stimulus onset (this last frame can occur anytime between −132 and −66 ms before the stimulus onset).

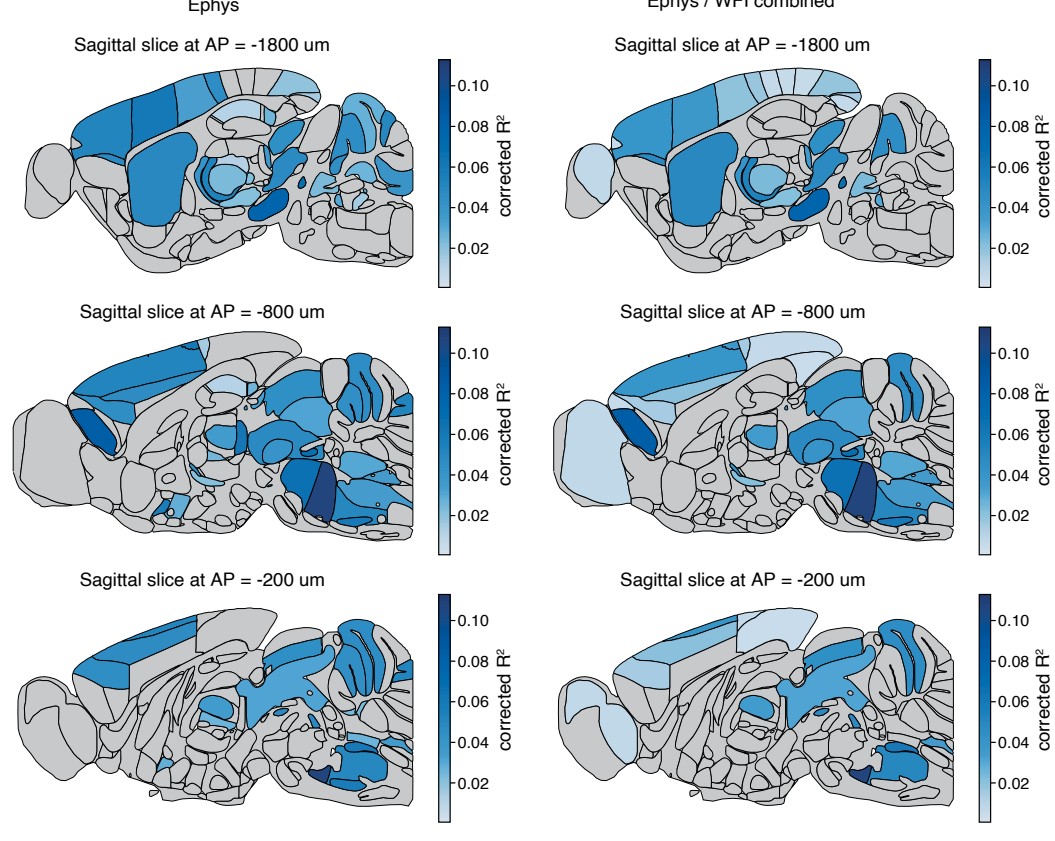

Ephys

Ephys / WFI combined

Sagittal slice at AP = -1800 um

Sagittal slice at AP = -1800 um

Sagittal slice at AP = -800 um

Sagittal slice at AP = -800 um

Sagittal slice at AP = -200 um

Sagittal slice at AP = -200 um

30.2% of significant regions (73/242)

24.0% of significant regions (58/242)

**Extended Data Fig. 3 | Encoding of the prior across the brain during the inter-trial interval.** Sagittal slices corresponding to the main decoding figure presented in Fig. 2b. Left: Ephys only. A region is deemed significant if the Fisher combined $p$-value is lower than 0.05. Right: Ephys and Widefield combined. Significance for regions is assessed with the Benjamini-Hochberg procedure, correcting for multiple comparisons, with a conservative false discovery rate of 1%.

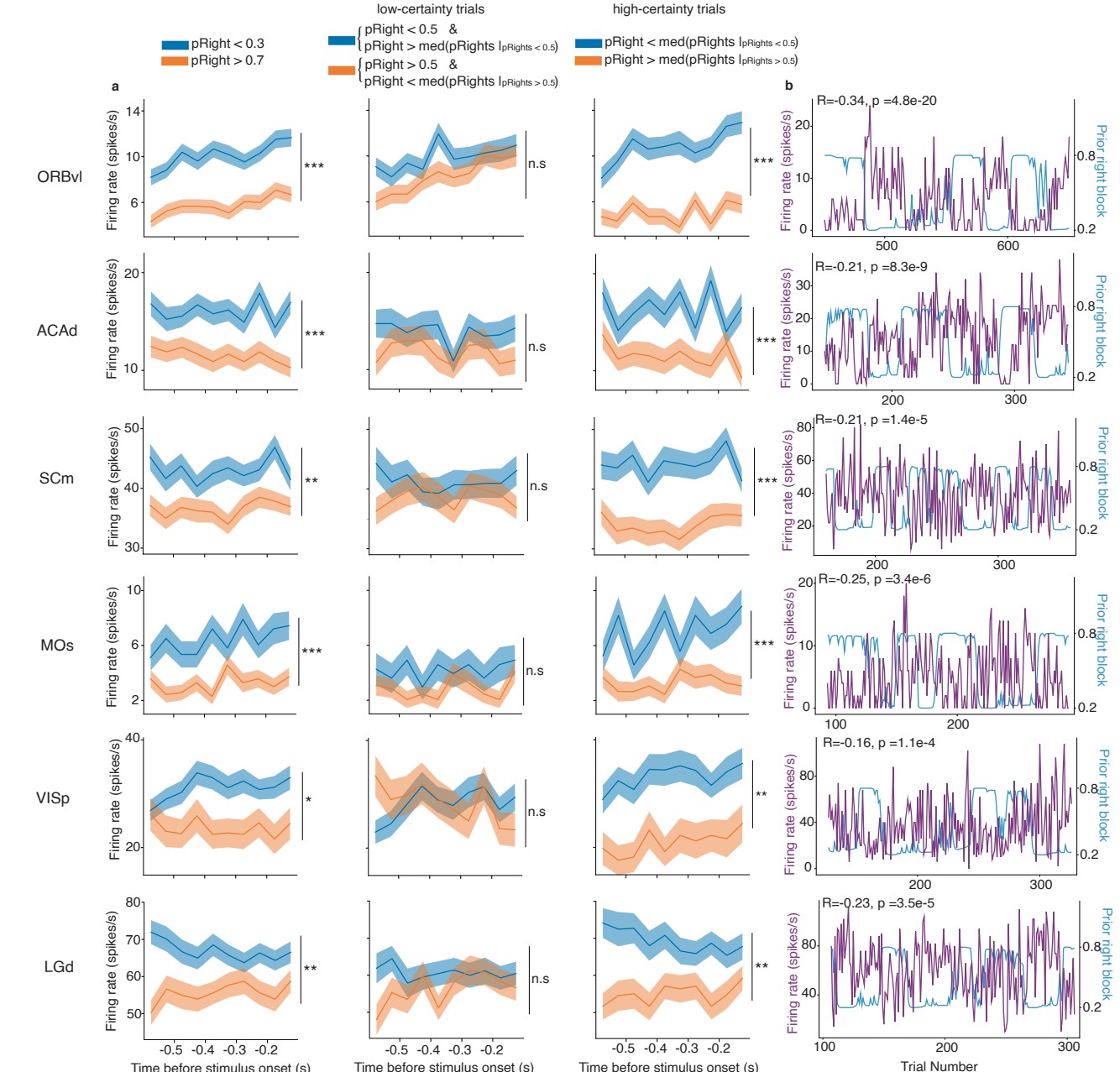

**Extended Data Fig. 4 | Six examples of neurons significantly encoding the Bayes-optimal prior (\*\*\*p < 0.001, \*\*p < 0.01, \*p < 0.05). a**. Peri-Stimulus Time Histograms (PSTHs) segmented by trials throughout the session. Left column conditions on the Bayes-optimal prior for the right side being less than 0.3 (blue) vs greater than 0.7 (orange). The middle and right columns depict PSTHs for trials under conditions of low certainty (pRight close to 0.5) and high certainty (pRight far from 0.5), respectively. "Med" refers to the median operation. Significance is assessed by testing the difference between the trial wise firing rates (averaging across time bins) of "left" (blue) and "right" (orange) trials with a two-sample Kolmogorov-Smirnov test. **b**. Spike counts of the neurons (purple line) during the intertrial interval in the [−600, −100] millisecond

time window before stimulus onset, along with the Bayes-optimal prior (blue) for a subset of trials within the session (Spearman correlations of the full session are reported on the graphs). All neurons on this panel show a preference for the left side, although, at the population level, we did not observe a bias for either the right or left side. Indeed, we examined the distribution of decoding weights and detected no discernible lateral bias concerning the weight distribution. Testing the significance of the decoder weight in each region yielded adjusted p-values all above 0.2 (Wilcoxon test), after adjusting for multiple comparisons using the Benjamini-Hochberg correction. Additionally, a combined analysis of all weights from the six regions lead to the same conclusion (two tailed signed Wilcoxon test: t = 16732, p-value = 0.31).

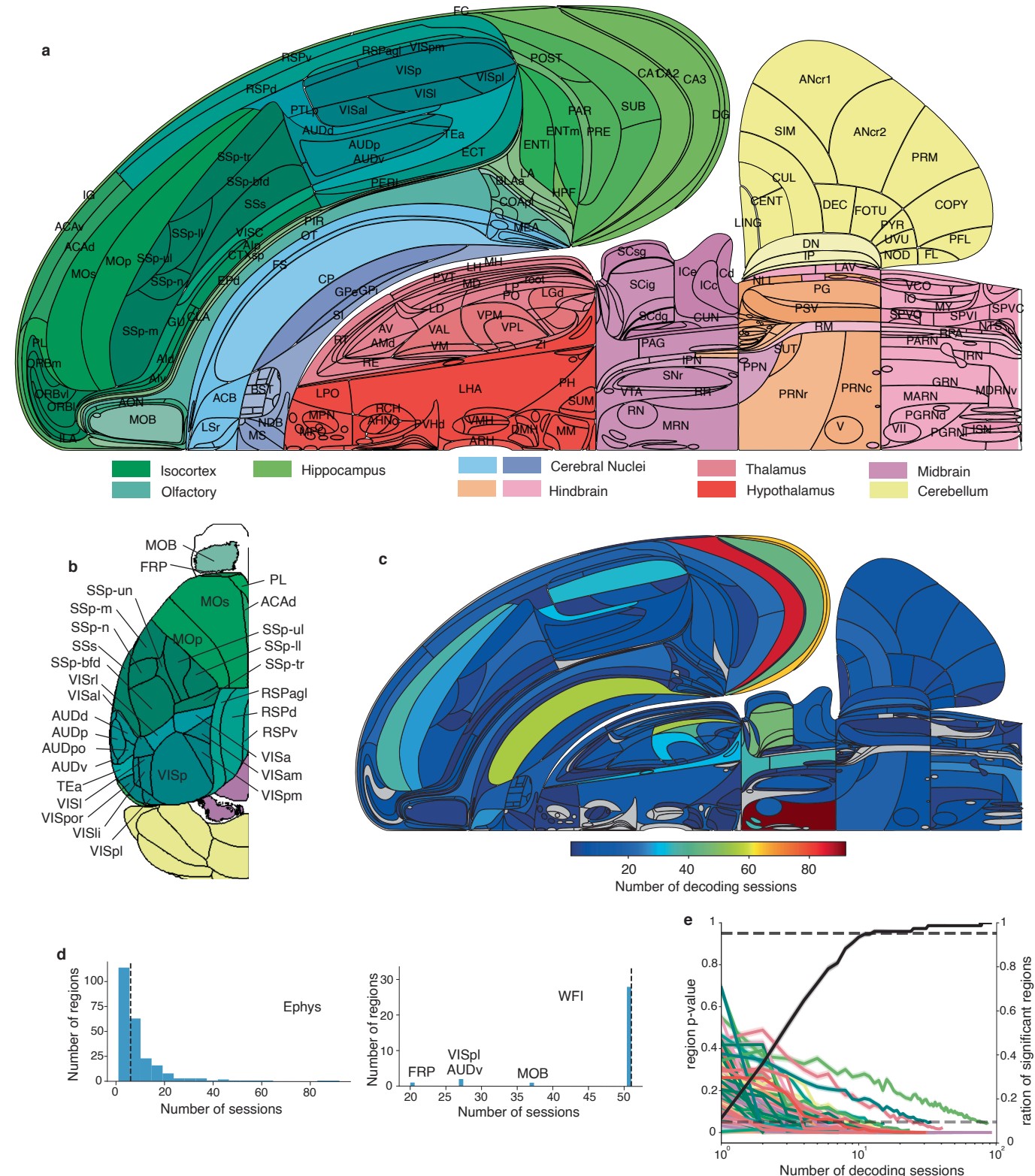

**Extended Data Fig. 5 | a.** Swanson map colour-coded to indicate distinct anatomical regions (Cosmos level). **b.** Dorsal brain slice recorded with widefield imaging, utilizing the same colour scheme for regional identification. **c.** Distribution of decoding sessions across different regions as mapped in the Swanson brain atlas. **d.** Histogram detailing the number of sessions per recording type: Electrophysiology (Ephys) and Widefield Imaging (WFI). The vertical lines indicate median values, with 6 sessions for Ephys and 51 for WFI. **e.** Dual-axis graph: the coloured lines (left axis) display the p-value of regional significance as a function of the number of decoding sessions, while the black line (right axis) shows the ratio of significant regions relative to the total number of sessions. It is estimated that approximately 10 sessions per region are necessary to identify 95% of significant regions highlighted in the main decoding analysis (refer to Methods section for more details). It is important to recognize that this analysis has limitations: it assumes uniformity across recordings and regions without considering, e.g., variations in effect size or number of units per recording. Despite these limitations, we concentrated on the number of recordings because it is a primary factor that experimenters can directly control.

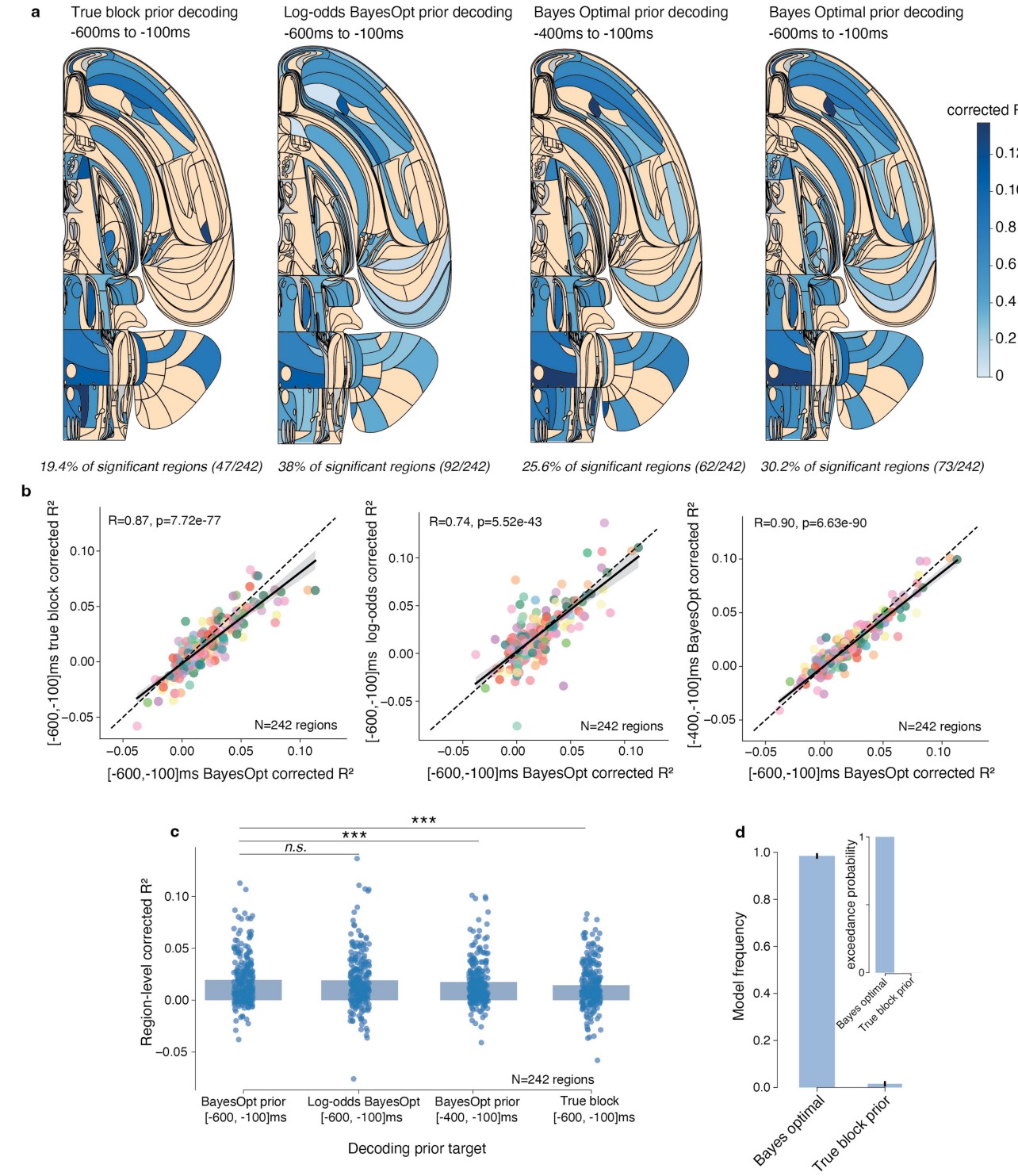

**Extended Data Fig. 6 | a.** Swanson maps showing corrected decoding $R^2$ values for various decoding priors. From left to right: True block prior, log odds ratio of the Bayes-optimal prior, Bayes-optimal prior on a narrower time window (−400 ms to −100 ms), and the Bayes-optimal prior from main Fig. 2b. A region is deemed significant if the Fisher combined $p$-value is lower than 0.05. **b.** Correlation analysis comparing Bayes optimal decoding from the extended window (shown in Fig. 2b) with the true block decoding (left panel), the log odds prior (middle panel), and the Bayes-optimal prior from the narrower window (right panel). In the three cases, we have a large correlation between

corrected $R^2$. **c.** Comparison of the corrected $R^2$ across the four decodings, testing whether the points panel **b.** are over or below the diagonal (2-tailed signed-rank paired Wilcoxon test, *n.s.* not significant, ***p < 0.001). **d.** Bayesian model comparison for 2 behavioural models, the Bayes optimal model, which infers a prior from past observations (see Methods and Supplementary Information), and a model that assumes the true block prior, which is not accessible to the mice. Our analysis shows that the Bayes optimal model more effectively explains the behaviour, with an exceedance probability greater than 0.999.

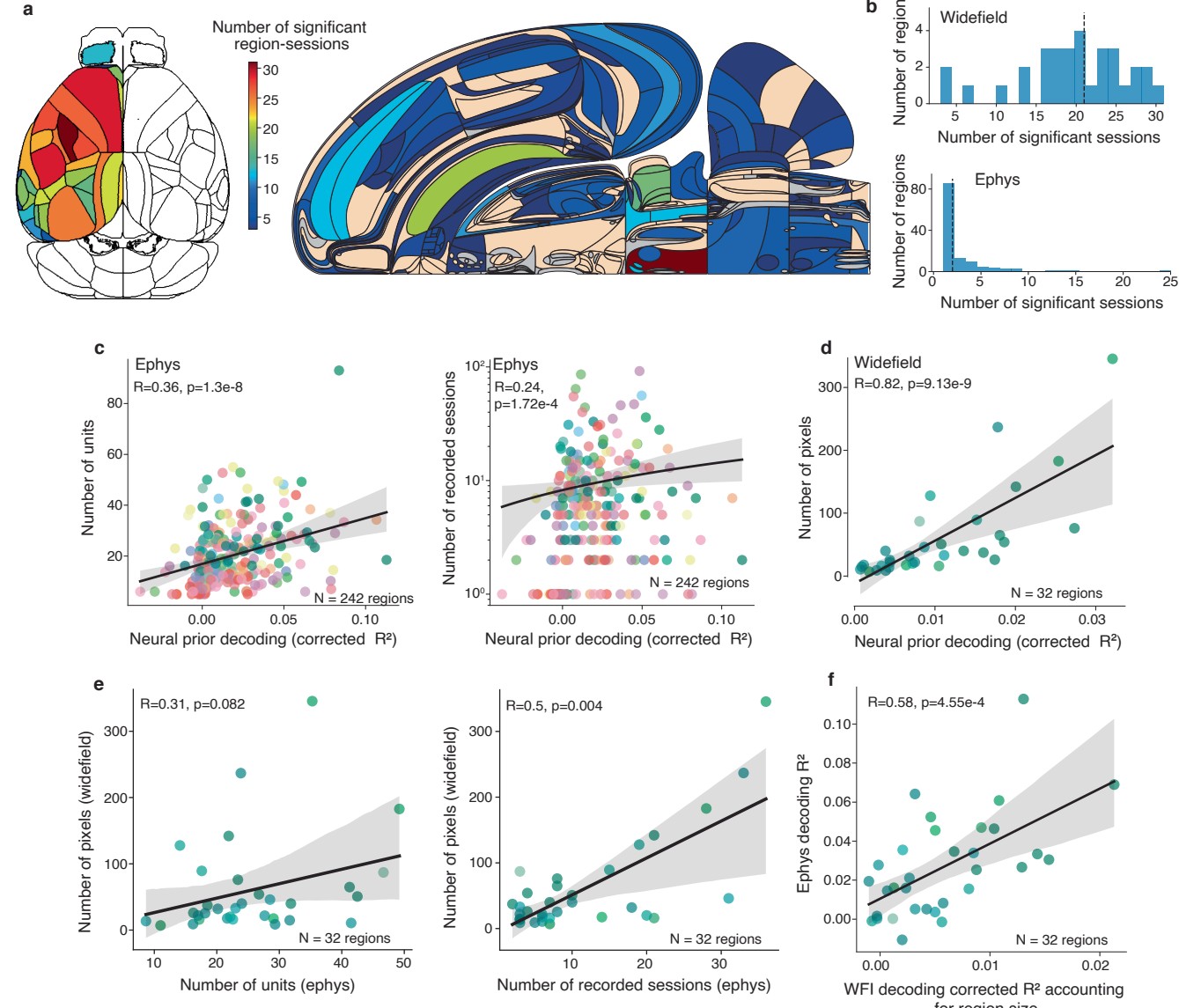

**Extended Data Fig. 7 | a.** Number of significant sessions for each region represented on the dorsal map for WFI and the Swanson map for Ephys. **b**. Histograms representing the distribution of the number of significant sessions for each region in both WFI (top) and Ephys (bottom). Note that the number of significant sessions per region in Ephys is low, which prevents us from making robust claims at the regional level. **c**. Number of units (left) and number of recorded sessions (right) as a function of the decoded $R^2$ for the Bayes-optimal prior in Ephys. **d**. Number of pixels as a function of the decoded

$R^2$ for the Bayes-optimal prior in WFI. **e**. Correlations between confounds across modalities. Left panel: Number of pixels in WFI as a function of the number of units in Ephys. Right: Number of pixels as a function of the number of recorded sessions in Ephys. **f**. Corrected $R^2$ for Ephys as a function of the corrected $R^2$ for WFI after correcting the WFI $R^2$ data for region size. Correcting for the region size in WFI was performed by subtracting the size-predicted $R^2$ (from panel **d**) from the WFI $R^2$. Each dot corresponds to one region. All Ephys regions (significant and non-significant) were included in this analysis.

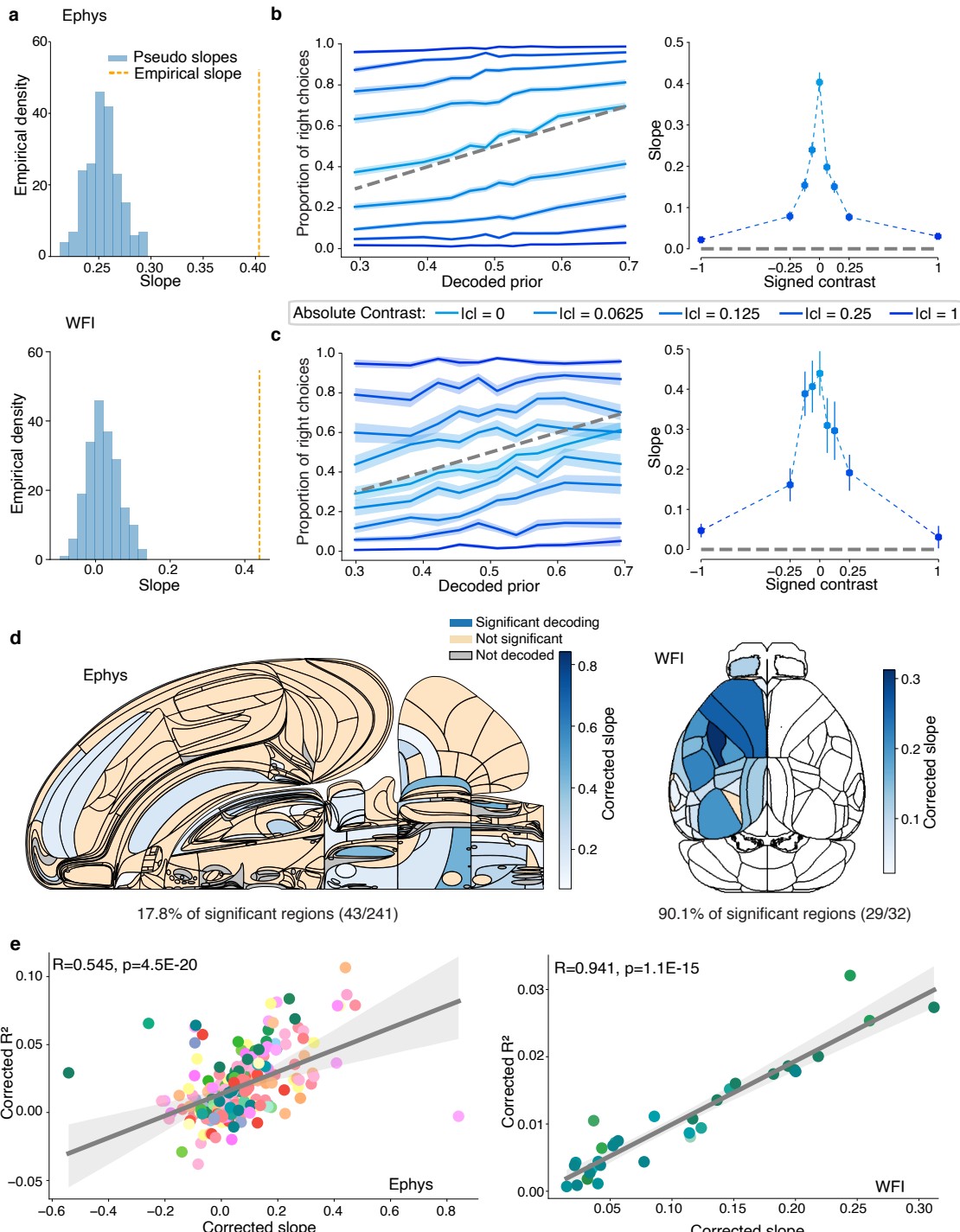

**Extended Data Fig. 8** | See next page for caption.

**Extended Data Fig. 8 | a.** Null distribution of the slopes for the proportion of right choice vs decoded prior on zero contrast trials. Slopes were estimated using logistic regression to predict the choice (left or right) as a function of the decoded prior. The null distribution was calculated using 200 pseudosessions. For each pseudosession, pseudoactions were generated from an action kernel behavioural model that was fit to each real session (see Methods for more details). We then obtained pseudoslopes by predicting (with logistic regression) the pseudoactions as a function of the decoded prior. The null distribution was obtained by averaging the pseudoslopes across all sessions (we thus obtain 200 averaged pseudoslopes). The empirical average slope (yellow dashed lines) does not overlap with the null distribution obtained with pseudosessions (blue histogram). Therefore the correlations between the predicted prior and proportion of right choice can not be explained away by spurious temporal correlations or drift in the neural recordings. top: ephys, bottom: WFI. **b.** Left: Proportion of right choices vs. cross-validated decoded Bayes-optimal prior from neural activity for all contrast strengths in Ephys. Different shades of blue denote different contrast strengths. Main Fig. 2e focused on the zero-contrast case; here we show the same analysis across all contrasts. Right: Slopes, estimated using logistic regression to predict choice from decoded prior (as in panel a - see Methods). Slopes are strongly modulated by contrast strengths, arguing against a mere perseverative motor bias, which would produce a slope that is invariant across contrasts **c.** Same as b. but in WFI. **d.** Proportion of right choices on zero contrast trials as a function of the decoded region-level Bayes-optimal prior. We decoded the Bayes-optimal prior for each region and computed the slope of this decoded prior as a function of the proportion of right choices (corrected using pseudo-sessions). This is the analysis presented in main Fig. 2e but at a region level (significance is assessed when the region-level p-values < 0.05, using Fisher's method for combining p-values). We observed that the slopes are significant in 17.8% of the regions in Ephys and 90.1% in Widefield, spanning every level of the hierarchy, including LGd, SCm, CP, MOs, and ACAd. It should be noted that the analysis for Ephys includes only 241 regions due to the exclusion of two sessions where the mouse made the same choice on every zero contrast trial. **e.** Correlation at the regional level between the decoded $R^2$ values and the corrected slopes. We find correlations in both modalities. These correlations prompt further investigation into whether they could be explained away by differences in how the Bayes optimal prior versus the action kernel model account for behaviour across sessions. Specifically, sessions that more closely follow the action kernel model could potentially show lower corrected $R^2$ and slopes, as these metrics are calculated using the Bayes optimal prior. In Ephys, we found no correlation between the log Bayes Factor (the difference in the marginal log likelihood between the action kernel and Bayes optimal models at the session level) and the corrected slopes (Spearman correlation: R = 0.05, P = 0.29, N = 412 sessions), with the corrected slopes averaged across regions for each session. In widefield, a small correlation was detected (Spearman correlation: R = −0.34, P = 0.014, N = 51 sessions). However, even after adjusting for the log Bayes factor (by removing the linear prediction of the log Bayes factor from the corrected slope), the correlation between the corrected $R^2$ and the adjusted corrected slope remained strong (Spearman correlation: R = 0.935, P = 4.7 × 10$^{-15}$, N = 32 regions). This suggests that the type of behavioural strategy the mice used does not confound the correlation between the corrected $R^2$ and the corrected slope.

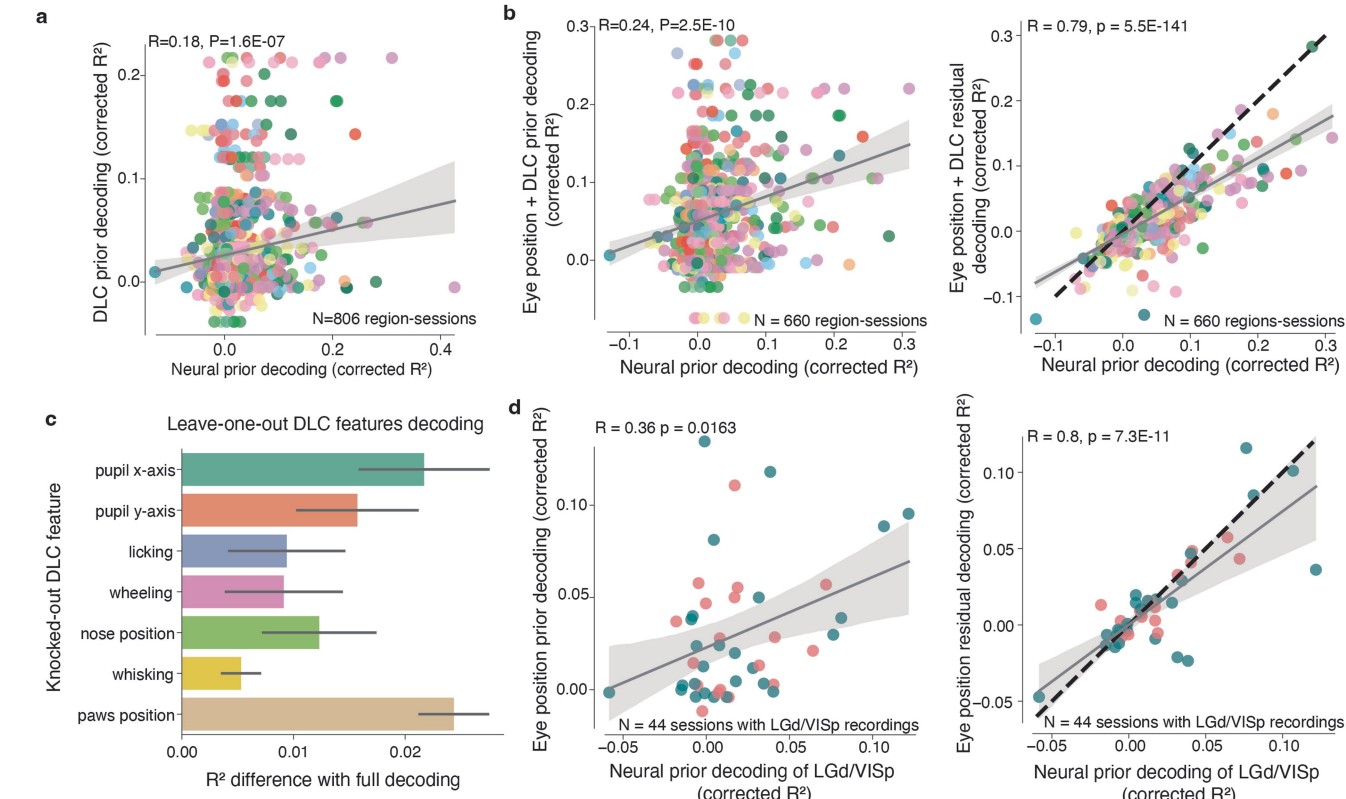

**Extended Data Fig. 9 | a.** The decoding $R^2$ for the Bayes-optimal prior from neural activity is significantly correlated with the decoding $R^2$ for the Bayes-optimal prior from DLC features (Pearson correlation R = 0.18, P = 1.6 × 10⁻⁷). **b**. Embodiment analysis accounting for both the DLC features and the eye position. Left: Decoding $R^2$ for the Bayes-optimal prior from neural activity against decoding $R^2$ for the Bayes-optimal prior from DLC features and eye position. The correlation between these two quantities is significant (Pearson correlation $R = 0.24$, $P = 2.5 × 10^{-10}$). Right: DLC + eye position residual decoding $R^2$ against neural decoding $R^2$. The residual decoding $R^2$ values are obtained by first regressing the Bayes-optimal prior from DLC features and eye position, and then regressing the prior residual (Bayes-optimal prior minus Bayes-optimal prior estimated from DLC features and eye position) against neural activity. The neural decoding $R^2$ corresponds to the $R^2$ when decoding the Bayes-optimal prior from neural activity. The two quantities are strongly correlated (Pearson correlation $R = 0.79$, $P = 5.5 × 10^{-141}$), suggesting that the prior cannot be entirely attributed to a combination of both DLC features and eye position. **c**. Regressor elimination approach: for each feature, we remove it to measure the decrease in the decoding score compared to the full model (see Methods). The first feature to impact the model significantly when removed is the paw position. In this

task, the paws are typically engaged to manipulate the wheel, which in turn adjusts the stimulus. It appears that the paws are positioned differently—likely on the wheel—depending on whether the prior suggests the next side will be left or right. The second key feature was the x-coordinate of the eye position, which aligns with the task setup where the stimulus is positioned along a horizontal plane, indicating that the mice tend to look in the direction suggested by the Bayes-optimal prior. **d**. Left: decoding $R^2$ for the Bayes-optimal prior from neural activity in VISp and LGd against decoding $R^2$ for the Bayes-optimal prior from eye position. The correlation between these two quantities is significant (Pearson correlation $R = 0.36$, $P = 0.0163$). Right: residual decoding $R^2$ against neural decoding $R^2$. The residual decoding $R^2$ values are obtained by first regressing the Bayes-optimal prior against eye position and then regressing the prior residual (Bayes-optimal prior minus Bayes-optimal prior estimated from eye position) against neural activity in VISp and LGd (see Methods). The neural decoding $R^2$ corresponds to the $R^2$ when decoding the Bayes-optimal prior from neural activity. The two quantities are strongly correlated (Pearson correlation $R = 0.8$, $P = 7.3 × 10^{-11}$), suggesting that the prior signals in LGd and VISp are not solely due to the position of the eyes across blocks.

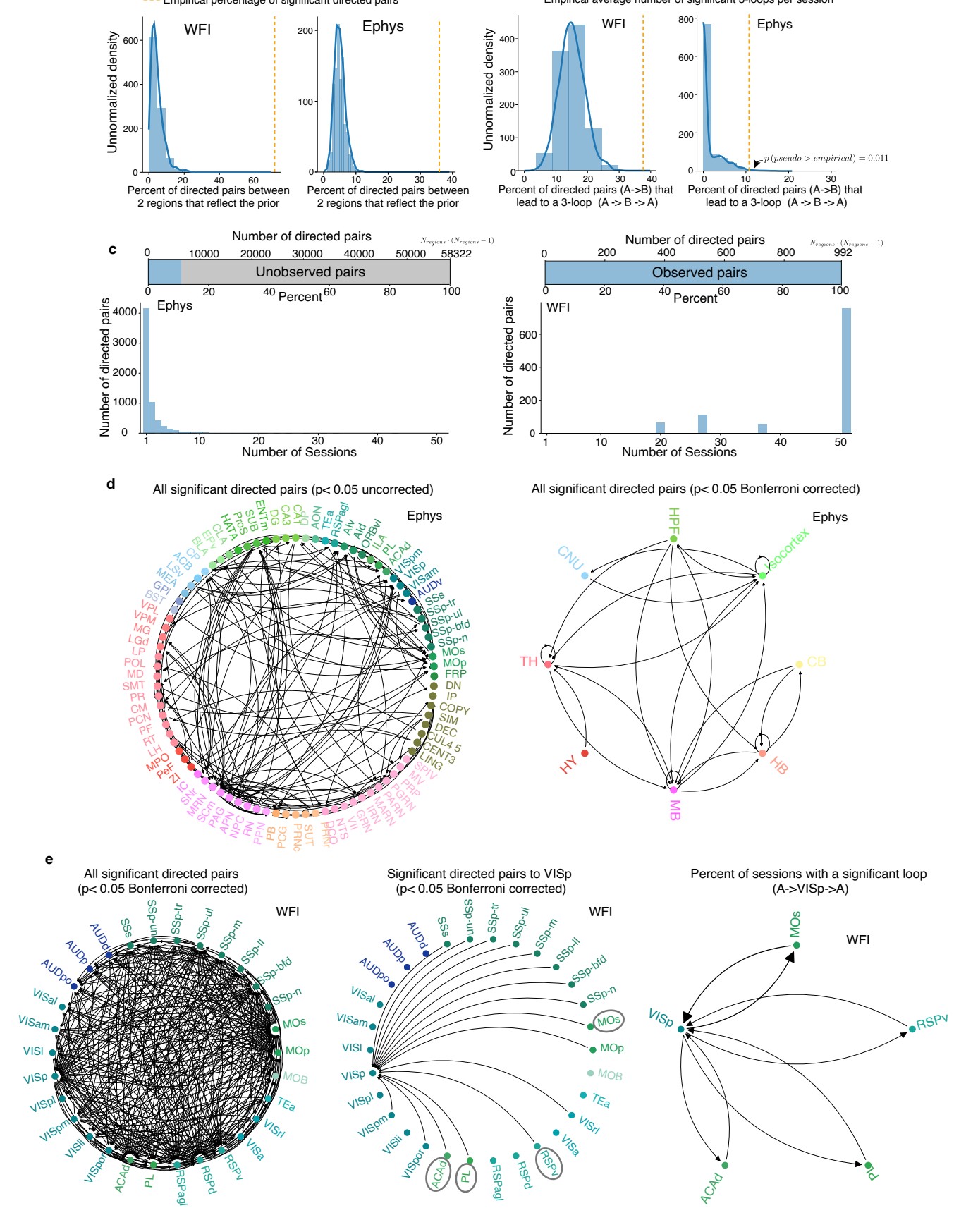

**Extended Data Fig. 10** | See next page for caption.

**Extended Data Fig. 10 | Granger causality analysis. a**. Average percentage of significant directed pairs between two regions that reflect the prior across sessions. When considering all pairs of regions encoding the prior significantly, and for which we had simultaneous recordings, we observed that information was significantly exchanged between 71% of these pairs in Widefield imaging and 36% in Ephys. Blue histograms: null distribution. **b**. Average percentage of significant directed pairs (A- > B) which are reciprocated within the same session by their counterparts (B- > A); we found this to occur 38% of the time in Widefield and 11% in Ephys. Blue histograms: null distribution. (same as in main Fig. 2h). **c**. Histogram showing the number of sessions for each directed pair and barplot showing the percentage of observed directed pairs (directed pairs with at least one session) versus unobserved pairs. Right: In Ephys, with a total of 242 observed regions, the possible number of pairs amounts to 242 × 241 = 58,322. Of these, approximately 10% of the directed pairs had been recorded simultaneously, but the vast majority (75%) of these pairs appeared in two or fewer sessions, highlighting their scarcity. Left: Widefield provides a richer dataset, because, with 32 regions recorded simultaneously, we can analyse a total of 992 possible directed pairs (32 × 31), most of them available on all sessions. **d**. Left**:** Complete connectivity graph from Ephys (p < 0.05 uncorrected for multiple comparisons). When correcting for multiple comparisons, none of the links remains significant. This lack of significant findings post-correction is likely due to the sparse nature of the observations in Ephys (see panel c.). Right: Connectivity graph in Ephys across Cosmos regions (p < 0.05 Bonferroni corrected). *p*-values across directed pairs of regions are aggregated at the Cosmos level with Fisher's method (see Methods, identical to main Fig. 2g left). **e**. Left: Complete connectivity graph from Widefield (p < 0.05 Bonferroni corrected). The graph is densely populated and consequently difficult to interpret. Middle: A partial connectivity graph from Widefield, highlighting significant directed pairs projecting to the Primary Visual Cortex (VISp), as shown in Fig. 2g (right). We uncover feedback connections from higher-order areas such as the Motor Cortex (MOs), Ventral Retrosplenial Cortex (RSPv), Prelimbic Cortex (PL), and Anterior Cingulate Area Dorsal (ACAd) – these regions are marked with grey circles for emphasis – to the early sensory area, the Primary Visual Cortex (VISp). Left: Percentage of sessions exhibiting significant reciprocal connections (A->VISp->A) for sessions in which the Bayes optimal prior could be significantly decoded from both VISp and the previously identified higher-order regions (MOs, RSPv, PL and ACAd). The size of the arrow is proportional to the percentage. Our findings indicate the existence of reciprocal connections in these sessions: 33.3% between MOs and VISp, 16.7% between ACAd and VISp, 20% between PL and VISp, and 18.75% between RSPv and VISp.

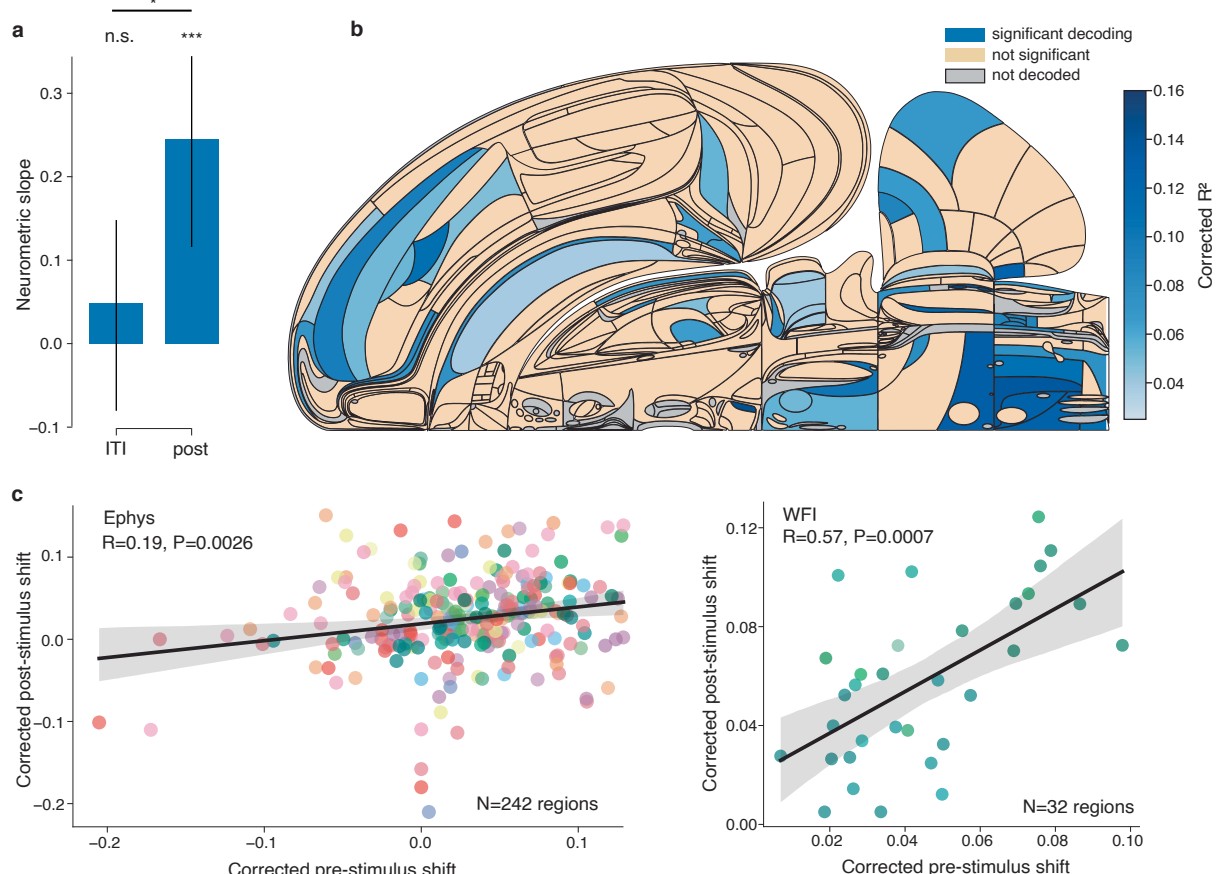

**Extended Data Fig. 11 | a.** The average slope of the neurometric curves is significantly different from 0 during the post stimulus period (2-tailed signed-rank Wilcoxon test, t = 9833, $P = 1.2 \times 10^{-5}$, $N = 242$ regions) but not during the ITI (t = 13547, $P = 0.29$, $N = 242$ regions). Also, neurometric slopes are significantly greater during the post-stimulus period than during the ITI (2-tailed signed-rank paired Wilcoxon test t = 12306, $P = 0.028$, $N = 242$ regions) (***$p < 0.001$, *$p < 0.05$, *n.s.* not significant). **b.** Swanson map of corrected neurometric post-stimulus shifts for Ephys data. **c.** The corrected post-stimulus shifts and corrected ITI shifts are significantly correlated in both Ephys (Spearman correlation $R = 0.19$, $P = 0.0026$, $N = 242$ regions) and WFI (Spearman correlation $R = 0.57$, $P = 0.0007$, $N = 32$ regions).

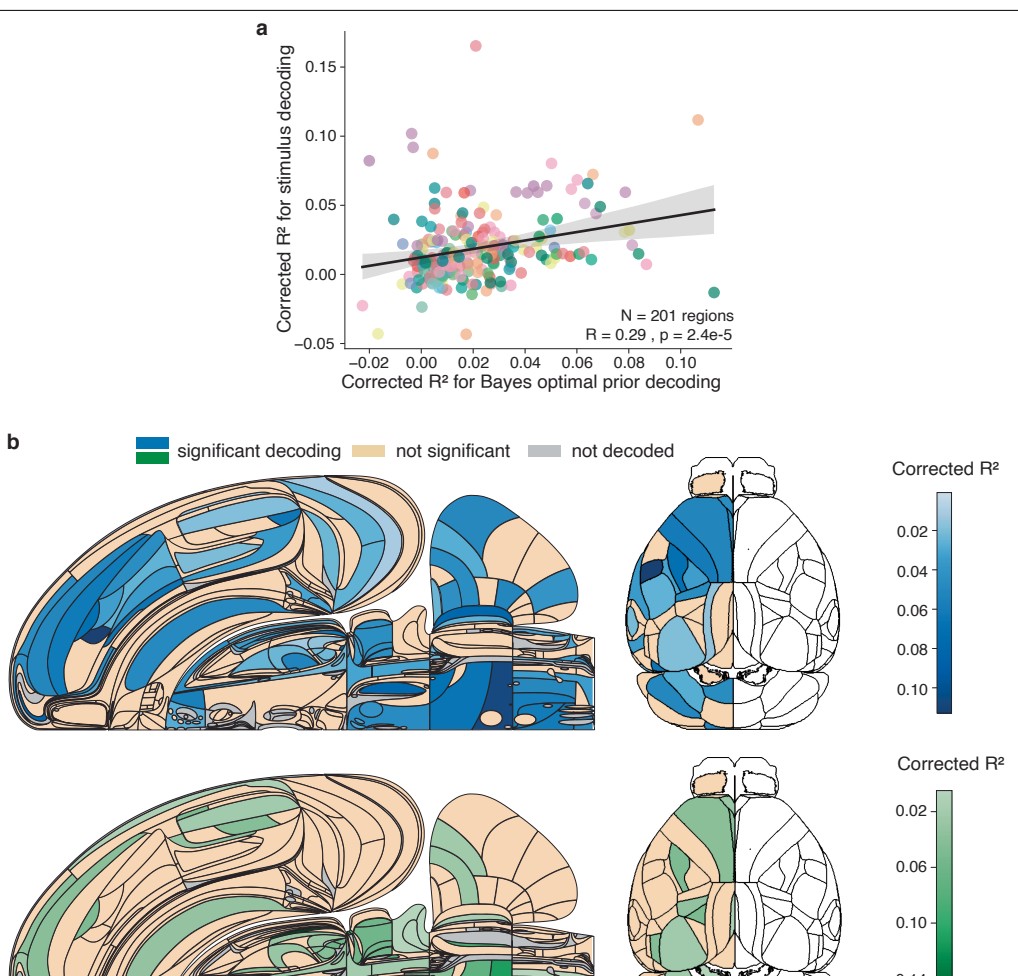

**Extended Data Fig. 12 | a.** The neural decoding $R^2$ for the stimulus side and the Bayes-optimal prior are significantly correlated across brain regions (Spearman correlation $R = 0.29$, $P = 2.4 \times 10^{-5}$). **b.** Swanson maps and dorsal cortical views of brain regions encoding the Bayes-optimal prior (blue, upper) and the stimulus side (green, lower) significantly based on Ephys data.

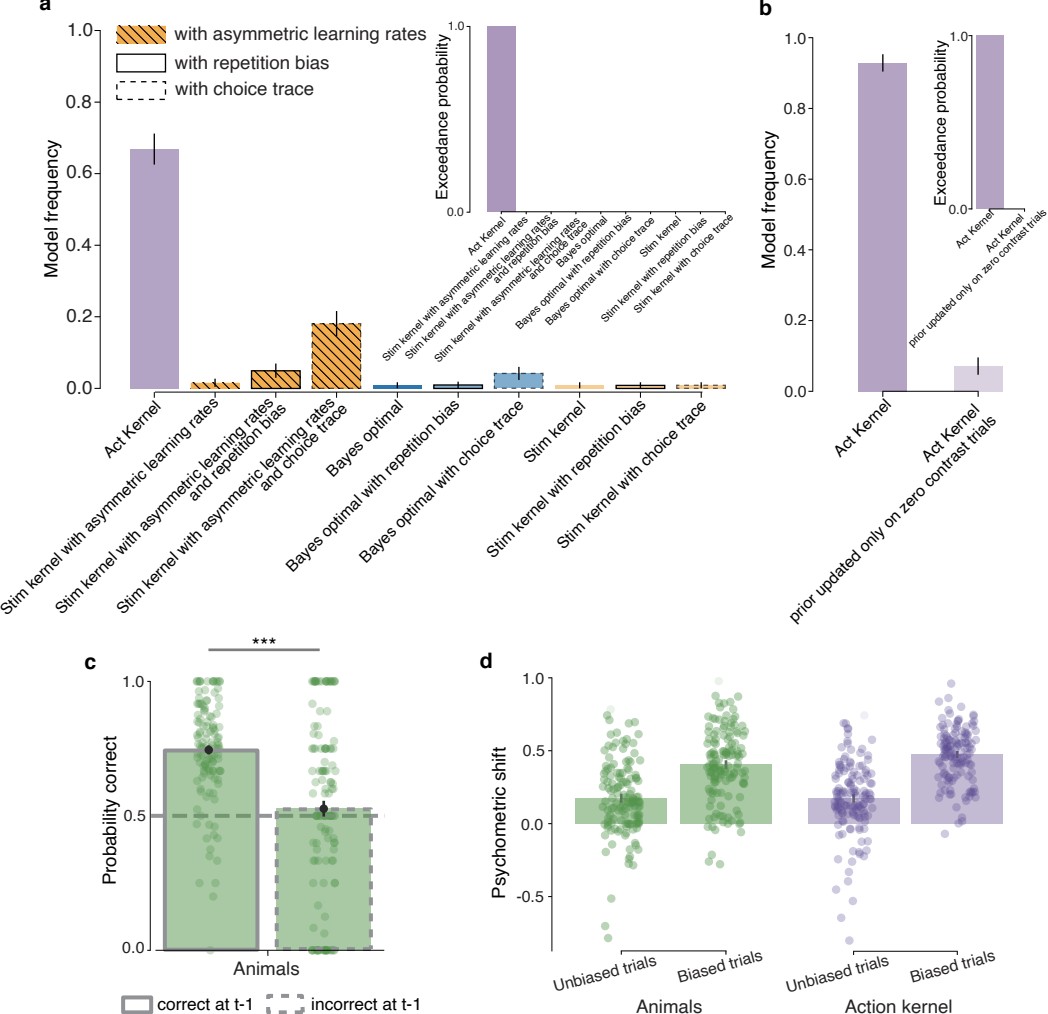

**Extended Data Fig. 13 | a.** Bayesian model comparison for 11 behavioural models, considering the possibility of one step repetition bias (i.e. a tendency to repeat the previous choice), multi-step repetition bias (i.e. a tendency to follow an exponentially decaying average of past choices), and, for the stimulus kernel model, the presence of positivity and confirmation biases as asymmetric learning rates (accounting for the possibility to learn differently from positive versus negative rewards and from information that confirms versus contradicts existing beliefs[40]. See Methods for more details on the Bayes-optimal, action kernel and stimulus kernel models and Supplementary information for the formal equations of the repetition bias and asymmetrical learning rates. Model frequency (the posterior probability of the model given the subjects' data, left panel) and exceedance probability (the probability that a model is more likely than any other models, right panel) are shown. The action kernel model offered the best account of the data even when including models with repetition, positivity and confirmation biases ($p_{exceedance} > 0.999$). **b.** Bayesian model comparison for two behavioural models, the action kernel and a variant that operates only during 0% contrast trials (by calculating an exponentially decaying average of chosen actions at 0% contrast trials). Our comparisons indicate that the action kernel, updating across all contrasts, more effectively explains behaviour (exceedance probability > 0.999), suggesting that mice do not limit their subjective prior estimations to 0% contrast trials alone. **c.** Performance on zero contrast trials, distinguishing whether the preceding action was correct or incorrect and considering that the previous contrast was non zero. This analysis mirrors the main analysis in Fig. 4c but is specifically restricted to previous trials with non-zero contrast. When considering behaviour within blocks, an agent using an action kernel prior should show a higher percentage

of correct responses following a correct, block-consistent action compared to an incorrect one. This is because, on incorrect trials, the prior is updated with an action corresponding to the incorrect stimulus side. Even when limited to previous trials with non-zero contrast, there is a notable difference in the probability of making a correct decision following an incorrect vs. a correct choice (Wilcoxon paired test, t = 11734, P = 1.1 × 10⁻¹⁵). This finding is confirmation that mice update their priors using information from all contrast levels, not solely zero contrast trials. **d.** Psychometric shift during both the first 90 trials (unbiased) and the other trials (biased) for animals and the action kernel. This shift is determined by analysing two psychometric curves, one conditioned on the action kernel prior being above 0.5 (favoring the right side) and the other conditioned on the action kernel prior being less than 0.5 (favoring the left side). We fit psychometric functions to these curves, and then calculate the psychometric shift as the vertical displacement of these curves at zero contrast. As predicted by the action kernel model, the analysis reveals a significant positive psychometric shift during the unbiased phase (first 90 trials). Furthermore, the shift in the behavioural data is less pronounced during the unbiased period compared to the biased period because the stimuli are more balanced in the unbiased phase, keeping the subjective prior closer to 0.5. Specifically, when distinguishing the trials that favour the right side (action kernel prior above 0.5) from those favoring the left side (action kernel prior below 0.5), the underlying action kernel priors remained close to 0.5 during the unbiased period. However, the presence of significant and comparable shifts between the animals and the action kernel model during the unbiased period indicates that mice exhibit a behavioural shift during the unbiased trials.

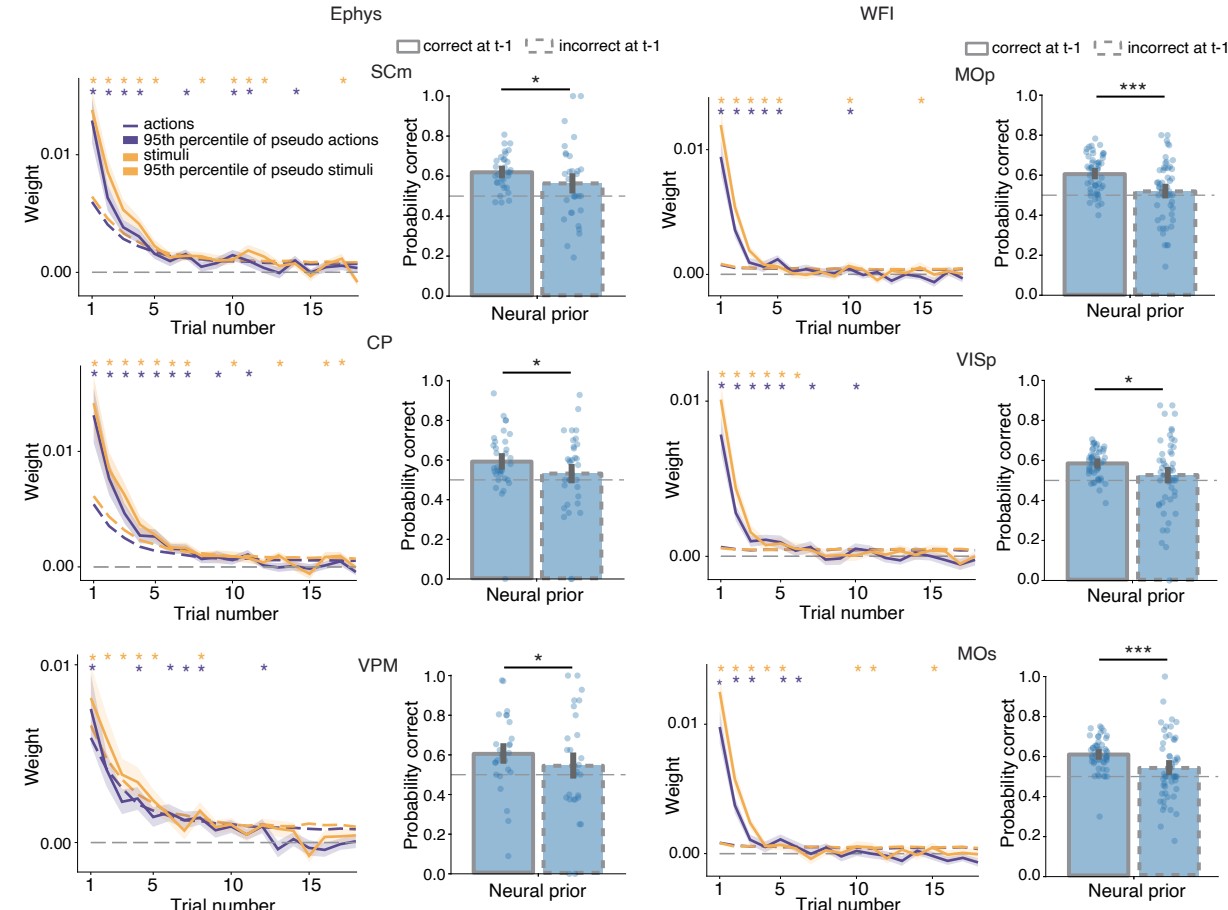

**Extended Data Fig. 14 | Same analysis as in Fig. 4c,e, but for three specific brain regions using Ephys (SCm, CP, VPM) or WFI data (right column, MOp, VISp, MOs) (\*p < 0.05, \*\*p < 0.01, \*\*\*p < 0.001).** For the influence of past actions on the decoded Bayes-optimal prior, significance is assessed in the same way as in the main Fig. 4e (see Methods). For the asymmetry effect, the effect being observed on a brain-wide level, we performed a 1-tailed signed-rank Wilcoxon paired test for assessing significance on the region level.

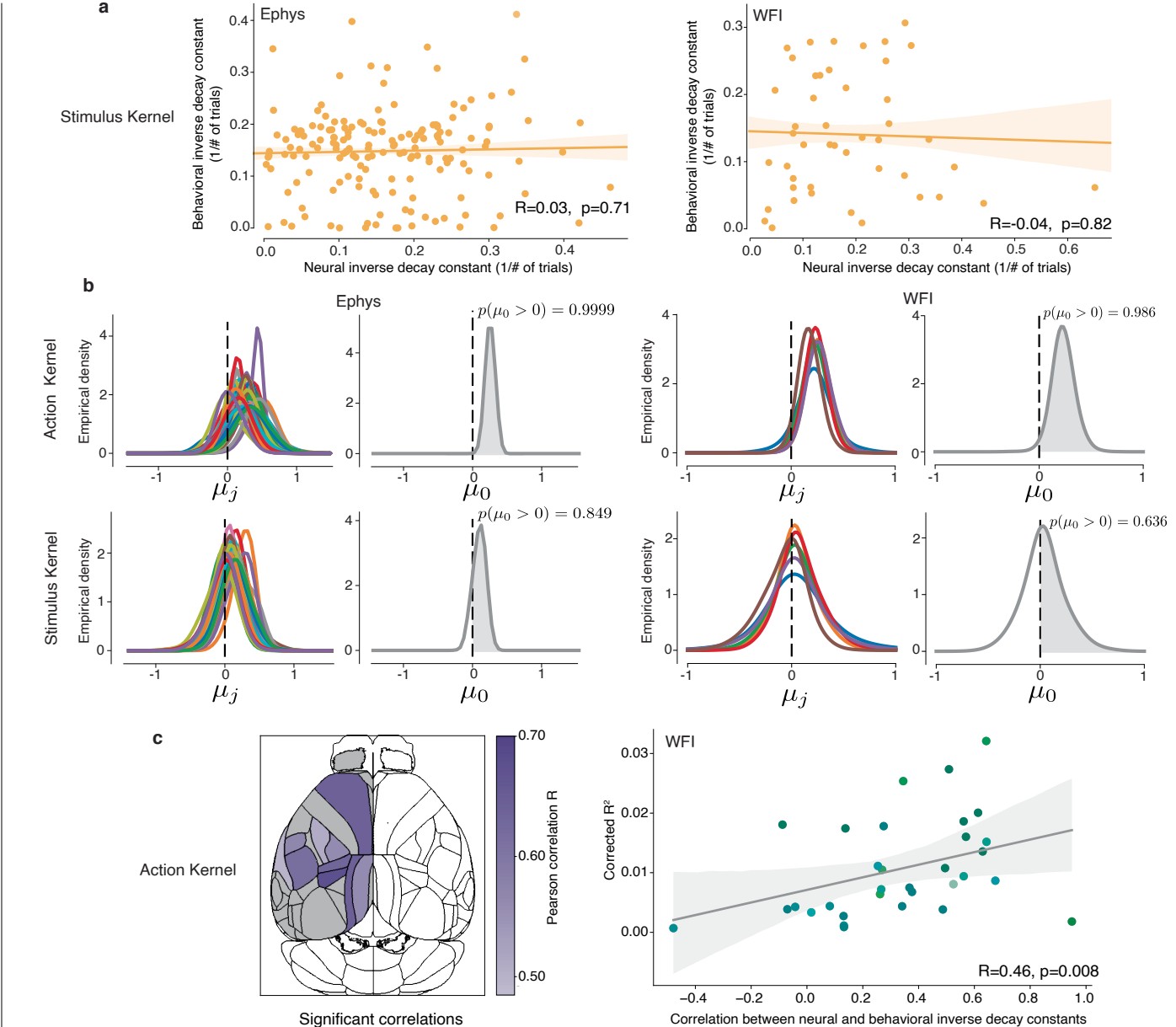

**Extended Data Fig. 15** | See next page for caption.

**Extended Data Fig. 15 | a.** Behavioural inverse decay constants, obtained by fitting the stimulus kernel model to the behaviour, as a function of the neural inverse decay constants, obtained by estimating the temporal dependency of the neural signals with respect to previous stimuli (see Methods). The neural and behavioural inverse decay constants are not significantly correlated for either Ephys (Pearson correlation $R = 0.03$, $P = 0.71$) or WFI (Pearson correlation $R = -0.04$, $p = 0.82$). **b.** Hierarchical modelling of the neural and behavioural inverse decay constants (also referred to here as learning rates). The parameter $\mu_j$, defined for each mouse $j$, is the slope (the multiplicative coefficient) of the linear regression predicting the neural learning rate from the behavioural learning rate (on the sessions of mouse j). These parameters $\mu_j$ are sampled from a common population level prior with mean $\mu_0$. The parameter $\mu_0$, defined at the population level, characterizes an overall relationship between neural and behavioural learning rates. We found that the relationship between neural and behavioural learning rates is significantly positive for the action kernel model (top row), both in electrophysiology (left column) and in widefield imaging (right column), which is not the case for the stimulus Kernel model (bottom row). Furthermore, when testing the difference in means of the population level parameter $\mu_0$ between action and stimulus kernels, we found that it was significantly greater for the action kernel, both in Ephys and in WFI. Significance was assessed by estimating the means of the $\mu_0$ distributions for the action and stimulus kernels with the BEST Bayesian test[63]. In both Ephys and WFI, we found that $p(\overline{\mu_0^{actKernel}} > \overline{\mu_0^{stimKernel}}) = 1$ with $\overline{\mu_0^{actKernel}}$ and $\overline{\mu_0^{stimKernel}}$ the means of the $\mu_0$ distributions for the action and stimulus kernels, respectively.

Regarding the effect sizes, with the same BEST procedure, we find an effect size of 2.53 in Ephys and 1.96 in widefield (effect sizes greater than 1.3 are commonly considered to be very large[64]). See Supplementary Information for the full specification of the hierarchical generative model. **c.** Correlation, at a region level, between neural inverse decay constants (estimating temporal dependency of the neural signals on previous actions), and behavioural inverse decay constants (from fitting the action kernel to behaviour). A decay constant is estimated for each pixel (as in Fig. 4f, refer to Methods), but now, averages are taken across pixels for each session and specific region. In the analysis Fig. 4f, session-level learning rates were obtained by averaging across all pixels, regardless of region identity. Left: Regions with a significant correlation between behavioural and neural inverse decay constants. As expected, only positive correlations emerge as significant. Right: Correlation between behavioural and neural inverse decay constants is correlated with the prior decoding corrected $R^2$ from the same regions. These two quantities were found to be also correlated (R = 0.46, P = 0.008). In other words, regions in which the prior decoding $R^2$ is large are also regions which best reflect the behavioural decay constant, i.e., these are the regions that are best correlated with the animals' cognitive strategies as assessed by the lengths of the action kernels. We did not repeat this analysis with the electrophysiology recordings because we only have a very limited number of significant sessions per region (1-2 for most regions, as opposed to around 20 sessions per region for the WFI data - see Extended Data Fig. 7a,b).

# Reporting Summary

## Statistics

For all statistical analyses, confirm that the following items are present in the figure legend, table legend, main text, or Methods section.

| n/a | Confirmed | |
|---|---|---|
| ☐ | ☒ | The exact sample size (*n*) for each experimental group/condition, given as a discrete number and unit of measurement |
| ☐ | ☒ | A statement on whether measurements were taken from distinct samples or whether the same sample was measured repeatedly |
| ☐ | ☒ | The statistical test(s) used AND whether they are one- or two-sided *Only common tests should be described solely by name; describe more complex techniques in the Methods section.* |
| ☐ | ☒ | A description of all covariates tested |
| ☐ | ☒ | A description of any assumptions or corrections, such as tests of normality and adjustment for multiple comparisons |
| ☐ | ☒ | A full description of the statistical parameters including central tendency (e.g. means) or other basic estimates (e.g. regression coefficient) AND variation (e.g. standard deviation) or associated estimates of uncertainty (e.g. confidence intervals) |
| ☐ | ☒ | For null hypothesis testing, the test statistic (e.g. *F*, *t*, *r*) with confidence intervals, effect sizes, degrees of freedom and *P* value noted *Give P values as exact values whenever suitable.* |
| ☐ | ☒ | For Bayesian analysis, information on the choice of priors and Markov chain Monte Carlo settings |
| ☐ | ☒ | For hierarchical and complex designs, identification of the appropriate level for tests and full reporting of outcomes |
| ☐ | ☒ | Estimates of effect sizes (e.g. Cohen's *d*, Pearson's *r*), indicating how they were calculated |

*Our web collection on statistics for biologists contains articles on many of the points above.*

## Software and code

Policy information about availability of computer code

| | |
|---|---|
| Data collection | please see https://int-brain-lab.github.io/iblenv/notebooks_external/data_structure.html |
| Data analysis | all our code is available at https://github.com/int-brain-lab/prior-localization<br>Requirements and versions are:<br>scikit-learn==1.5.1<br>psychofit @ git+https://github.com/cortex-lab/psychofit.git<br>behavior_models @ git+https://github.com/int-brain-lab/behavior_models.git<br>brainwidemap @ git+https://github.com/int-brain-lab/paper-brain-wide-map.git |

For manuscripts utilizing custom algorithms or software that are central to the research but not yet described in published literature, software must be made available to editors and reviewers. We strongly encourage code deposition in a community repository (e.g. GitHub). See the Nature Portfolio guidelines for submitting code & software for further information.

## Data

Policy information about <u>availability of data</u>

All manuscripts must include a <u>data availability statement</u>. This statement should provide the following information, where applicable:

- Accession codes, unique identifiers, or web links for publicly available datasets
- A description of any restrictions on data availability
- For clinical datasets or third party data, please ensure that the statement adheres to our <u>policy</u>

The electrophysiology data for this paper are available via http://viz.internationalbrainlab.org and https://int-brain-lab.github.io/iblenv/notebooks_external/data_release_brainwidemap.html

The widefield and pupil tracking data are also available through the public IBL database (https://openalyx.internationalbrainlab.org) and can be accessed via the ONE API using the tag "2023_Q3_Findling_Hubert_et_al" as described here https://int-brain-lab.github.io/ONE/notebooks/one_search/one_search.html#Searching-data-with-a-release-tag.

The Swanson flat map can be found at https://int-brain-lab.github.io/iblenv/notebooks_external/atlas_swanson_flatmap.html

## Research involving human participants, their data, or biological material

Policy information about studies with <u>human participants or human data</u>. See also policy information about <u>sex, gender (identity/presentation), and sexual orientation</u> and <u>race, ethnicity and racism</u>.

| | |
|---|---|
| Reporting on sex and gender | N/A |
| Reporting on race, ethnicity, or other socially relevant groupings | N/A |
| Population characteristics | N/A |
| Recruitment | N/A |
| Ethics oversight | N/A |

Note that full information on the approval of the study protocol must also be provided in the manuscript.

# Field-specific reporting

Please select the one below that is the best fit for your research. If you are not sure, read the appropriate sections before making your selection.

☒ Life sciences ☐ Behavioural & social sciences ☐ Ecological, evolutionary & environmental sciences

For a reference copy of the document with all sections, see nature.com/documents/nr-reporting-summary-flat.pdf

# Life sciences study design

All studies must disclose on these points even when the disclosure is negative.

| | |
|---|---|
| Sample size | No statistical methods were used to predetermine sample sizes. For electrophysiology, data were collected from 699 Neuropixels probe insertions across 459 sessions in 139 mice, with 414 sessions meeting inclusion criteria. For widefield calcium imaging, 51 sessions from 6 mice were included. Regions were analyzed if at least 5 well-isolated units (Ephys) or pixels (WFI) passed quality control. These sample sizes ensured sufficient statistical power for decoding analyses and robust brain-wide coverage, as supported by estimates indicating ~10 recordings per region are sufficient (see fig S5e). |
| Data exclusions | Pre-established exclusion criteria were applied to ensure data quality. Trials were excluded if mice did not respond, or if reaction times were <80ms or >2s. Sessions with <250 included trials were excluded (41 Ephys, 1 WFI). For electrophysiology, only neurons passing strict quality control (amplitude >50μV, noise cut-off <20, and no refractory period violations) were included. Regions required ≥5 QC-passed units (Ephys) or pixels (WFI) to be analyzed. These criteria were defined prior to analysis and are detailed in the Methods. |
| Replication | The main behavioral and neural analyses were replicated across two independent recording modalities—Neuropixels electrophysiology (699 insertions across 459 sessions in 139 mice) and widefield calcium imaging (51 sessions in 6 mice). Behavioral effects and neural decoding of the prior were consistent across both modalities. |
| Randomization | Randomization into experimental groups was not applicable, as all mice were trained on the same task and underwent the same recording procedures. To assess significance in decoding neural representations, we employed a pseudo-session resampling procedure to construct null distributions, as detailed in the Methods. |
| Blinding | Blinding was not performed because all mice were trained using identical protocols and recorded using standardized procedures. There were no experimental groups or treatment conditions to blind against. |

# Reporting for specific materials, systems and methods

We require information from authors about some types of materials, experimental systems and methods used in many studies. Here, indicate whether each material, system or method listed is relevant to your study. If you are not sure if a list item applies to your research, read the appropriate section before selecting a response.

## Materials & experimental systems

| n/a | Involved in the study |
|-----|----------------------|
| ☒ | ☐ Antibodies |
| ☒ | ☐ Eukaryotic cell lines |
| ☒ | ☐ Palaeontology and archaeology |
| ☐ | ☒ Animals and other organisms |
| ☒ | ☐ Clinical data |
| ☒ | ☐ Dual use research of concern |
| ☒ | ☐ Plants |

## Methods

| n/a | Involved in the study |
|-----|----------------------|
| ☒ | ☐ ChIP-seq |
| ☒ | ☐ Flow cytometry |
| ☒ | ☐ MRI-based neuroimaging |

## Animals and other research organisms

Policy information about studies involving animals; ARRIVE guidelines recommended for reporting animal research, and Sex and Gender in Research

| | |
|---|---|
| Laboratory animals | we used C57BL/6 laboratory mice |
| Wild animals | No wild animals were used in this study |
| Reporting on sex | both sexes were used; and are reported |
| Field-collected samples | No field-collected samples were used in this study |
| Ethics oversight | All experimental procedures involving animals were conducted in accordance with local laws and approved by the relevant institutional ethics committees. Approvals were granted by the Animal Welfare Ethical Review Body of University College London, under licences P1DB285D8, PCC4A4ECE, and PD867676F, issued by the UK Home Office. Experiments conducted at Princeton University were approved under licence 1876-20 by the Institutional Animal Care and Use Committee (IACUC). At Cold Spring Harbor Laboratory, approvals were granted under licences 1411117 and 19.5 by the IACUC. The University of California at Los Angeles granted approval through IACUC licence 2020-121-TR-001. Additional approvals were obtained from the University Animal Welfare Committee of New York University (licence 18-1502); the IACUC at the University of Washington (licence 4461-01); the IACUC at the University of California, Berkeley (licence AUP-2016-06-8860-1); and the Portuguese Veterinary General Board (DGAV) for experiments conducted at the Champalimaud Foundation (licence 0421/0000/0000/2019). |

Note that full information on the approval of the study protocol must also be provided in the manuscript.

## Plants

| | |
|---|---|
| Seed stocks | N/A |
| Novel plant genotypes | N/A |
| Authentication | N/A |

