## [Peer Review File · Nature]

Brain-wide representations of prior information in mouse decision-making

Corresponding Author: Charles Findling

Version 0:

Reviewer comments:

Referee #1

(Remarks to the Author)

In this pair of companion studies, the authors (members of an international consortium for experimental neuroscience called the International Brain Lab; IBL) present an unprecedented dataset, collected across 11 labs, comprised of 547 high-density neural recordings sampled across the entire mouse brain during a standardized decision-making task. In one study, the dataset (curated and now made publicly available) and methodology are described, and then used to perform a battery of correlational analyses across the brain to identify encoding of simple task variables during decision-making (including sensory, choice, outcome, movement, and prior information). Based on this investigation and application of rigorous statistical methods, the authors conclude that different task variables are encoded in different ways (some are represented more widely across the brain, while others are more localized), confirming the results of countless previous reports. In the second study, the authors investigate more deeply the encoding of prior information using a more refined set of analytical tools and combining their electrophysiology dataset with another large calcium imaging dataset collected by IBL participants. The main conclusion of this second study is that, in contrast to the initial conclusion of the first study, which suggested that prior information is encoded in a restricted set of brain regions, subjective priors are actually encoded more widely across cortical, subcortical, and midbrain regions. Further, this more widespread subjective prior is partially embodied in the animal's behavior and is driven by action history (as opposed to sensory and/or outcome history exclusively).

The dataset collected here, and particularly the effort to generate a dataset of such comprehensive scale with rigorous control and standardization of task conditions, preprocessing parameters, etc. is obviously noteworthy and is inarguably a highly unique and important contribution to the field. The presented impetus for this enormous collaborative effort is that our capacity to really gain insight into how the brain processes sensory information, computes decisions, and generates behavioral outputs (i.e. how the brain works) is substantially hindered by the limitations of individual lab groups to survey brain structures comprehensively under consistent conditions, and the idiosyncrasies with which different groups design and implement their experiments that limit our capacity to draw accurate comparisons and generalizations. The assumption underlying and framing the effort is that, with a big enough dataset, collected with careful methodological standardization, we should be able to mine the data to gain novel insight into how the brain works. It is, in some sense, the extreme terminus of "data-driven" approaches.

Somewhat unfortunately, the studies, even taken together, fall woefully short of this laudable goal. In the first study, there are in many cases no explicit hypotheses given to ground the questions the authors ask of the data, or if there are, they are so vague as to be almost useless (e.g. visual information may be encoded to some degree beyond classically defined visual areas). We are presented with what amounts to a brain-wide "screen" of task variable neural correlates, and learn, in the end, essentially nothing new about how the brain works – visual stimuli are predominantly represented in visual cortex and thalamus; movement correlates are widespread across the brain, as are responses to salient rewards and punishment. And, at least in this particular case, this is doubly a problem because the specific framing of the knowledge gap the work is intended to fill is itself defeated – the need for this type of approach totally obviated – by the apparent lack of novel insight. If recording from 300,000 neurons sampled from the entire mouse brain during a well-designed decision-making task does not yield any novel insight, or change our understanding of the brain in any substantial way, then what is the value of such an approach at all?

In the second companion paper, the authors go a bit further. They are able to draw some interesting, useful conclusions from

the work that might have been difficult (but this is not totally clear) to fully capture with more limited, fragmented efforts. In particular, they show that while “objective” prior information is only rarely encoded in restricted brain regions (in the first paper), “subjective” prior information is, in fact, much more robustly encoded, and across several levels of brain structure – and this prior takes the form of a partially-embodied action prior, as opposed to other formulations. In the context of value-based decision, this is indeed an important finding, but in the specific context of these two studies, it is still limited in scope for at least two reasons: First, it somewhat contradicts the conclusions of their first study – the authors make a big deal in study 1 about how the restricted nature of prior encoding is interesting because it stands in contrast to how the other task variables are encoded, but then proceed to explain this difference by arguing that this is probably just because animals do not have access to an objective prior, and in fact, encode subjective priors quite robustly. If this is the case, why not just present the analysis of subjective priors alongside other task variables in the first study? Second, it is not clear to what extent the positive conclusions they are able to draw about subjective priors really depend on the nature of the dataset they have compiled. Could this have not been discovered with much more limited methodology?

On balance, while this work is definitely noteworthy even just by virtue of the unprecedented scale of the accompanying dataset and its public availability, in order to merit publication in Nature, we would need to be shown more convincingly that such an approach has the potential to expand our understanding of how the brain works. Because as it stands right now, the biggest conclusion that one walks away from reading these papers is, ironically, that generating and then investigating enormous datasets with correlative analytics and without grounding hypotheses is a fairly impotent approach that is likely not worth the effort.

Fortunately, I think this is not actually the case, and there are probably countless ways that the authors could spend a bit more time with this dataset to make some genuine, likely impactful discoveries. Without going deep into specifics, I will provide a few ideas for avenues that the authors could pursue to achieve this (avenues which are likely already being explored).

1) The dataset contains neurophysiological and behavioral measures from 115 mice performing the exact same, standardized task. Presumably then, there is some degree of individual variability in task performance (objective as well as idiosyncratic variability in task strategy). This seems like a perfect opportunity to investigate, even in a coarse manner, how such differences in behavioral variability may be related to underlying differences in neural dynamics and mechanisms. For example, are there specific brain regions, or clusters of regions that, beyond encoding task variables to whatever degree, are highly predictive of task performance, or are correlated with the extent to which animals use a particular cognitive strategy (such as how much they depend on estimates of priors?).

2) While restricted because of the geometry of single probe insertions, the authors have access to a wealth of information about simultaneous neural dynamics recorded across different brain regions (at least those that are dorsoventral neighbors). Despite the repeated mention of “loops” and “widespread, distributed activity,” there is astonishingly no effort here to leverage the dataset to understand anything about inter-areal communication during decision-making. It would be nice to get at least some insight into this knowledge gap that the authors allude to, certainly have the capacity to address, but then say nothing substantial about.

3) Given the unprecedented nature of the dataset, it would also be interesting to see some quantitative assessment of how useful enormous amounts of neurophysiological data really are. The authors are in a unique position to address this. For example, how strongly do conclusions about task variable coding maps hold as a function of the number of neurons recorded, the number of sessions, mice, etc.? If it turns out to be the case that most of the main conclusions could be drawn with much more restricted coverage, i.e. that there are dramatic diminishing returns on scaling up the sheer quantity of neurophysiological data, then this would seem to be an important conclusion to spell out so that future efforts can be better directed toward more fruitful avenues.

Finally, there is a potential problem in the task design. In most trials, whether animals are rewarded or not is determined by the correctness of the choice based on the sensory cue. However, in the 0% contrast trials, where there is no “correct” choice, the probabilities of reward on the left and right sides were set to 20% or 80% according to the prior probability of left and right stimuli. This means that, in principle, mice can use these differential reward probabilities in 0% contrast trials to bias their choice. In other words, the bias in reward probability during 0% contrast trials, rather than the prior probabilities of left and right trials, could have contributed to or been the main cause of the choice bias. It is important to exclude this possibility to support the overall conclusions.

Overall, my sense is that this work could ultimately be impactful and unique enough to merit publication in Nature, but definitely not in its current form. The authors must do a better job of convincing us that this approach is worthwhile in terms of real intellectual and scientific return – the dataset alone, while obviously impressive (and exciting), is not enough.

Referee #2

(Remarks to the Author)

Key results: The manuscript from Findling, Hubert et al examines how and where prior information is encoded in the brain of mice performing the reproducible visuomotor decision task established by the International Brain Laboratory. To tackle this question, the authors leveraged the design of the task: In most trials of a session, the probability of the visual stimulus

appearing on the right or the left side is not 50% and this probability bias (20%/80% respectively 80%/20%) switches in blocks of variable lengths. Due to this, in trials without stimulus presentation, the information about the probability bias of the current block (prior information) can be used by the mouse to inform its decision. The authors demonstrate that mice perform above chance on zero-contrast trials and that the probability bias modulates the mice's choices across all stimuli contrasts. Moreover, they demonstrate that the prior information can be decoded from >20% of brain regions spanning the whole hierarchy from sensory to high-level brain regions. They verified that the prior information is not solely due to the behavior of the mouse, that the results are reproducible across modalities (albeit with different regions), and that an action kernel model captures well the strategy adopted by the mouse to estimate the prior in this task. The widespread nature of the prior information is interpreted as consistent with a neural model of Bayesian interference.

Originality and significance: This manuscript offers a unique and comprehensive perspective on prior encoding made possible by the dataset collected through the International Brain Laboratory initiative.

Data & Methodology: The claims made in the paper are well supported by a range of careful analyses and convincing controls. The data quality is very high, as it benefited from all the quality control steps implemented by the International Brain Laboratory. As a disclaimer, the reviewer is not specialized in the field of computational neuroscience and therefore cannot fully assess the correctness and novelty of the different mathematical models used (i.e. action kernel model versus Bayes optimal prior). The manuscript is well-written, easy to follow, and logically articulated.

Appropriate use of statistics and treatment of uncertainties: The authors took great care to address potential problems, for example linked to unknown underlying distributions by creating pseudo sessions.

Conclusions: robustness, validity, reliability. The conclusion, namely that the prior information is distributed in the brain, is solid and demonstrated with different independent datasets.

Suggested improvements: We noted that the authors extract a single, focused message from the high-dimensional data collected: the prior information is distributed across the hierarchy. While this is appreciable for clarity, it also reduces the level of details, i.e. on differences across specific brain regions that could be extracted from such a dataset and might be of interest to readers. We make some suggestions below on where to push the analyses further, and we also propose additional controls that we think would contribute to testing the robustness of the findings.

Main points:

1. Brain-maps of the prior decoding: A valuable addition to the analysis in Figure 2 would be the presentation of metrics that would allow us to compare the significantly decodable regions. i.e. using single-cell statistics. The population decoding in one region could arise from a strong effect in a sparse number of cells or be distributed across all cells in the region. It may be hard to answer this question given the nature of the data that pools different sessions and different neurons. Nonetheless, any metric that would indicate how the prior decoding is distributed at the single-cell level could reveal differences between brain regions (or types of regions, sensory, motor, "higher" cortical). Or is the assumption that the prior is encoded in the same way in all these regions? Along the same lines, the Swanson maps provided indicate both significance and effect size, but the effect size is not discussed. Can we draw conclusion from the effect size, i.e. from the R2 of MOp being bigger than VISp? If not, why?
2. Concerning decoding from the prior residual (Figure 2f), a map at the region level would be useful. How does each region perform? Is the decoding lower in motor/sensory areas than in higher-level cortical areas after removing the embodied prior, for example? Here we are left with a global picture but having more details would be more satisfying. Assumptions can be made from the known roles of certain regions and tested accordingly. This has been done nicely for visual regions and eye position, but could be extended. Similarly, is prior decoding possible from residuals after regressing out the contribution of both uninstructed movements and eye position?
3. Prediction of the behavior: Figure 2e nicely shows that the decoded prior is predictive of the animal's decision. How does this observation hold true for individual regions or systems (sensory, motor, "higher" cortical)?
4. When merging WFI and ephys data (Figure 2b), the results are presented in the same brain space as for the electrophysiology data. Wouldn't it make sense to mask deep regions that are not recorded by WFI? Generally, it is hard to grasp what we gain from this mixed map, compared to Fig 2c, and if it is interpretable (especially as in this combinatorial map additional units were included that were not separately analyzed elsewhere).
5. Decoding window: The results and conclusions are mostly based on one decoding window (-600ms to -100ms). It would be a valuable addition to see how the overall conclusion (number of regions with significant decoding and types of regions) changes by varying the decoding window. It would be especially interesting to get an impression of the regions with significant decoding during the time where there is by definition of the task no wheel movement (-200ms to 0s). This could be included in Figure 3, where the authors already demonstrate similar results in a post-stimulus time window.
6. Different ITIs: A concern regarding the generalizability of the conclusions is to what degree the very rapid task structure contributes to the observed effects, i.e. on how many past actions do contribute to the prior decoding. The large number of analyzed behavior trials may allow for comparing prior information encoding during trials with longer versus shorter ITIs at the behavioral as well as neuronal levels. Is prior encoding in the brain sensitive to the length of the ITI ?

7. PSTHs: The PSTHs shown in Figure S3 for significant neurons are very intuitive. It would be interesting to see how the firing rates of these neurons look specifically during time points when the prior is stable (trials at the end of a block) versus unstable (90 trials where there is no bias in stimulus presentation, or the trials just after a block switch).

8. On a similar note, the authors in each session have data from 90 trials where the probability of stimulus presentation is at 50%. The authors could provide control plots of the behavior on 0 contrast trials during these 90 trials to show that no bias exists (in Fig 1c).

Minor points:

1. Can the authors comment on the inversion of prior representation in the V1Sp example neuron compared to the other regions (Figure S3). Is this a region-wide effect?

2. In Figure S8, the authors raise an interesting point of embodiment but could analyze in more detail which DLC parameters do successfully decode the prior (whisking, posture etc.)

3. Colorbar labels in Figure 2b are too small

4. The Swanson plots are not very intuitive. For readers familiar with the mouse brain anatomy, it would also help to have R2 maps presented as a set of coronal slices of the Allen Brain Institute reference atlas (in a supplementary figure for example).

5. In Figure 3b, the fit seems to depend heavily on the two extreme points. Can you comment on the robustness of the fit?

6. The validity of the statistics depends heavily on Fisher's method, so a short description of the assumptions and validity criteria would be useful for readers unfamiliar with the method.

7. Bar plots such as in Figure 4c,d: violin plots would better convey the shape of the compared distributions

8. Comment: it would be ideal if the authors could discuss the generalization of their findings. For example, mice follow the action kernel model: is this applicable/valid in other tasks? Of course, acquiring data with another task is not feasible and beyond the scope, but discussing the literature in more detail, or analyzing another publicly available dataset would be an alternative.

9. Generally, the discussion is focused on what type of neural code could sustain the prior and could be richer regarding the behavioral strategy of mice and the involvement of specific regions in the "distributed" encoding.

10. Could the authors discuss why the mice perform worse than the optimal agent and follow an action kernel? That is very interesting. For this simple task, shouldn't we expect optimal behavior?

Referee #3

(Remarks to the Author)

In their manuscript, "Brain-wide representations of prior information in mouse decision-making", the authors leverage the rich data set assembled by the International Brain Laboratory to address how prior information about the state of the world is represented in neural circuits. One possibility is that information about prior probability affects choices in the late stages of decision-making, and is therefore represented quite narrowly. Alternatively, probabilistic information about the state of the world may be carried broadly across neural networks. By using standardized neural recordings acquired across many brain regions simultaneously in mice performing a standardized decision-making task, in combination with recordings of motor activity, the authors have a broad data set to explore this question. They use a decoding model to predict the prior probability that a stimulus will be presented on the right side from neurons recorded in the contralateral fore- and midbrain and ipsilateral cerebellum and hindbrain. In the behavioral task, the prior alternates, in blocks of random lengths, between 0.8 and 0.2. Each session begins with a block of 0.5 prior probability. The main finding is a broad representation of the Bayes-optimal prior in neural activity – whether measured as single-unit electrophysiology or widefield imaging. Evidence of the neural signal of the prior persists when taking into account the movement of the animal as well as response history in the task.

While understandable given that many of the methods refer to other papers produced from the consortium – some published, some in preprint, reading this as a stand-alone article is a bit challenging. One correction to provide clarity would be to add a citation for the companion bioRxiv pre-print (<https://doi.org/10.1101/2023.07.04.547681>), which they refer to as the BWM paper in the Methods. Currently, the authors cite the International Brain Lab et al. (2023) data release, but also refer to the BWM preprint using the same citation. Please include the companion paper preprint in the references and clarify throughout which 2023 paper is being referenced.

Compared with the BWM preprint, the authors here take a much deeper dive into what information can be decoded about the Bayes-optimal prior (subjectively ascertained) during the quiescent period before stimulus onset. The results here are quite different from the BWM preprint, which shows very sparse and localized neural decoding of the objective priors. The authors list several methodological differences between the two papers, but it is not obvious how much these contribute:

- What is the impact of the using a longer vs shorter time window?

- How much difference comes from estimating the Bayes-optimal prior vs. true block prior (based on Fig 2A, S3, animals

seem to track the priors quite well - they do not diverge extensively)?

- To what extent does the power of including more sessions make it possible to see the decoding?

While all are speculated as contributing, have the authors quantified the sources of discrepancies – this is relevant for the companion paper as well, which presents a very different result.

The authors focus on the 0.8:0.2 blocks of trials in test sessions. Each session begins with 90 trials of 0.5:0.5 priors. Do the areas that decode priors in the 0.8 and 0.2 blocks not decode in the 0.5 block?

More broadly for discussion, what does it mean for a brain area to show activity that significantly modulates with prior probability? How should this differ between areas that are more or less heterogeneous? It is not known what types of cells are recorded – projection neurons, interneurons, etc. Might these encode task variables differently, and to what extent does this limit the decoding model?

Also broadly, the results here suggest a large proportion of brain areas carry information about expected probability. These analyses relied on a single, relatively long (in neural coding terms) period of time. Could the data be leveraged in the future to show temporal dynamics among the many areas that show biased activity in line with the action kernel/bayes-optimal prior? This kind of temporal resolution should be available in the ephys data and provide a jumping off point for direct comparisons of time-series differences between recording techniques.

Very minor suggestion: It is difficult to make quantitative comparisons across the ephys and WBI analysis. It would be easier to see both the relationships and their magnitudes by equating the axes in Figures 2e, 3d, 4d, 4e, 4f.

Version 1:

Reviewer comments:

Referee #1

(Remarks to the Author)

The authors have performed novel analyses to reinforce their arguments and present novel findings that were only possible with the unique dataset. Overall, this revision has substantially improved the manuscript. The team effort to collect an extensive dataset spanning many brain regions is significant and offers a valuable resource for the neuroscience community to build upon and explore future research directions.

Referee #2

(Remarks to the Author)

In the revised version of the manuscript "Brain-wide representations of prior information in mouse decision-making," the authors have adequately addressed all of the reviewer's comments. The reviewer appreciates the inclusion of additional analyses, such as controls for the robustness of decoding and for potential confounds like motor features and eye position.

The core finding of the paper is that prior information is encoded across all levels of the brain, not just in decision-making areas, thus resolving an ongoing debate that smaller-scale neuroscience studies could not address. As such, the paper leverages and demonstrates the power of large-scale collaborative data collection initiatives in systems neuroscience. We recommend its publication, alongside the companion paper.

Referees' comments:

All reviewers appreciated the unique contribution of the dataset, the breadth of the questions addressed, and the statistical rigor employed. In response to their feedback, we have removed all analyses of the block prior from the companion paper and have incorporated more region-level analyses, including a Granger causality analysis of areas recorded simultaneously. Moreover, we have substantially expanded the dataset, now reporting on the activity of approximately three times the original number of neurons and covering 50% more brain areas than previously.

Referee #1 (Remarks to the Author):

In this pair of companion studies, the authors (members of an international consortium for experimental neuroscience called the International Brain Lab; IBL) present an unprecedented dataset, collected across 11 labs, comprised of 547 high-density neural recordings sampled across the entire mouse brain during a standardized decision-making task. In one study, the dataset (curated and now made publicly available) and methodology are described, and then used to perform a battery of correlational analyses across the brain to identify encoding of simple task variables during decision-making (including sensory, choice, outcome, movement, and prior information). Based on this investigation and application of rigorous statistical methods, the authors conclude that different task variables are encoded in different ways (some are represented more widely across the brain, while others are more localized), confirming the results of countless previous reports. In the second study, the authors investigate more deeply the encoding of prior information using a more refined set of analytical tools and combining their electrophysiology dataset with another large calcium imaging dataset collected by IBL participants. The main conclusion of this second study is that, in contrast to the initial conclusion of the first study, which suggested that prior information is encoded in a restricted set of brain regions, subjective priors are actually encoded more widely across cortical, subcortical, and midbrain regions. Further, this more widespread subjective prior is partially embodied in the animal's behavior and is driven by action history (as opposed to sensory and/or outcome history exclusively).

The dataset collected here, and particularly the effort to generate a dataset of such comprehensive scale with rigorous control and standardization of task conditions, preprocessing parameters, etc. is obviously noteworthy and is inarguably a highly unique and important contribution to the field. The presented impetus for this enormous collaborative effort is that our capacity to really gain insight into how the brain processes sensory information, computes decisions, and generates behavioral outputs (i.e. how the

brain works) is substantially hindered by the limitations of individual lab groups to survey brain structures comprehensively under consistent conditions, and the idiosyncrasies with which different groups design and implement their experiments that limit our capacity to draw accurate comparisons and generalizations. The assumption underlying and framing the effort is that, with a big enough dataset, collected with careful methodological standardization, we should be able to mine the data to gain novel insight into how the brain works. It is, in some sense, the extreme terminus of “data-driven” approaches.

Somewhat unfortunately, the studies, even taken together, fall woefully short of this laudable goal. In the first study, there are in many cases no explicit hypotheses given to ground the questions the authors ask of the data, or if there are, they are so vague as to be almost useless (e.g. visual information may be encoded to some degree beyond classically defined visual areas). We are presented with what amounts to a brain-wide “screen” of task variable neural correlates, and learn, in the end, essentially nothing new about how the brain works – visual stimuli are predominantly represented in visual cortex and thalamus; movement correlates are widespread across the brain, as are responses to salient rewards and punishment. And, at least in this particular case, this is doubly a problem because the specific framing of the knowledge gap the work is intended to fill is itself defeated – the need for this type of approach totally obviated – by the apparent lack of novel insight. If recording from 300,000 neurons sampled from the entire mouse brain during a well-designed decision-making task does not yield any novel insight, or change our understanding of the brain in any substantial way, then what is the value of such an approach at all?

In the second companion paper, the authors go a bit further. They are able to draw some interesting, useful conclusions from the work that might have been difficult (but this is not totally clear) to fully capture with more limited, fragmented efforts. In particular, they show that while “objective” prior information is only rarely encoded in restricted brain regions (in the first paper), “subjective” prior information is, in fact, much more robustly encoded, and across several levels of brain structure – and this prior takes the form of a partially-embodied action prior, as opposed to other formulations. In the context of value-based decision, this is indeed an important finding, but in the specific context of these two studies, it is still limited in scope for at least two reasons: First, it somewhat contradicts the conclusions of their first study – the authors make a big deal in study 1 about how the restricted nature of prior encoding is interesting because it stands in contrast to how the other task variables are encoded, but then proceed to explain this difference by arguing that this is probably just because animals do not have access to an objective prior, and in fact, encode subjective priors quite robustly. If this is the case, why not just present the analysis of subjective priors alongside other task variables in the first study? Second, it is not clear to what extent the positive conclusions they are able to draw about subjective priors really depend on the nature of the dataset they have compiled. Could this have not been discovered with much more limited methodology?

On balance, while this work is definitely noteworthy even just by virtue of the unprecedented scale of the accompanying dataset and its public availability, in order to merit publication in Nature, we would need to be shown more convincingly that such an approach has the potential to expand our understanding of how the brain works. Because as it stands right now, the biggest conclusion that one walks away from reading these papers is, ironically, that generating and then investigating enormous datasets with correlative analytics and without grounding hypotheses is a fairly impotent approach that is likely not worth the effort.

We agree that collecting such an extensive dataset on decision-making is only worthwhile if it enables us to test specific hypotheses. This was our goal with the present paper, which allowed us to explore a long standing issue in the domain of neural Bayesian inference, namely, whether prior expectations are incorporated at late stages of processing, or percolate throughout the brain. The original manuscript argued for the latter but the reviewer made important and valuable suggestions for further analysis and critical controls. As highlighted below, we have explored all of these suggestions in depth and confirmed our initial conclusions. The Granger causality analysis was particularly insightful, revealing that the prior signal is transmitted through multiple loops, beyond what is expected by chance, and consistent with a loopy Bayesian network. We have also added several other key analyses in response to the other reviewers' comments. We hope the reviewer will agree with us that the revised manuscript now makes a compelling case.

Fortunately, I think this is not actually the case, and there are probably countless ways that the authors could spend a bit more time with this dataset to make some genuine, likely impactful discoveries. Without going deep into specifics, I will provide a few ideas for avenues that the authors could pursue to achieve this (avenues which are likely already being explored).

- 1) The dataset contains neurophysiological and behavioral measures from 115 mice performing the exact same, standardized task. Presumably then, there is some degree of individual variability in task performance (objective as well as idiosyncratic variability in task strategy). This seems like a perfect opportunity to investigate, even in a coarse manner, how such differences in behavioral variability may be related to underlying differences in neural dynamics and mechanisms. For example, are there specific brain regions, or clusters of regions that, beyond encoding task variables to whatever degree, are highly predictive of task performance, or are correlated with the extent to which animals use a particular cognitive strategy (such as how much they depend on estimates of priors?).**

To look into the relationship between behavioral performance and specific regions, we first asked whether the neural and behavioral learning rates are correlated at the region level. The original manuscript had a similar analysis but at the session level. This analysis was carried out using widefield calcium imaging data (WFI) for the session-regions that significantly reflect the prior, as shown in our main analysis (see Figure 2c,b, with about 20 sessions per region, refer to Fig S7a,b).

Briefly, the action kernel prior is derived by calculating an exponentially weighted average of recent past actions, with the behavioral learning rate governing the rate of the exponential decay. This variable is inversely proportional to the decay constant of the action kernel. The neural learning rates are obtained by fitting similar auto-regressive models to the neural activity of single pixels. Neural learning rates are then averaged across all pixels within a region to obtain a neural learning rate per region and per session.

Figure S19. Correlation between behavioral and neural inverse decay constants across sessions at the region level. A decay constant is estimated for each pixel (as in figure 4f, refer to Methods), but now, averages are taken across pixels for each session and specific region. In the analysis figure 4f, session-level learning rates were obtained by averaging across all pixels, regardless of region identity. **a.** Regions with a significant correlation between behavioral and neural inverse decay constants. As expected, only positive correlations emerge as significant. **b.** Correlation between behavioral and neural inverse decay constants is correlated with the prior decoding corrected R^2 from the same regions. Remarkably, these two quantities were found to be also correlated ($R=0.46$, $p=0.008$). In other words, regions in which the prior decoding R^2 is large are also regions which best reflect the behavioral decay constant, i.e., these are the regions that are best correlated with the animals' cognitive strategies as assessed by the lengths of the action kernels. We did not repeat this analysis with the electrophysiology recordings because we only have a very limited number of significant sessions per region (1-2 for most regions, as opposed to around 20 sessions per region for the WFI data - see Fig. S7a,b)

We found that the neural and behavioral learning rates are significantly correlated, indicating that when the animal uses a longer decay constant to estimate the prior, this is reflected in the time constant of neural fluctuations in individual brain regions (see figure above). This seems to be particularly pronounced for associative areas, such as secondary motor cortex and retrosplenial areas, as opposed to primary visual and primary motor cortex.

Next, we asked whether this correlation between behavioral and neural learning rate is correlated with the prior decoding corrected R^2 from the same regions. Remarkably, these two quantities were found to be also correlated. In other words, regions in which the prior decoding R^2 is large are also regions which best reflect the behavioral decay constant, i.e., these are the regions that are best correlated with the animals' cognitive strategies as assessed by the lengths of the action kernels.

We did not repeat this analysis with the electrophysiology recordings because we only have a very limited number of significant sessions per region (1-2 for most regions, as opposed to around 20 sessions per region for the WFI data - see Supplementary Figure 7a,b)

These results are all reported at the end of the results of the main manuscript, along with new supplementary figures S19. The following text was added:

We next examined the link between behavioral performance and specific brain regions by comparing their neural inverse decay constants with the behavioral inverse decay constants. Notably, associative areas like the secondary motor cortex and retrosplenial areas more closely mirrored these behavioral constants than the primary visual and motor cortex (fig S19a). We also observed that the correlation between behavioral and neural decay constants reflected the prior corrected R^2 from the same regions, indicating that regions with higher prior decoding R^2 scores best align with the animal's cognitive strategies as measured by the action kernel lengths (fig S19b). This analysis was not extended to electrophysiology recordings due to the limited number of available sessions per region (see fig S7a,b).

2) While restricted because of the geometry of single probe insertions, the authors have access to a wealth of information about simultaneous neural dynamics recorded across different brain regions (at least those that are dorsoventral neighbors). Despite the repeated mention of “loops” and “widespread, distributed activity,” there is astonishingly no effort here to leverage the dataset to understand anything about inter-areal communication during decision-making. It would be nice to get at least some insight into this knowledge gap that the authors allude to, certainly have the capacity to address, but then say nothing substantial about.

We have addressed the concern regarding inter-areal communication by conducting a Granger causality analysis using both electrophysiology (Ephys) and wide-field imaging (WFI) data. This analysis was performed during the same inter-trial interval (ITI) used for prior signal decoding. Crucially, we first project the neural activity from each brain region onto the Bayes Optimal prior decoding axis. This projection involved decoding neural activity from 50ms time bins in Ephys, and from each frame in Widefield (66 ms per frame), to create time-series of decoded prior information for each area. Granger causality was then assessed on these time series of decoded prior information from each region. Similarly to all the other analyses in the paper, the null distribution is defined here by applying the same pipeline to pseudo priors - see Methods for more details.

Our analysis first reveals notable communication concerning the prior between regions.

Average percentage of significant directed pairs between 2 regions that reflect the prior across sessions. When considering all pairs of regions encoding the prior significantly, and for which we had simultaneous recordings, we observed that information was significantly exchanged between 71% of these pairs in Widefield imaging and 36% in Ephys. Blue histograms: null distribution.

When considering all pairs of regions encoding the prior significantly, and for which we had simultaneous recordings, we observed that information was significantly exchanged between approximately 71% of these pairs in Widefield imaging and 36% in Ephys. This level of inter-regional communication significantly surpasses the rates that would be anticipated by chance, indicating a robust pattern of directional information flow between these areas.

When looking particularly at the Granger directed pairs themselves, we find the following graph in Ephys:

All significant directed pairs ($p < 0.05$ uncorrected)

Complete connectivity graph from Ephys ($p < 0.05$ uncorrected for multiple comparison).

It should be noted that the pairs are considered significant if their associated p-value is below 0.05, without applying any corrections for multiple comparisons. However, when adjustments for multiple comparisons are made, none of the links remains significant. This lack of significant findings post-correction is attributed to the sparse nature of the observations in Ephys. With a total of 242 observed regions, the possible number of pairs amounts to $242 \times 241 = 58,322$. Of these, approximately 10% of the directed pairs had been recorded simultaneously, and the vast majority (75%) of these pairs appears in two or fewer sessions, highlighting their sparsity.

Histogram showing the number of sessions for each directed pair and barplot showing the percentage of observed directed pairs (directed pairs with at least one session) versus unobserved pairs. In Ephys, with a total of 242 observed regions, the possible number of pairs

amounts to $242 \times 241 = 58,322$. Of these, approximately 10% of the directed pairs had been recorded simultaneously, but the vast majority (75%) of these pairs appeared in two or fewer sessions, highlighting their scarcity.

To address the issue, we aggregated the data into broader regional categories defined by the Cosmos classification, encompassing Isocortex, Hippocampal Formation (HPF), Cortical Nuclear (CNU), Thalamus (TH), Hypothalamus (HY), Midbrain (MB), Hindbrain (HB), and Cerebellum (CB). By pooling the results across these Cosmos areas and employing Fisher's method to combine the p-values, followed by a Bonferroni correction for multiple comparisons, we revealed a significant flow of prior information throughout the entire mouse brain. This flow occurs both from subcortical to cortical regions, such as from the Isocortex to the Thalamus, and reciprocally, from regions like the Thalamus and Midbrain back to the Isocortex.

All significant directed pairs ($p < 0.05$ Bonferroni corrected)

Connectivity graph in Ephys across Cosmos regions ($p < 0.05$ Bonferroni corrected). Effect sizes across directed pairs of regions are aggregated at the Cosmos level with Fisher's method

Widefield provides a richer dataset, because, with 32 regions recorded simultaneously, we can analyze a total of 992 possible directed pairs (32×31), most of them available on all sessions.

Histogram showing the number of sessions for each directed pair and barplot (top) showing the percentage of observed directed pairs (directed pairs with at least one session) versus unobserved pairs. Widefield provides a richer dataset, because, with 32 regions recorded simultaneously, we can analyze a total of 992 possible directed pairs (32×31), most of them available on all sessions.

The complete connectivity graph from the Widefield data, corrected for multiple comparisons using the Bonferroni method, is densely populated and consequently difficult to interpret.

All significant directed pairs ($p < 0.05$ Bonferroni corrected)

Complete connectivity graph from Widefield ($p < 0.05$ Bonferroni corrected).

However, by concentrating on the directed pairs projecting to the primary visual cortex, we uncover feedback connections from higher-order areas — such as the Motor Cortex (MOs), Ventral

Retrosplenial Cortex (RSPv), Prelimbic Cortex (PL), and Anterior Cingulate Area Dorsal (ACAd)
— to the early sensory area, the Primary Visual Cortex (VISp).

Significant directed pairs to VISp ($p < 0.05$ Bonferroni corrected)

A partial connectivity graph from Widefield, highlighting significant directed pairs projecting to the Primary Visual Cortex (VISp). Consistent with predictions from a Bayesian network model of the brain, we uncover feedback connections from higher-order areas such as the Motor Cortex (MOs), Ventral Retrosplenial Cortex (RSPv), Prelimbic Cortex (PL), and Anterior Cingulate Area Dorsal (ACAd) — these regions are marked with grey circles for emphasis — to the early sensory area, the Primary Visual Cortex (VISp)

Up to this point, we have demonstrated that the flow of prior information between regions exceeds what would typically be expected by chance. This flow encompasses communications throughout the entire brain and includes feedback connections within the cortex, from higher-order to early sensory areas.

We can now investigate whether this flow of prior information forms loops. By examining how often a significant directed pair (A->B) is reciprocated within the same session by its counterpart (B->A), we found this to occur approximately 38% of the time in Widefield and 11% in Ephys. In both instances, these rates significantly exceed what would be anticipated by chance.

Average percentage of significant directed pair (A->B) which is reciprocated within the same session by its counterpart (B->A), we found this to occur 38% of the time in Widefield and 11% in Ephys. Blue histograms: null distribution.

We investigated the existence of loops in the connections identified in Widefield imaging between the primary visual cortex (VISp) and higher-order areas — Motor Cortex (MOs), Ventral Retrosplenial Cortex (RSPv), Prelimbic Cortex (PL), and Anterior Cingulate Area Dorsal (ACAd), as illustrated in the second-to-last figure. We focused on sessions where both VISp and a higher-order region (A) significantly reflected the prior, computing the percentage of these sessions that exhibited a significant reciprocal connection pattern (A->VISp->A):

Percent of sessions with a significant loop (A->VISp->A)

Percentage of sessions exhibiting significant reciprocal connections (A->VISp->A) for sessions in which the Bayes optimal prior could be significantly decoded from both VISp and the

higher-order regions. The size of the arrow is proportional to the percentage. Our findings indicate the existence of reciprocal connections in these sessions: 33.3% between MOs and VISp, 16.7% between ACAd and VISp, 20% between PL and VISp, and 18.75% between RSPv and VISp.

Our analysis revealed the presence of significant loops for each region: 33.3% of sessions showed a significant loop between MOs and VISp; for ACAd and VISp, the percentage was 16.7%; for PL and VISp, 20%; and for RSPv and VISp, 18.75%.

In conclusion, by employing Granger causality analysis on both Ephys and wide-field imaging data, we confirmed that the subjective prior is being communicated back and forth between regions at all levels of processing. We thank the reviewer for inspiring this new analysis which nicely reinforces our original conclusions that the brain is akin to a loopy Bayesian network.

These results are all reported in the main manuscript, along with new supplementary figures S11 along with the following paragraph in the main text:

Our decoding analysis reveals a robust, distributed representation of the Bayes-optimal prior throughout the brain, suggesting a complex network of information flow. To investigate the dynamics of the prior information network, we conducted a Granger Causality analysis during the ITI, between the time series of the decoded prior from one brain region and that of another (see Methods and Supp Fig. S11). This analysis revealed several key findings: 1) The flow of prior information between brain areas is significantly greater than expected by chance (Supp Fig. S11a), 2) This prior flow includes comprehensive communications across the entire brain, from subcortical to cortical areas and vice versa (Fig. 2g left panel), 3) It includes significant feedback connections from higher-order areas to early sensory areas (Fig. 2g right panel), and 4) There is a higher prevalence of loops within this communication network than would be anticipated by chance (Fig. 2h), including between higher-order and early sensory areas (Supp Fig. S11e). These results collectively highlight a loopy and intricate inter-area communication of prior information within the brain.

3) Given the unprecedented nature of the dataset, it would also be interesting to see some quantitative assessment of how useful enormous amounts of neurophysiological data really are. The authors are in a unique position to address this. For example, how strongly do conclusions about task variable coding maps hold as a function of the number of neurons recorded, the number of sessions, mice, etc.? If it turns out to be the case that most of the main conclusions could be drawn with much more restricted coverage, i.e. that there are dramatic diminishing returns on scaling up the sheer quantity of neurophysiological data, then this would seem to be an important conclusion to spell out so that future efforts can be better directed toward more fruitful avenues.

Thank you for highlighting this question concerning the value of our extensive neurophysiological dataset. The question of the relevance of the dataset's size in our study is complex to address. Our objective with the brain-wide map effort is to achieve a comprehensive coverage of the mouse brain, targeting over 95% of brain volume. This goal introduces significant challenges due to the varied sizes, shapes and positions of these regions, necessitating numerous recordings, particularly to reach smaller and subcortical nuclei.

When examining the number of sessions per region (see figure below), it's clear that many areas, especially subcortical nuclei, do not have extensive recordings. The distribution of the number of sessions shows a fair amount of variability spanning from 3 (first quartile) to 11 (third quartile) with a median of 6 recordings per region.

Histogram detailing the number of sessions per region in Ephys

Some nuclei, such as the anterior hypothalamic nucleus, the peripeduncular nucleus, and the subgeniulate nucleus, have as few as one recording. This highlights a significant unavoidable variability in data coverage across different brain regions. Reducing the number of sessions recorded will lead to a particular loss of coverage of these small regions.

One interesting question that we can answer with our dataset is the number of recordings necessary per region to achieve significant prior decoding. To address this, we analyzed the number of recordings required to reach the significance threshold in regions identified as significant (according to the main decoding analysis presented fig 2b).

Dual-axis graph displaying the p-value of regional significance as a function of the number of decoding sessions (left axis) and the ratio of significant regions relative to the total number of sessions (right axis). It is estimated that approximately 10 sessions per region are necessary to identify 95% of significant regions highlighted in the main decoding analysis (refer to Methods section for more details). It is important to recognize that this analysis has limitations: it assumes uniformity across recordings and regions without considering, e.g., variations in effect size or number of units per recording. Despite these limitations, we concentrated on the number of recordings because it is a primary factor that experimenters can directly control.

For each region, color-coded according to the Swanson map (Fig. S5a), we randomly sampled a variable number of recordings and tested for significance, repeating this process 1'000 times per region. We report the median p-value across these repeated samplings. This analysis allows us to identify the average number of sessions required to reach the significance threshold for a given region. We then ask how many recordings are required in order to find 95% of the regions to be significant.

We found that approximately 10 recordings per region are necessary to reach this 95% target (see cumulative curve of significant regions shown in black). The results of this analysis suggest that the median of six recordings per region falls on the lower end of what is needed to yield robust decoding results.

We have added the following sentence to the manuscript:

An analysis to determine the necessary number of recordings per region indicated that around 10 recordings per region are required to reach the obtained significance levels (see Fig. S5e). Given that the median number of sessions per region in Ephys is 6 (see Fig. S5d), it is likely that the reported levels of significance are underestimated.

Finally, there is a potential problem in the task design. In most trials, whether animals are rewarded or not is determined by the correctness of the choice based on the sensory cue. However, in the 0% contrast trials, where there is no “correct” choice, the probabilities of reward on the left and right sides were set to 20% or 80% according to the prior probability of left and right stimuli. This means that, in principle, mice can use these differential reward probabilities in 0% contrast trials to bias their choice. In other words, the bias in reward probability during 0% contrast trials, rather than the prior probabilities of left and right trials, could have contributed to or been the main cause of the choice bias. It is important to exclude this possibility to support the overall conclusions.

This is indeed an important point. To determine whether animals exclusively use the 0% contrast trials to update their prior, we considered two behavioral models. The first is our best fitting behavioral model: the action Kernel model which assumes that they estimate a prior probability that the stimulus will appear on the right by performing an exponentially weighted average over previous chosen actions. The second is an action Kernel model which performs the same operation but only on the 0% contrast trials. In other words, it performs an exponentially decaying average over previous 0% contrast chosen actions. When performing model comparison, we found that the first model explains behavior significantly better (exceedance probability > 0.999 - see figure below) showing that mice do not restrict themselves to 0% contrast trials to estimate the prior over stimulus side.

Bayesian model comparison for 2 behavioral models, the action kernel and a variant that operates only during 0% contrast trials (by calculating an exponentially decaying

average of chosen actions at 0% contrast trials). Our comparisons indicate that the action kernel, updating across all contrasts, more effectively explains behavior (exceedance probability > 0.999), suggesting that mice do not limit their subjective prior estimations to 0% contrast trials alone.

Additionally, we can confirm that mice utilize all contrast levels to update their priors by examining the behavioral signature of the action kernel model (fig 4c). When considering behavior within blocks, an agent using an action kernel prior should show a higher percentage of correct responses following a correct, block-consistent action compared to an incorrect one. This is because, on incorrect trials, the prior is updated with an action corresponding to the incorrect stimulus side. This asymmetry, which is absent in models using either the Bayes-optimal prior or the stimulus kernel prior (since these models update based on the actual stimulus, discernible from the combination of action and reward), is evident in mice behavior.

Performance on zero contrast trials conditioned on whether the previous action is correct or incorrect for various behavioral models and for the animals' behavior.

This analysis was initially performed considering all contrasts for the previous trial. We can repeat this analysis, restricting it to non-zero contrast trials. If an asymmetry in behavior is still observed, it would imply that mice update their priors based on non-zero contrast trials as well.

Performance on zero contrast trials, analyzed based on whether the preceding action was correct or incorrect and considering that the previous contrast was non-zero, for the animals' behavior. Performance on zero contrast trials shows significant modulation (Wilcoxon test, $T=11734$, $p=1.1E-15$)

Even when limited to previous trials with non-zero contrast, there is a notable difference in the probability of making a correct decision following an incorrect vs. a correct choice. This demonstrates that mice update their priors based on all contrast levels, not just zero.

We have added these panels to supplementary figure S15 and the following sentence to the main text:

Additionally, mice relied on more than just zero contrast trials to update their subjective prior (supp Fig S15b,c)

Overall, my sense is that this work could ultimately be impactful and unique enough to merit publication in Nature, but definitely not in its current form. The authors must do a better job of convincing us that this approach is worthwhile in terms of real intellectual and scientific return – the dataset alone, while obviously impressive (and exciting), is not enough.

We appreciate the reviewer's insights and believe our revisions address the concerns raised. Accordingly, we have broadened our analysis to include: (1) a detailed exploration of behavioral variability by examining region-level neural learning rates, (2) a comprehensive investigation into the flow of prior information across the brain, and (3) a thorough effect size analysis to evaluate the criticality and impact of our extensive dataset.

Referee #2 (Remarks to the Author):

Key results: The manuscript from Findling, Hubert et al examines how and where prior information is encoded in the brain of mice performing the reproducible visuomotor decision task established by the International Brain Laboratory. To tackle this question, the authors leveraged the design of the task: In most trials of a session, the probability of the visual stimulus appearing on the right or the left side is not 50% and this probability bias (20%/80% respectively 80%/20%) switches in blocks of variable lengths. Due to this, in trials without stimulus presentation, the information about the probability bias of the current block (prior information) can be used by the mouse to inform its decision. The authors demonstrate that mice perform above chance on zero-contrast trials and that the probability bias modulates the mice's choices across all stimuli contrasts. Moreover, they demonstrate that the prior information can be decoded from >20% of brain regions spanning the whole hierarchy from sensory to high-level brain regions. They verified that the prior information is not solely due to the behavior of the mouse, that the results are reproducible across modalities (albeit with different regions), and that an action kernel model captures well the strategy adopted by the mouse to estimate the prior in this task. The widespread nature of the prior information is interpreted as consistent with a neural model of Bayesian interference.

Originality and significance: This manuscript offers a unique and comprehensive perspective on prior encoding made possible by the dataset collected through the International Brain Laboratory initiative.

Data & Methodology: The claims made in the paper are well supported by a range of careful analyses and convincing controls. The data quality is very high, as it benefited from all the quality control steps implemented by the International Brain Laboratory. As a disclaimer, the reviewer is not specialized in the field of computational neuroscience and therefore cannot fully assess the correctness and novelty of the different mathematical models used (i.e. action kernel model versus Bayes optimal prior). The manuscript is well-written, easy to follow, and logically articulated.

Appropriate use of statistics and treatment of uncertainties: The authors took great care to address potential problems, for example linked to unknown underlying distributions by creating pseudo sessions.

Conclusions: robustness, validity, reliability. The conclusion, namely that the prior information is distributed in the brain, is solid and demonstrated with different independent datasets.

Thank you for your thoughtful review and constructive feedback on our manuscript. We appreciate your recognition of the comprehensive analyses and robust data quality enabled by the International Brain Laboratory's efforts. Based on your comments, we have expanded our analyses to further enhance the manuscript. This includes additional decodings to assess the robustness against variations in decoding window lengths and potential confounds such as motor features and eye position. We have also expanded our region-level analyses where possible and refined our single-neuron analyses to include measures of certainty in prior information. These enhancements strengthen the validity and thoroughness of our findings.

Suggested improvements: We noted that the authors extract a single, focused message from the high-dimensional data collected: the prior information is distributed across the hierarchy. While this is appreciable for clarity, it also reduces the level of details, i.e. on differences across specific brain regions that could be extracted from such a dataset and might be of interest to readers. We make some suggestions below on where to push the analyses further, and we also propose additional controls that we think would contribute to testing the robustness of the findings.

Main points:

- 1. Brain-maps of the prior decoding: A valuable addition to the analysis in Figure 2 would be the presentation of metrics that would allow us to compare the significantly decodable regions. i.e. using single-cell statistics. The population decoding in one region could arise from a strong effect in a sparse number of cells or be distributed across all cells in the region. It may be hard to answer this question given the nature of the data that pools different sessions and different neurons. Nonetheless, any metric that would indicate how the prior decoding is distributed at the single-cell level could reveal differences between brain regions (or types of regions, sensory, motor, “higher” cortical). Or is the assumption that the prior is encoded in the same way in all these regions?**

This question is indeed intriguing and crucial, yet not straightforward to address given that our dataset is optimized to cover the entire brain rather than providing a detailed and comprehensive description of specific regions. Therefore, one must be careful when investigating the sparsity or distribution of the code across regions, as our recordings themselves are sparse.

In response, we examined the sparsity or distribution of prior representations by calculating the proportion of cells significantly contributing to prior decoding (see the *Assessing decoding weights significance* Method section for more details). We used null distributions of weight values derived from decoding pseudo-session priors based on neural activity to determine the percentage of weights that were significant relative to the null. It is important to note that this approach has significant limitations, particularly in terms of explaining away phenomena, where stronger neurons may mask the contributions of weaker ones. Consequently, the results from this analysis of decoder weights should be interpreted with caution.

This analysis targeted sessions and regions where the prior was notably decoded and was applied in both widefield and electrophysiological modalities.

We found an average ratio of significant weights of 53.7% +/- 24.7% in widefield (N=32 regions) and of 24.4% +/- 11.9% in Ephys (N=128 regions) (mean +/- std).

Left: Ratio of significant weights for regions with at least one significant decoding in WFI. A weight is deemed significant if it is unlikely under the pseudo distribution of weights obtained by decoding pseudo priors (see Methods for more details). Right: Same for Ephys.

The differences in ratios are likely due to the regularization methodology employed: Ridge in widefield due to higher input correlations and Lasso in Ephys.

We also explored whether there was a correlation between the ratio of significant weights across the two modalities. Among the 32 regions analyzed in widefield, only 21 had at least one significant session in Ephys. In these 21 regions, we found a correlation (Spearman $R=0.56$, $p=0.008$) between the ratios of significant weights, suggesting that these ratios provide meaningful information.

Correlation between the ratio of significant weights between the two modalities. Note that we only have 21 regions because we only have 21 regions (out of the 32) that have at least one significant decoding in Ephys. We find a significant correlation between the ratio of significant weights between the two modalities.

While these findings are intriguing and hold promise, we leave it to future (ongoing) research to determine precisely how sparse or distributed the coding is across different regions. Given the limitations of decoder weights in accurately representing the data and the potential for misleading conclusions, we refrain from publishing claims about the region-level distribution of neural coding.

Along the same lines, the Swanson maps provided indicate both significance and effect size, but the effect size is not discussed. Can we draw conclusion from the effect size, i.e. from the R² of MOp being bigger than VISp? If not, why?

Interpreting the effect size, which reflects the presence of the prior in different regions, is challenging due to several confounding factors, including the numbers of sessions and recorded units.

In Ephys, we observed a correlation between the prior decoding effect size and both the number of units and the number of recorded sessions.

Number of units (left) and number of recorded sessions (right) as a function of the decoded R^2 for the Bayes-optimal prior in Ephys.

In widefield, we found a correlation of the prior decoding effect size with the number of pixels (all regions have the same number of recorded sessions).

Number of pixels as a function of the decoded R^2 for the Bayes-optimal prior in WFI.

These confounds complicate the interpretation of the effect size. A challenge we faced was showing that the correlation between Ephys and widefield effect sizes reported in Fig 2d was not explained away by correlations between these confounds across modalities. Indeed, we observed (1) significant correlations between the number of recorded sessions in Ephys and the number of pixels in widefield (Spearman correlation $R=0.50$, $p=0.0036$), and (2) a positive correlation (although not significant given a p-value limit of 0.05) between the number of units in Ephys and the number of pixels in widefield (Spearman correlation $R=0.31$, $p=0.08$).

Correlations between confounds across modalities. Left panel: Number of pixels in WFI as a function of the number of units in Ephys. Right: Number of pixels as a function of the number of recorded sessions in Ephys.

To assess the robustness of these correlations despite confounds, we adjusted the widefield effect size for the number of pixels and retested the correlation. We found that the correlation not only persisted but actually strengthened (Supplementary Figure S7f).

Corrected R^2 for Ephys as a function of the corrected R^2 for WFI after correcting the WFI R^2 data for region size. Correcting for the region size in WFI was performed by subtracting the size-predicted R^2 from the WFI R^2 . Each dot corresponds to one region. All Ephys regions (significant and non-significant) were included in this analysis.

We have added the following paragraph to the main text to make this point about confounds clearer:

Interpreting the effect size in both Ephys and widefield modalities is challenging due to confounding factors such as the number of sessions and units in Ephys, and the number of pixels in widefield (Fig. S7c,d). To control for correlations between these confounds across modalities (Fig S7e), we corrected the widefield effect size for region sizes. Despite this correction, the correlation between effect sizes across modalities remained significant, and even strengthened (Fig. S7f) - thus suggesting that the effect sizes we decode are, at least partly, specific to the decoded regions.

2. Concerning decoding from the prior residual (Figure 2f), a map at the region level would be useful. How does each region perform? Is the decoding lower in motor/sensory areas than in higher-level cortical areas after removing the embodied prior, for example? Here we are left with a global picture but having more details would be more satisfying. Assumptions can be made from the known roles of certain regions and tested accordingly. This has been done nicely for visual regions and eye position, but could be extended.

Thank you for your comment. We agree that an analysis at the regional level could indeed be informative. Accordingly, we updated Figure 2f with a color scheme that assigns colors to points based on their regional identity, as defined in Figure S5. This coloring provides clearer visual information about the placement of different regions.

The corrected R² values for decoding the prior from neural activity are correlated with the corrected R² values for decoding the residual prior (Bayes-optimal prior minus Bayes-optimal prior decoded from DLC features). This correlation implies that the Bayes-optimal prior decoded from neural activity cannot be explained simply by the motor features extracted by DLC (see Fig. S5a for color scheme).

We explored potential differences between motor/somatosensory, visual and higher-level cortical areas, focusing on ['SSp-bfd', 'SSp-ll', 'SSp-m', 'SSp-n', 'SSp-tr', 'SSp-ul', 'SSp-un', 'SSs', 'MOp', 'MOs'] for motor/somatosensory, ['VISA', 'VISAm', 'VISI', 'VISli', 'VISp', 'VISpl', 'VISpm', 'VISpor'] for visual regions and ['ACAd', 'ACAv', 'ORBvI', 'ORBm', 'ORBI', 'FRP', 'RSPagl', 'RSPd', 'RSPv', 'ILA', 'PL'] for higher-level cortical regions. We conducted a one-way ANOVA to test for differences between these groups, which yielded no significant results (F-statistic=0.84, p-value=0.434). Similarly, the non-parametric Kruskal-Wallis test also supported this finding (statistic=3.02, p-value=0.221). Therefore, we were unable to demonstrate significant differences in decoding scores across these regional types after adjusting for the embodied prior.

Residual decoding R^2 for region-sessions, dissociating between three types of areas: somatomotor, higher-order and visual regions. A one-way anova revealed no effect of the region identity on the residual decoding R^2 (F-statistic=0.84, p-value=0.434). All region-sessions were considered here, including those that did not significantly reflect the prior.

This rather comprehensive analysis did not yield any detectable effect, possibly due to the inclusion of all data irrespective of their significance. Restricting our focus to only those region-sessions that significantly reflected the prior, we dealt with a much smaller dataset: 6 region-sessions for somatomotor, 4 for higher-order, and 2 for visual regions. However, this limited analysis also showed no significant effect (F-statistic=0.0403, p-value=0.961), which may simply be a

consequence of the very small number of sessions analyzed. Consequently, we believe our current dataset is inadequate for conclusively addressing this question. Targeted recordings with a specific focus on this question would be required for a more definitive answer.

Given our inconclusive results, we have not included this analysis in the manuscript.

Similarly, is prior decoding possible from residuals after regressing out the contribution of both uninstructed movements and eye position?

We have now conducted this analysis, regressing out both motor features (i.e., uninstructed movements) and eye position, and found that prior decoding remains possible (note that including both motor features and eye positions results in a loss of about 20% of the region-sessions compared to the analysis that only considered motor features).

As we had observed after regressing out only the motor features, there is a correlation between the R^2 decoded from the combined motor features and eye position and the R^2 for the prior decoded from neural activity. Moreover, and critically, we found a significant correlation between the residual prior score and the prior score itself, indicating that prior decoding remains viable even after removing the influence of both motor features and eye position.

Embodiment analysis accounting for both the DLC features and the eye position. Left: Decoding R^2 for the Bayes-optimal prior from neural activity against decoding R^2 for the Bayes-optimal prior from DLC features and eye position. The correlation between these two quantities is significant (Pearson correlation $R=0.24, p=2.5E-10$). Right: DLC + eye position residual decoding R^2 against neural decoding R^2 . The residual decoding R^2 values are obtained by first regressing the Bayes-optimal prior from DLC features and eye position, and then regressing the prior residual (Bayes-optimal prior minus Bayes-optimal prior estimated from DLC features and eye position) against neural activity. The neural decoding R^2 corresponds to the R^2 when decoding the Bayes-optimal prior from neural activity. The two

quantities are strongly correlated (Pearson correlation $R=0.79$, $p=5.5E-141$), suggesting that the prior can not be entirely attributed to a combination of both DLC features and eye position.

We now include these two figures in the supplementary figure 10 and have added the following sentence in the main text:

To enhance the robustness of our analysis further, we repeated the embodiment study, this time also including eye position data (on sessions on which these were available). This additional step demonstrated that the neural prior could not be entirely attributed to a combination of both motor features and eye position (see Fig. S10b and Fig. S10c for the feature importance).

3. Prediction of the behavior: Figure 2e nicely shows that the decoded prior is predictive of the animal's decision. How does this observation hold true for individual regions or systems (sensory, motor, "higher" cortical)?

To answer this question, we performed the same analysis, but now at the region level. This involved extracting the decoded Bayes-optimal prior (the prior probability that the stimulus will appear on the right side) for each region and computing the slope of this decoded prior as a function of the proportion of rightwards choices (corrected using pseudo-sessions).

These region-level corrected slopes are shown below for Ephys and Widefield.

Proportion of right choices on zero contrast trials as a function of the decoded region-level Bayes-optimal prior. We decoded the Bayes-optimal prior for each region and computed the slope of this decoded prior as a function of the proportion of right choice (corrected using pseudo-sessions). This is the analysis presented in Fig 2e but at a region level. **a**. Region-level corrected slopes for Ephys and Widefield (significance is assessed when the region-level p -values < 0.05 , using Fisher's method for combining p -values). We observed that the slopes are significant in 17.8% of the regions in Ephys and 90.1% in Widefield, spanning every level of the hierarchy, including LGd, SCm, CP, MOs, and ACAd. It should be noted that the analysis for Ephys includes only 241 regions due to the exclusion of two sessions where the mouse made the same choice on every zero contrast trial.

We observed that the slopes are significant in 17.8% of the regions in Ephys and 90.1% in Widefield, spanning every level of the hierarchy, including LGd, SCm, CP, MOs, and ACAd. Additionally, we identified a correlation at the regional level between the decoded R^2 values and the corrected slopes.

Correlation at the regional level between the decoded R^2 values and the corrected slopes. We find a correlation in both modalities.

This means that the more the prior is reflected in a particular region, the more it predicts the animal's decision, suggesting that the decoded prior is relevant for behavior. These correlations prompt further investigation into whether they could be explained away by differences in how the Bayes optimal prior versus the action kernel model account for behavior across sessions. Specifically, sessions that more closely follow the action kernel model could potentially show lower corrected R^2 and slopes, as these metrics are calculated using the Bayes optimal prior. In Ephys, we found no correlation between the log Bayes Factor (the difference in the marginal log likelihood between the action kernel and Bayes optimal models at the session level) and the corrected slopes (Spearman correlation: $R=0.05$, $p=0.29$, $N=412$ sessions), with the corrected slopes averaged across regions for each session. In widefield, a small correlation was detected (Spearman correlation: $R=-0.34$, $p=0.014$, $N=51$ sessions). However, even after adjusting for the log Bayes factor (by removing the linear prediction of the log Bayes factor from the corrected slope), the correlation between the corrected R^2 and the adjusted corrected slope remained strong (Spearman correlation: $R=0.935$, $p=4.7E-15$, $N=32$ regions). This suggests that the type of behavioral strategy the mice used does not confound the correlation between the corrected R^2 and the corrected slope.

We have added these figures as a supplementary figure S9 and added the following sentence to the main text:

Further analysis at the regional level (Fig. S9a) shows a significant relationship in 17.8% of Ephys regions and 90.1% of Widefield regions across all hierarchical levels (LGd, SCm, CP, MOs, ACAd). Additionally, regions that more strongly reflect the prior were more predictive of the animal's decisions, suggesting the behavioral relevance of the decoded prior (Fig. S9b).

4. When merging WFI and Ephys data (Figure 2b), the results are presented in the same brain space as for the electrophysiology data. Wouldn't it make sense to mask deep regions that are not recorded by WFI? Generally, it is hard to grasp what we gain from this mixed map, compared to Fig 2c, and if it is interpretable (especially as in this combinatorial map additional units were included that were not separately analyzed elsewhere).

The left panel of Fig 2b presents results from each recording modality without controlling for multiple comparisons. This may raise concerns about the robustness of the results across all areas, given the lack of correction for multiple comparisons.

To address this, we had previously leveraged all available data, including all Ephys recordings—even those considered to be multiunits—and integrated them with the widefield data before correcting for multiple comparisons.

In the revised version of the manuscript, because we have more data and improved spike-sorting quality, we decided to focus only on units that met quality control (excluding in particular suspected multiunit data). The updated right panel of Fig. 2b shows the results when aggregating the new Ephys data with widefield results (and as before after correcting for multiple comparisons).

It remains true that the resulting merged map is heterogeneous since the results for the dorsal cortical areas involve two recording modalities, while the other brain regions only rely on Ephys data. However, in this figure, the brain regions which are solely based on Ephys recordings are now corrected for multiple comparisons and, in that respect, complement the uncorrected results shown on the left side of Fig 2b.

We apologise for having not properly explained this point in the original manuscript. We have therefore adjusted the figure caption and main text to ensure that the reader understands the hybrid nature of the merged map and the difference with the pure Ephys map:

Swanson maps of cross-validated corrected R2 for areas that have been deemed significant (using Fisher's method for combining p-values, see Methods) Left: Swanson map of R2 for Ephys data. Right: R2 across Ephys and WFI. For the left map, a region is deemed significant if the Fisher combined p-value is lower than 0.05. For the right map, combining Ephys and WFI, significance is assessed with the Benjamini-Hochberg procedure, correcting for multiple comparisons, with a conservative false discovery rate of 1%. 30.2% (left) and 24.0% (right) of the areas encode the prior significantly, at all levels of brain processing in both cases.

5. **Decoding window:** The results and conclusions are mostly based on one decoding window (-600ms to -100ms). It would be a valuable addition to see how the overall conclusion (number of regions with significant decoding and types of regions) changes by varying the decoding window. It would be especially interesting to get an impression of the regions with significant decoding during the time where there is by definition of the task no wheel movement (-200ms to 0s). This could be included in Figure 3, where the authors already demonstrate similar results in a post-stimulus time window.

In response to your suggestion, we have expanded our analysis to explore the impact of varying the decoding window. We conducted a supplementary analysis with the narrower window (-400ms to -100ms) in which the wheel velocity is strictly zero (by task design) and presented the results in Supplementary Figure S6. Note that the windows does not extend all the way to 0ms because, on some trials, the stimulus can appear slightly before 0ms due to hardware and software constraints. To ensure that our prior decoding is never contaminated by the stimulus, we decided to stop the decoding windows at -100ms.

Swanson maps showing corrected decoding R^2 values for the Bayes-optimal prior on a narrower time window (-400ms to -100ms),

This analysis reveals that 25.6% of brain regions show significant decoding activity in this reduced window, encompassing all levels of brain processing. Interestingly, the overall distribution of significant regions and the Swanson map remain similar between the extended and restricted windows, as evidenced by the very strong correlation between decoding scores ($R=0.90$, $p=6.7e-90$).

Correlation analysis comparing Bayes optimal decoding from the extended window (shown in Fig. 2b) and the Bayes-optimal prior from the narrower window

As anticipated, decoding from a narrower time window yielded results that were not as robust as those from the extended window. This was evident from both a lower percentage of significant regions (25.6% compared to 30% in the extended time window) and the region-level scores comparison (Wilcoxon test $t=10292$, $p=5.2e-5$).

Comparison of the corrected R^2 between the Bayes-optimal prior from main Fig. 2b and the Bayes-optimal prior on a narrower time window (-400ms to -100ms).

We added this following sentence to the manuscript:

we decoded the Bayes-optimal prior [...] and the Bayes-optimal prior on a narrower decoding time window (from -400 to -100ms). [...] When decoded from a narrower time window (-400ms to -100ms),

the Bayes-optimal prior was still significantly decoded across all brain processing levels, albeit with reduced overall decodability (25.6% of regions, 62/242 regions, Fig. S6).

- 6. Different ITIs: A concern regarding the generalizability of the conclusions is to what degree the very rapid task structure contributes to the observed effects, i.e. on how many past actions do contribute to the prior decoding. The large number of analyzed behavior trials may allow for comparing prior information encoding during trials with longer versus shorter ITIs at the behavioral as well as neuronal levels. Is prior encoding in the brain sensitive to the length of the ITI ?**

We examined the impact of inter-trial interval (ITI) lengths on behavioral learning by fitting learning rates based on whether the preceding ITI was longer or shorter than the median ITI for each session.

Following the findings by Iigaya et al. (<https://www.nature.com/articles/s41467-018-04840-2>), we expected that choices following shorter inter-trial intervals (ITIs) might involve narrower integration windows, characterized by higher learning rates and aligning more with a win-stay-lose-switch strategy. Conversely, choices after longer ITIs were predicted to demonstrate broader integration windows, leading to lower learning rates, typical of conventional reinforcement learning.

Our results confirm these expectations: the learning rate for short ITIs was 0.21 ± 0.005 , and for long ITIs, it was 0.18 ± 0.004 . The significant difference between the two was supported by a Wilcoxon test ($t=2538$, $p=9.96E-7$).

Learning rates determined at the session level based on the action kernel model, with different rates applied to trials following long versus short inter-trial intervals (ITIs). For this analysis, ITIs are categorized as long or short relative to the session's median ITI, classifying them as longer or shorter than this median value, respectively.

We then performed a neural analysis where we compared the two action kernel models: one with a single learning rate (as detailed in the paper) and another (previously introduced) with two learning rates, one for trials following short, and one for trials following long ITIs. We anticipated that the dual-rate model would yield better decoding results due to its better account of behavior. However, the findings were not as clear-cut as expected.

Comparison of the region Level R^2 between two action Kernel models: one with a single or one with two learning rates.

Although the trend was in the expected direction (Wilcoxon test, $t=13496$, p -value = 0.16 in favor of the dual learning rate models), the results were not significant. This lack of significance could stem from several factors. For instance, our method of fitting all trials with the same weights potentially overlooks differences between fast and slow learning systems. Additionally, the inconsistent sampling of ITIs, with longer ITIs more prevalent in the latter part possibly due to confounding factors like satiation, could also bias the results.

Given the inconclusive nature of these findings, we decided not to report them in the revised manuscript.

7. PSTHs: The PSTHs shown in Figure S3 for significant neurons are very intuitive. It would be interesting to see how the firing rates of these neurons look specifically during time points when the prior is stable (trials at the end of a block) versus unstable (90 trials where there is no bias in stimulus presentation, or the trials just after a block switch).

Unfortunately, it is very hard to extract robust results from cases with limited numbers of trials (90 trials do not suffice) and fast changes (block switches).

Thus, we explored an alternative approach. In Figure S4, we currently present PSTHs differentiated by the Bayes optimal prior's value being above 0.7 or below 0.3, indicating evidence for right versus left block, respectively.

Additionally, we now introduce here a second criterion for differentiation: uncertainty. This is achieved by performing a median split of the trials based on the distance between the prior and 0.5. Trials with Bayes-optimal priors nearest to 0.5 are classified as low-certainty (unstable prior), and those furthest as high-certainty (stable prior), ensuring an equal number of trials for each category. This analysis demonstrates distinct firing patterns: no noticeable differences in low-certainty trials across the six regions, whereas in high-certainty trials, a systematic modulation of firing rates is observed, highlighting how single neuron activity varies with the uncertainty of the prior.

Figure S4. Six examples of neurons encoding the Bayes-optimal prior significantly (** $p < 0.001$, ** $p < 0.01$, * $p < 0.05$). **a.** Peri-Stimulus Time Histograms (PSTHs) segmented by trials throughout the session. Left column conditions on the Bayes-optimal prior for the right side being less than 0.3 (blue) vs greater than 0.7 (orange). The middle and right columns depict PSTHs for trials under conditions of low certainty (p_{Right} close to 0.5) and high certainty (p_{Right} far from 0.5), respectively. Significance is assessed by testing the difference between the trial wise firing rates (averaging across time bins) of “left” (blue) and “right” (orange) trials with a two-sample Kolmogorov-Smirnov test. **b.** Spike counts of the neurons (purple line) during the intertrial interval in the $[-600, -100]$ millisecond time window before stimulus onset, along with the Bayes-optimal prior (blue) for a subset of trials within the session (Spearman correlations of the full session are reported on the graphs). All neurons on this panel show a preference for the left side though, at the population level, we did not observe a bias for either the right or left side. Indeed, we examined the distribution of decoding weights and detected

no discernible trend concerning the weight distribution. Testing the significance of the decoder weight in each region yielded adjusted p-values all above 0.2 (Wilcoxon test), after adjusting for multiple comparisons using the Benjamini-Hochberg correction. Additionally, a combined analysis of all weights from the six regions lead to the same conclusion (two tailed signed Wilcoxon test: $t=16732$, $p\text{-value}=0.31$).

- 8. On a similar note, the authors in each session have data from 90 trials where the probability of stimulus presentation is at 50%. The authors could provide control plots of the behavior on 0 contrast trials during these 90 trials to show that no bias exists (in Fig 1c).**

Indeed, each session starts with 90 trials during which the prior probability is set to 0.5. In principle, the animals may set their subjective prior to 0.5 but this would require that the animals understand the special status of the first 90 trials which is not guaranteed.

To investigate whether mice bias their subjective prior during the first unbiased trials, we estimated the psychometric shift during both the first 90 trials (unbiased) and the others trials (biased). This shift is determined by analyzing two psychometric curves, one conditioned on the action kernel prior being above 0.5 (favoring the right side) and the other conditioned on the action kernel prior being less than 0.5 (favoring the left side). We fitted psychometric functions to these curves (using the psychofit toolbox), and then calculated the psychometric shift as the vertical displacement of these curves at zero contrast.

The analysis of both the action kernel model and the mice revealed a significant positive psychometric shift during the unbiased phase (first 90 trials). This shift was less pronounced during the unbiased phase than during the biased period for both groups. This shift decrease during the unbiased phase can be explained by the subjective priors, which were closer to 0.5, mirroring the true block prior set at 0.5 for this period. Specifically, when distinguishing the trials that favor the right side (action kernel prior above 0.5) from those favoring the left side (action kernel prior below 0.5), the underlying action kernel priors remained close to 0.5 during the unbiased period.

However, the presence of significant and comparable shifts between the animals and the action kernel model during the unbiased period indicates that mice exhibit a behavioral shift during the unbiased trials, adjusting their behavior in response to the updated priors.

Psychometric shift during both the first 90 trials (unbiased) and the others trials (biased) for animals and the action kernel. This shift is determined by analyzing two psychometric curves, one conditioned on the action kernel prior being above 0.5 (favoring the right side) and the other conditioned on the action kernel prior being less than 0.5 (favoring the left side). We fit psychometric functions to these curves (using the psychofit toolbox), and then calculate the psychometric shift as the vertical displacement of these curves at zero contrast.

We have added this panel to Supplementary figure S15 where we perform the model comparison establishing that the action kernel model best explains behavior. Additionally, we have added this sentence to the main text:

Consistent with the action kernel model, mice updated their subjective prior on the first 90 unbiased trials, even though the true prior is set to 0.5 during that phase (supp Fig S15d)

Additionally, to test whether mice assume that the first 90 trials are biased or not, we fitted two models to the behavior of the mice, both based on the Bayes optimal prior. One assumes that mice do not estimate the prior on the first 90 trials and just uses a fixed value of 0.5, and another assumes that mice use a subjective prior on the first 90 trials. This second Bayes Optimal model assumes no differences between biased and unbiased trials and considers them all as biased trials. Our results decisively support the latter model (Bayesian Model Selection, $p_{\text{exceedance}} > 0.999$, in Supplementary Information).

Bayesian Model Selection (BMS) comparing two Bayesian models: one that assumes the first block to be an unbiased block but with the same statistics as the biased blocks (ignoring unbiased structure), and one that assumes the first block to be unbiased and of length 90 (accounting for unbiased structure). Left is the model frequency and right gives this exceedance probability. The BMS favors the model that ignores the structure of the first unbiased block (pexceedance > 0.999)

Minor points:

1. Can the authors comment on the inversion of prior representation in the VISp example neuron compared to the other regions (Figure S3). Is this a region-wide effect?

We observed no general trend in the weight distributions across the regions of interest initially presented in Figure S3 (now Figure S4): ORBvl, ACAd, SCm, MOs, VISp, and LGd. Individual analyses of these regions, along with significance testing of the weights, revealed no bias toward either positive or negative weights. Wilcoxon tests for each region, adjusted for multiple comparisons using the Benjamini-Hochberg correction, resulted in p-values greater than 0.2. Furthermore, a combined analysis of all regions yielded a Wilcoxon result of (statistic=16732; p-value=0.31).

In summary, the decoding weight distribution for VISp does not show a biased pattern to the right or to the left.

The initial inversion observed in VISp was due to selection bias in our coding setup. We have now adjusted our reporting to include only those units consistent in direction, and we note in the legend that no trend was detected concerning the sign of the weights.

2. In Figure S8, the authors raise an interesting point of embodiment but could analyze in more detail which DLC parameters do successfully decode the prior (whisking, posture etc.)

Following the reviewer's comment, we conducted a detailed analysis of the DLC parameters that successfully decode the prior. This includes an examination of motor features (DLC parameters) and eye position, as previously suggested by the same reviewer (see major comment 2 from this reviewer, where our findings confirm that prior decoding is still achievable when accounting for both motor features and eye position). For the detailed analysis, we employed a regressor elimination approach: for each feature, we removed it to measure the decrease in the decoding score compared to the full model (refer to the figure below)

Regressor elimination approach: for each feature, we remove it to measure the decrease in the decoding score compared to the full model. The first feature to significantly impact the model when removed is the paw position. In this task, the paws are typically engaged to manipulate the wheel, which in turn adjusts the stimulus. It appears that the paws are positioned differently—likely on the wheel—depending on whether the prior suggests the next side will be left or right. The second key feature was the x-coordinate of the eye position, which aligns with the task setup where the stimulus is positioned along a horizontal plane, indicating that the mice tend to look in the direction suggested by the Bayes-optimal prior.

Interestingly, the feature with the most significant impact on the model when removed was the paw position. In this task, the paws are typically engaged to manipulate the wheel, which in turn adjusts the stimulus. It appears that the paws are positioned differently—likely on the wheel—depending on whether the prior suggests the next side will be left or right. The second key feature was the x-coordinate of the eye position, which aligns with the task setup where the stimulus is positioned

along a horizontal plane, suggesting that the mice tend to look in the direction suggested by the Bayes-optimal prior.

We have now added this analysis in Supplementary Figure 10 and reference it in the main text.

3. Colorbar labels in Figure 2b are too small

We have increased the size of the colorbar labels in Figure 2b.

4. The Swanson plots are not very intuitive. For readers familiar with the mouse brain anatomy, it would also help to have R2 maps presented as a set of coronal slices of the Allen Brain Institute reference atlas (in a supplementary figure for example).

We have added corresponding slices to the main decoding results presented in Figure 2b; these can now be found in Supplementary Figure 3. We have also referenced these additions in the main text. After evaluating both coronal and sagittal slices, we decided that the sagittal slices provide a clearer view of the 242 recorded regions.

Figure S3. Encoding of the prior across the brain during the inter-trial interval. Sagittal slices corresponding to the main decoding figure presented Fig. 2b. Left: Ephys only. A region is deemed significant if the Fisher combined p-value is lower than 0.05. Right: Ephys and Widefield combined. Significance for regions is assessed with the Benjamini-Hochberg procedure, correcting for multiple comparisons, with a conservative false discovery rate of 1%.

5. In Figure 3b, the fit seems to depend heavily on the two extreme points. Can you comment on the robustness of the fit?

Indeed, the fitted neurometric curves are strongly dependent on the proportion of decoded right choices for the +1 and -1 contrast. However, we are not so interested in the shape of the tuning curves but rather in the vertical displacement between the two curves at 0 contrast. For instance, in the example shown below and in the manuscript, removing the extrema would not change the fact that the orange dots are well above the blue dots.

Note that whether the neurometric curves are flat or not is mostly a sanity check. Since, in this case, we are decoding choices before the animal has the chance to see the side stimulus and observe its contrast, the neurometric curves must be flat. Non flat neurometric curves would reveal a problem in our experimental design, such as a premature appearance of the stimulus during the time period we decode. Reassuringly, we do not seem to have such a problem.

6. The validity of the statistics depends heavily on Fisher's method, so a short description of the assumptions and validity criteria would be useful for readers unfamiliar with the method.

We have added a description of Fisher's method to the following paragraph in the methods section:

Fisher's Method:

Fisher's method is a statistical technique used to combine independent p -values to assess the overall significance. It works by transforming each p -value into a chi-squared statistic and summing these statistics. Specifically, for a set of p -values (one per session given a region): p_1, p_2, p_3, \dots Fisher's method computes the test statistic:

$$X^2 = -2 \sum_i \ln(p_i)$$

This statistic follows a chi-squared distribution with $2 \cdot N_{sessions}$ degrees of freedom, $\chi^2_{2 \cdot N_{sessions}}$, under the null hypothesis that all individual tests are independent and their null hypotheses are true. If the computed test statistic X^2 exceeds a critical value from the chi-squared distribution, the combined p -value $p(\chi^2_{2 \cdot N_{sessions}} \geq X^2)$ is considered significant and the null hypothesis is rejected.

7. Bar plots such as in Figure 4c,d: violin plots would better convey the shape of the compared distributions

In consultation, we have found that violin plots may not so effectively highlight a critical result: the fact that for the neural prior (right plot), the probability of being correct is lower when following an incorrect action compared to a correct one.

We believe the current version in the manuscript presents this information more clearly, though we have worked on the plot to enhance the visibility of individual data points in the plot.

9. Comment: it would be ideal if the authors could discuss the generalization of their findings. For example, mice follow the action kernel model: is this applicable/valid in other tasks? Of course, acquiring data with another task is not feasible and beyond the scope, but discussing the literature in more detail, or analyzing another publicly available dataset would be an alternative.

We should have indeed been clear that we are not the first to observe a strong influence of past choices on current choice, We now mentioned two studies, one in rodents (<https://doi.org/10.1038/s41593-022-01021-9>) and one in monkeys (<https://doi.org/10.1901/jeab.2005.110-04>) documenting a similar effect.

9. Generally, the discussion is focused on what type of neural code could sustain the prior and could be richer regarding the behavioral strategy of mice and the involvement of specific regions in the “distributed” encoding.

The discussion now starts with a brief summary of the behavioral results. With regard to the second point, we have added the conclusions of a Granger causality analysis (elaborated in response to Reviewer 1) to the discussion, which revealed the presence of multiple communication loops. Unfortunately, we cannot go much beyond this as we lack the data that would allow us to determine more specifically the role of each region.

10. Could the authors discuss why the mice perform worse than the optimal agent and follow an action kernel? That is very interesting. For this simple task, shouldn't we expect optimal behavior?

This is indeed an interesting observation. We can only speculate at this stage as to why this is the case but two explanations come to mind: 1- using this strategy is computationally very simple, since it does not even involve waiting for feedback, and 2- while suboptimal, it results in a reward rate only marginally smaller than the optimal one (1.9%). In fact, during the initial phase of shaping, before the animals have even experienced biased sessions, they already show evidence of an action kernel (Bruijns et al, in preparation). Note that the mice at least appear to choose a near-optimal value for the decay rate of this action kernel.

We now report this marginal loss of performance figure 4 and added the following sentence to the main text:

Remarkably, this is close to the value of the decay constant which maximizes the percentage of correct responses, given this (suboptimal) form of prior, losing only 1.9% compared to the performance of the Bayes-optimal version (Fig. 4b).

Referee #3 (Remarks to the Author):

In their manuscript, “Brain-wide representations of prior information in mouse decision-making”, the authors leverage the rich data set assembled by the International Brain Laboratory to address how prior information about the state of the world is represented in neural circuits. One possibility is that information about prior probability affects choices in the late stages of decision-making, and is therefore represented quite narrowly. Alternatively, probabilistic information about the state of the world may be carried broadly across neural networks. By using standardized neural recordings acquired across many brain regions simultaneously in mice performing a standardized decision-making task, in combination with recordings of motor activity, the authors have a broad data set to explore this question. They use a decoding model to predict the prior probability that a stimulus will be presented on the right side from neurons recorded in the contralateral fore- and midbrain and ipsilateral cerebellum and hindbrain. In the behavioral task, the prior alternates, in blocks of random lengths, between 0.8 and 0.2. Each session begins with a block of 0.5 prior probability. The main finding is a broad representation of the Bayes-optimal prior in neural activity – whether measured as single-unit electrophysiology or widefield imaging. Evidence of the neural signal of the prior persists when taking into account the movement of the animal as well as response history in the task.

While understandable given that many of the methods refer to other papers produced from the consortium – some published, some in preprint, reading this as a stand-alone article is a bit challenging. One correction to provide clarity would be to add a citation for the companion bioRxiv pre-print (<https://doi.org/10.1101/2023.07.04.547681>), which they refer to as the BWM paper in the Methods. Currently, the authors cite the International Brain Lab et al. (2023) data release, but also refer to the BWM preprint using the same citation. Please include the companion paper preprint in the references and clarify throughout which 2023 paper is being referenced.

Compared with the BWM preprint, the authors here take a much deeper dive into what information can be decoded about the Bayes-optimal prior (subjectively ascertained) during the quiescent period before stimulus onset. The results here are quite different from the BWM preprint, which shows very sparse and localized neural decoding of the objective priors. The authors list several methodological differences between the two papers, but it is not obvious how much these contribute:

- What is the impact of the using a longer vs shorter time window?

Thank you for your comment. To look into the impact of using a long vs shorter time window, we performed decoding on a narrower window between -400 and -100ms (instead of -600 to -100ms).

Swanson maps showing corrected decoding R^2 values for the Bayes-optimal prior on a narrower time window (-400ms to -100ms),

This analysis reveals that 25.6% of brain regions show significant decoding activity, encompassing all levels of brain processing as before. Interestingly, the overall distribution of significant regions and the Swanson map remains similar between the extended and restricted windows, as evidenced by a strong correlation between decoding scores ($R=0.90$, $p=6.7e-90$).

Correlation analysis comparing Bayes optimal decoding from the extended window (shown in Fig. 2b) and the Bayes-optimal prior from the narrower window

As anticipated, decoding from a narrower time window yielded results that were not as robust as those from the extended window. This was evident from both a lower percentage of significant regions (25.6% compared to 30.2% in the extended time window) and the region-level scores comparison (Wilcoxon test $t=10292$, $p=5.2e-5$).

Comparison of the corrected R^2 between the Bayes-optimal prior from main Fig. 2b and the Bayes-optimal prior on a narrower time window (-400ms to -100ms).

We have added this following sentence to the manuscript:

When decoded from a narrower time window (-400ms to -100ms), the Bayes-optimal prior was still significantly decoded across all brain processing levels, albeit with reduced overall decodability (25.6% of regions, 62/242 regions, Fig. S6).

- How much difference comes from estimating the Bayes-optimal prior vs. true block prior (based on Fig 2A, S3, animals seem to track the priors quite well - they do not diverge extensively)?

To address this question, we conducted an analysis where we decoded the block prior within the same inter-trial decoding window used in our main analysis (as shown in Figure 2).

Decoding with the true block prior identified 19.4% of significant regions (47 out of 242), compared to 30.2% (73 out of 242 regions) when using the Bayes-optimal prior.

Swanson maps showing corrected decoding R^2 values for the true block prior

In line with these findings, the region-level R^2 values were significantly lower when decoding the true prior as opposed to the Bayes-optimal prior (Wilcoxon test, $t = 7409$, $p = 2.24e-11$). However, and as expected, there was a strong correlation between the two sets of scores ($R = 0.87$, $p = 7.7e-77$).

Comparison of the corrected R^2 between the Bayes-optimal prior from main Fig. 2b and the true block prior.

Correlation analysis comparing Bayes optimal decoding (shown in Fig. 2b) and the true block prior

Overall, decoding the true block prior yielded fewer significant regions, as expected given that the Bayes-optimal prior more closely reflects the mice's subjective prior than the block prior (as shown below).

Bayesian model comparison for 2 behavioral models, the Bayes optimal model, which infers a prior from past observations, and a model that assumes the true block prior, which is not accessible to the mice. Our analysis shows that the Bayes optimal model more effectively explains the behavior, with an exceedance probability greater than 0.999.

We have now included this analysis in Supplementary Figure S6 and referenced the block prior decoding in the main text as follows:

An even smaller percentage of regions (19.4%; 47/242 regions, Fig. S6) were found to encode the prior significantly when decoding the true block prior suggesting that the animal's subjective prior aligns more closely with the Bayes-optimal prior than with the true block prior. This observation is supported by a behavioral analysis, which revealed that a model using the true block as a prior was less effective at explaining behavior compared to the Bayes-optimal model (Fig. S6d)

- To what extent does the power of including more sessions make it possible to see the decoding?

An outstanding question that we can answer with our dataset is the number of recordings necessary per region to achieve significant prior decoding. To address this, we analyzed the number of recordings required to reach the significance threshold in regions identified as significant (according to the main decoding analysis presented fig 2b).

Dual-axis graph displaying the p-value of regional significance as a function of the number of decoding sessions (left axis) and the ratio of significant regions relative to the total number of sessions (right axis). It is estimated that approximately 10 sessions per region are necessary to identify 95% of significant regions highlighted in the main decoding analysis (refer to Methods section for more details). It's important to recognize that this analysis has limitations: it assumes uniformity across recordings and regions without considering, e.g., variations in effect size or number of units per recording. Despite these limitations, we concentrated on the number of recordings because it is a primary factor that experimenters can directly control.

For each region, color-coded according to the Swanson map (Fig. S5a), we randomly sampled a variable number of recordings and tested for significance, repeating this process 1'000 times per region. We report the median p-value across these repeated samplings. This analysis allows us to identify the average number of sessions required to reach the significance threshold for a given region. We then ask how many recordings are required in order to find 95% of the regions to be significant.

We found that approximately 10 recordings per region are necessary to reach this 95% target (see cumulative curve of significant regions shown in black).

When looking at the number of sessions that we have per region, we find a median of 6 recordings per region:

Histogram detailing the number of sessions per region in Ephys

This suggests that the median of six recordings per region falls on the lower end of what is needed to yield robust decoding results.

We have added the following sentence to the manuscript:

An analysis to determine the necessary number of recordings per region indicated that around 10 recordings per region are required to reach the obtained significance levels (see Fig. S5e). Given that the median number of sessions per region in Ephys is 6 (see Fig. S5d), it is likely that the reported levels of significance are underestimated.

While all are speculated as contributing, have the authors quantified the sources of discrepancies – this is relevant for the companion paper as well, which presents a very different result.

This is indeed a critical point, and we have given it thorough consideration. After careful reflection, we have decided to remove the block prior analysis from the companion paper.

To address your question more fully, the key differences between the two papers stem from the following factors:

The companion paper decoded the block prior, while in this paper we decode the Bayes-optimal prior.

The time window for decoding in the companion paper was [-400, -100 ms], whereas here it is [-600, -100 ms].

As outlined in previous responses to this reviewer's comments, switching from decoding the Bayes-optimal prior to the block prior reduces the percentage of significant regions from 30.2% to 19.4%. Additionally, narrowing the time window results in a reduction of significant regions from

30.2% to 25.6%. While both factors contribute to the discrepancies in the percentage of significant regions, the shift from the block prior to the Bayes-optimal prior accounts for the larger portion of the difference.

The authors focus on the 0.8:0.2 blocks of trials in test sessions. Each session begins with 90 trials of 0.5:0.5 priors. Do the areas that decode priors in the 0.8 and 0.2 blocks not decode in the 0.5 block?

Sorry for the confusion. We include all trials for decoding, motivated by our observation that mice use the subjective prior even during the initial 90 unbiased trials, as detailed in our response to Reviewer 2, question 8.

Given this behavioral observation, one would expect that removing the first 90 trials from decoding would slightly degrade our results. This is indeed what we observed. Excluding the first 90 unbiased trials from decoding resulted in a slight decrease in the proportion of significant regions, from 30.2% (73 / 242 regions) to 28.1% (68 / 242), though we did not find a significant difference in region-level decoding R2 (Wilcoxon test, $T=13237$, $p=0.18$).

More broadly for discussion, what does it mean for a brain area to show activity that significantly modulates with prior probability? How should this differ between areas that are more or less heterogeneous? It is not known what types of cells are recorded – projection neurons, interneurons, etc. Might these encode task variables differently, and to what extent does this limit the decoding model?

To our knowledge, there are only a few theoretical proposals as to how specific cell types might contribute to Bayesian inference. Unfortunately, our current dataset does not allow us to test these theories, as we cannot reliably identify cell types using neuropixel recordings or wide-field imaging. Therefore, we regrettably cannot address this topic in the revised manuscript. We, of course, agree that the question is important and interesting.

Also broadly, the results here suggest a large proportion of brain areas carry information about expected probability. These analyses relied on a single, relatively long (in neural coding terms) period of time. Could the data be leveraged in the future to show temporal dynamics among the many areas that show biased activity in line with the action kernel/bayes-optimal prior? This kind of temporal resolution should be available in the Ephys data and provide a jumping off point for direct comparisons of time-series differences between recording techniques.

Indeed, much of our analysis has utilized an extended time window from -600ms to -100ms before stimulus onset. We examined the impact of a narrower decoding window, spanning from -400ms to -100ms prior to stimulus presentation.

We thus performed decoding on a narrower window between -400 and -100ms (instead of -600 to -100ms).

Swanson maps showing corrected decoding R^2 values for the Bayes-optimal prior on a narrower time window (-400ms to -100ms),

This analysis reveals that 25.6% of brain regions show significant decoding activity, encompassing all levels of brain processing as before. Interestingly, the overall distribution of significant regions and the Swanson map remains similar between the extended and restricted windows, as evidenced by a strong correlation between decoding scores ($R=0.90$, $p=6.7e-90$).

Correlation analysis comparing Bayes optimal decoding from the extended window (shown in Fig. 2b) and the Bayes-optimal prior from the narrower window

As anticipated, decoding from a narrower time window yielded results that were not as robust as those from the extended window. This was evident from both a lower percentage of significant regions (25.6% compared to 30.2% in the extended time window) and the region-level scores comparison (Wilcoxon test $t=10292$, $p=5.2e-5$).

Comparison of the corrected R^2 between the Bayes-optimal prior from main Fig. 2b and the Bayes-optimal prior on a narrower time window (-400ms to -100ms).

Our findings indicate that the conclusions remain consistent even within this more constrained time frame. We added this following sentence to the manuscript:

we decoded [...] the Bayes-optimal prior on a narrower decoding time window (from -400 to -100ms). [...] When decoded from a narrower time window (-400ms to -100ms), the Bayes-optimal prior was still significantly decoded across all brain processing levels, albeit with reduced overall decodability (25.6% of regions, 62/242 regions, Fig. S6).

We have found that further shrinking the decoding time window considerably reduces the quality of the decoding, making it difficult to reach any conclusions. However, we have applied a Granger causality analysis on both electrophysiology (Ephys) and widefield imaging data to help us map the flow of prior information during the inter-trial interval (ITI), providing insights into the underlying network dynamics (also elaborated in response to the comment 2 of Reviewer 1).

Crucially, we first project the neural activity from each brain region onto the Bayes Optimal prior decoding axis. This projection involved decoding neural activity from 50ms time bins in Ephys, and from each frame in Widefield (66ms per frame), to create time-series of decoded prior information for each area. Granger causality was then assessed on these time series of decoded prior information from each region. Similarly to all the other analyses in the paper, the null distribution is defined here by applying the same pipeline to pseudo priors - see Methods for more details.

Our analysis first reveals notable communication concerning the prior between regions.

Average percentage of significant directed pairs between 2 regions that reflect the prior across sessions. When considering all pairs of regions encoding the prior significantly, and for which we had simultaneous recordings, we observed that information was significantly exchanged between 71% of these pairs in Widefield imaging and 36% in Ephys.

When considering all pairs of regions encoding the prior significantly, and for which we had simultaneous recordings, we observed that information was significantly exchanged between approximately 71% of these pairs in Widefield imaging and 36% in Ephys. This level of inter-regional communication significantly surpasses the rates that would be anticipated by chance, indicating a robust pattern of directional information flow between these areas.

When looking particularly at the Granger directed pairs themselves, we find the following graph in Ephys:

All significant directed pairs ($p < 0.05$ uncorrected)

Complete connectivity graph from Ephys ($p < 0.05$ uncorrected for multiple comparison).

It should be noted that the pairs are considered significant if their associated p-value is below 0.05, without applying any corrections for multiple comparisons. However, when adjustments for multiple comparisons are made, none of the links remains significant. This lack of significant findings post-correction is attributed to the sparse nature of the observations in Ephys. With a total of 242 observed regions, the possible number of pairs amounts to $242 \times 241 = 58,322$. Of these, approximately 10% of the directed pairs had been recorded simultaneously, and the vast majority (75%) of these pairs appears in two or fewer sessions, highlighting their sparsity.

Histogram showing the number of sessions for each directed pair and barplot showing the percentage of observed directed pairs (directed pairs with at least one session) versus

unobserved pairs. In Ephys, with a total of 242 observed regions, the possible number of pairs amounts to $242 \times 241 = 58,322$. Of these, approximately 10% of the directed pairs had been recorded simultaneously, but the vast majority (75%) of these pairs appeared in two or fewer sessions, highlighting their scarcity.

To address the issue, we aggregated the data into broader regional categories defined by the Cosmos classification, encompassing Isocortex, Hippocampal Formation (HPF), Cortical Nuclear (CNU), Thalamus (TH), Hypothalamus (HY), Midbrain (MB), Hindbrain (HB), and Cerebellum (CB). By pooling the results across these Cosmos areas and employing Fisher's method to combine the p-values, followed by a Bonferroni correction for multiple comparisons, we revealed a significant flow of prior information throughout the entire mouse brain. This flow occurs both from subcortical to cortical regions, such as from the Isocortex to the Thalamus, and reciprocally, from regions like the Thalamus and Midbrain back to the Isocortex.

All significant directed pairs ($p < 0.05$ Bonferroni corrected)

Connectivity graph in Ephys across Cosmos regions ($p < 0.05$ Bonferroni corrected). Effect sizes across directed pairs of regions are aggregated at the Cosmos level with Fisher's method

Widefield provides a richer dataset, because, with 32 regions recorded simultaneously, we can analyze a total of 992 possible directed pairs (32×31), most of them available on all sessions.

Histogram showing the number of sessions for each directed pair and barplot showing the percentage of observed directed pairs (directed pairs with at least one session) versus unobserved pairs. Widefield provides a richer dataset, because, with 32 regions recorded simultaneously, we can analyze a total of 992 possible directed pairs (32×31), most of them available on all sessions.

The complete connectivity graph from the Widefield data, corrected for multiple comparisons using the Bonferroni method, is densely populated and consequently difficult to interpret.

All significant directed pairs ($p < 0.05$ Bonferroni corrected)

Complete connectivity graph from Widefield ($p < 0.05$ Bonferroni corrected).

However, by concentrating on the directed pairs projecting to the primary visual cortex, we uncover feedback connections from higher-order areas—such as the Motor Cortex (MOs), Ventral Retrosplenial Cortex (RSPv), Prelimbic Cortex (PL), and Anterior Cingulate Area Dorsal (ACAd)—to the early sensory area, the Primary Visual Cortex (VISp).

Significant directed pairs to VISp ($p < 0.05$ Bonferroni corrected)

A partial connectivity graph from Widefield, highlighting significant directed pairs projecting to the Primary Visual Cortex (VISp). Consistent with predictions from a Bayesian network model of the brain, we uncover feedback connections from higher-order areas such as the Motor Cortex (MOs), Ventral Retrosplenial Cortex (RSPv), Prelimbic Cortex (PL), and Anterior Cingulate Area Dorsal (ACAd) — these regions are marked with grey circles for emphasis — to the early sensory area, the Primary Visual Cortex (VISp).

Up to this point, we have demonstrated that the flow of prior information between regions exceeds what would typically be expected by chance. This flow encompasses communications throughout the entire brain and includes feedback connections within the cortex, from higher-order to early sensory areas.

We can now investigate whether this flow of prior information forms loops. By examining how often a significant directed pair (A->B) is reciprocated within the same session by its counterpart (B->A), we found this to occur approximately 38% of the time in Widefield and 11% in Ephys. In both instances, these rates significantly exceed what would be anticipated by chance.

Average percentage of significant directed pair (A->B) which is reciprocated within the same session by its counterpart (B->A), we found this to occur 38% of the time in Widefield and 11% in Ephys.

We investigated the existence of loops in the connections identified in Widefield imaging between the primary visual cortex (VISp) and higher-order areas—Motor Cortex (MOs), Ventral Retrosplenial Cortex (RSPv), Prelimbic Cortex (PL), and Anterior Cingulate Area Dorsal (ACAd), as illustrated in the second-to-last figure. We focused on sessions where both VISp and a higher-order region (A) significantly reflected the prior, computing the percentage of these sessions that exhibited a significant reciprocal connection pattern (A->VISp->A):

Percent of sessions with a significant loop (A->VISp->A)

Percentage of sessions exhibiting significant reciprocal connections (A->VISp->A) for sessions in which the Bayes optimal prior could be significantly decoded from both VISp and the previously identified higher-order regions (MOs, RSPv, PL and ACAd). The size of the arrow is proportional to the percentage. Our findings indicate the existence of reciprocal connections in these sessions:

33.3% between MOs and VISp, 16.7% between ACAd and VISp, 20% between PL and VISp, and 18.75% between RSPv and VISp.

Our analysis revealed the presence of significant loops for each region: 33.3% of sessions showed a significant loop between MOs and VISp; for ACAd and VISp, the percentage was 16.7%; for PL and VISp, 20%; and for RSPv and VISp, 18.75%.

In conclusion, by employing Granger causality analysis on both Ephys and wide-field imaging data, we confirmed that the subjective prior is being communicated back and forth between regions at all levels of processing. We thank the reviewer for inspiring this new analysis which nicely reinforces our original conclusions that the brain is akin to a loopy Bayesian network.

These results are all reported in the main manuscript, along with new supplementary figures S11 along with the following paragraph in the main text:

Our decoding analysis reveals a robust, distributed representation of the Bayes-optimal prior throughout the brain, suggesting a complex network of information flow. To investigate the dynamics of the prior information network, we conducted a Granger Causality analysis during the ITI, between the time series of the decoded prior from one brain region and that of another (see Methods and Supp Fig. S11). This analysis revealed several key findings: 1) The flow of prior information between brain areas is significantly greater than expected by chance (Supp Fig. S11a), 2) This prior flow includes comprehensive communications across the entire brain, from subcortical to cortical areas and vice versa (Fig. 2g left panel), 3) It includes significant feedback connections from higher-order areas to early sensory areas (Fig. 2g right panel), and 4) There is a higher prevalence of loops within this communication network than would be anticipated by chance (Fig. 2h), including between higher-order and early sensory areas (Supp Fig. S11e). These results collectively highlight a loopy and intricate inter-area communication of prior information within the brain.

Very minor suggestion: It is difficult to make quantitative comparisons across the Ephys and WBI analysis. It would be easier to see both the relationships and their magnitudes by equating the axes in Figures 2e, 3d, 4d, 4e, 4f.

The analysis derived directly from neural activity presents quantitative results that are inherently difficult to compare, largely due to greater variability in electrophysiological data. Aligning the axes across all figures, as suggested, could compromise the visibility of data in Widefield imaging. Consequently, we hope that it is satisfactory to maintain different axis scales for Figures 3d, 4d, and the x-axis of 4f, mostly to ensure clarity and visibility of the Widefield data. The other axes have been equated.